# Light-modulated stem cells in the camera-type eye of an annelid model for adult brain plasticity

Nadja Milivojev [1,2,3], Federico Scaramuzza [1,2], Pedro Ozório Brum [1,4,5], Camila L. Velastegui Gamboa [1,8], Gabriele Andreatta [1,3,9], Florian Raible [1,3] ✉ & Kristin Tessmar-Raible [1,6,7] ✉

Camera-type eyes in vertebrates and cephalopods are striking examples of parallel evolution of a complex structure. While comparisons have focused on these two groups, camera-type eyes with likely high functionality are also found in other invertebrate phyla with simpler brains. Employing single-cell RNA sequencing, we identify neurogenic cells in the adult eyes and brain of the marine annelid worm *Platynereis dumerilii*. Distinct neural stem cells in the camera-type adult eyes, located at the edge of the cup-shaped retina, and adjacent to the glass body/lens, produce radial lines of cells, reminiscent of stem cells in ciliary marginal zones of vertebrate eyes exhibiting life-long growth. Normal proliferation in the eye depends on ambient light, a phenomenon that depends on the integrity of the photoreceptor gene *c-opsin1*, which is present in emerging rhabdomeric photoreceptors, and impacts on their differentiation. During reproductive maturation, proliferation in the eye as well as the entire brain sharply declines, while cells upregulate molecular characteristics of mammalian adult neural stem cell quiescence. Our data provide insights into the development and modulation of annelid head and brain cells, revealing similarities and differences to vertebrate eye development, neurogenesis and brain plasticity.

For evolutionary research, camera-type eyes serve as textbook cases of parallel evolution. Most focus has been on comparison between the eyes of cephalopods and vertebrates, two groups considered to have advanced sensory and cognitive abilities[1,2]. However, camera-type eyes also exist in simpler animals such as annelid worms[3–5], model systems well suited to probe the evolution of nervous systems[6]. Notwithstanding their seeming simplicity, selected annelid eyes have recently been demonstrated to provide high-resolution vision comparable to vertebrates, cephalopods or insects[7]. Similar to cephalopods and anamniote vertebrates, annelid eyes and nervous systems exhibit lifelong growth, which requires a tight regulation to maintain functional accuracy. In the eyes of those vertebrates exhibiting life-long growth, a dedicated stem cell area, termed the ciliary marginal zone (CMZ), is the source of new neurons and pigment cells[8–10]. During

[1]Department of Neurosciences and Developmental Biology, Faculty of Life Sciences, University of Vienna, Vienna, Austria. [2]Vienna BioCenter PhD Program, Doctoral School of the University of Vienna and Medical University of Vienna, Vienna, Austria. [3]Research Platform Single-Cell Regulation of Stem Cells (SinCeReSt), University of Vienna, Vienna, Austria. [4]Institute of Molecular Biotechnology of the Austrian Academy of Sciences (IMBA), Vienna, Austria. [5]Vienna Doctoral School in Cognition, Behaviour and Neuroscience (CoBeNe), University of Vienna, Vienna, Austria. [6]Alfred Wegener Institute Helmholtz Centre for Polar and Marine Research, Bremerhaven, Germany. [7]Carl von Ossietzky University Oldenburg, Oldenburg, Germany. [8]Present address: Institute of Science and Technology Austria, Klosterneuburg, Austria. [9]Present address: University of Padua, Department of Comparative Biomedicine and Food Science, Legnaro, Italy. ✉e-mail: florian.raible@univie.ac.at; kristin.tessmar-raible@univie.ac.at

adulthood, fish retinal stem cells from the CMZ contribute to eye size increase by precise radial divisions at rates matching the required shape, thereby assuring eye functionality during continuous growth[9]. The characterisation of adult neuronal stem cells (aNSCs), including those in the retina, significantly advances our insight into the neurogenic plasticity of eyes and brains and how they contribute to functional organ maintenance during adult life.

A common denominator of vertebrate development is the decline of active NSCs with increasing age, albeit the respective timing differs. Even in teleosts, prime examples for lifelong growth, retinal stem cells decline with aging[11]. In sharks, retinal neurogenesis is downregulated upon sexual maturation[12], while adult roles for retinal stem cells are limited in birds, and lost in postnatal stages of mammals[10,13]. Like the eyes, also the adult vertebrate brain is constantly modified by the addition of new neurons. In fishes and amphibians, aNSCs support lifelong brain growth[14], while in the mammalian brain, aNSCs are not only confined to specific niches, but the ability of aNSCs to enter into neurogenic states is further limited by quiescence[15]. Active versus quiescent neural stem cell states are characterised by specific molecular signatures of receptor signalling activity, transcription and translation[16]. These are tuned by poorly understood internal and environmental signals.

Whereas adult neuronal stem cells are relevant for the regulated growth of eyes and defined brain regions in vertebrates, there are only scarce comparative data for neurogenic processes in invertebrates. In several invertebrate groups, enriched environmental stimuli can impact on neurogenesis in related processing centres: in red flour beetles, but not the fruitfly, the adult mushroom bodies retain some neurogenic potential, which depends on the olfactory environment[17]. Similarly, environmental changes impact on adult neurogenesis in the olfactory lobes of the shore crab[18] as well as the octopus[19]. Regulated changes in neurogenesis along an organism's life history as they occur in vertebrates are even less explored in invertebrates. In ants, cast-specific hormone signatures lead to significant changes in the adult brain, mainly thought to occur via molecular regulation in differentiated neurons, not by changes in neurogenesis[20,21]. Finally, analyses of cephalopod camera-type eyes during early development revealed progenitor cells with interkinetic nuclear migration and an involvement of Notch signalling in differentiation, providing developmental similarities to vertebrates[22]. While additional transcriptomic and expression studies have provided insights into cephalopod optic systems and their cell types[23–25], there is currently no clear understanding if and how aNSCs contribute to the life-long growth and modulation of invertebrate camera-type eyes, nor any insight how these processes can be tuned in response to the internal/external environmental cues.

In order to gain deeper insight into neurogenesis and brain plasticity in an invertebrate model, we selected the marine bristleworm *Platynereis dumerilii*, a valuable model for cross-comparisons of nervous systems of other bilaterian groups[6]. Comparisons of *Platynereis* larval cell types have already helped to clarify the evolutionary relationship of specific neuronal cell types. For instance, *Platynereis* larvae possess two molecularly distinct groups of photoreceptors that are demarcated by the expression of canonical rhabdomeric (*r-opsin*) and ciliary (*c-opsin*) opsin genes, respectively, the latter of which had long been assumed to be a vertebrate innovation[26]. Functional experiments imply *r-opsin1* – expressed in the adult eyes of the worms – in high-sensitivity light detection[27]. By contrast, *c-opsin1* – encoding a UVA/violet-light sensitive opsin[28,29] – acts in larval depth sensing[30], as well as in wavelength ratio detection involved in seasonal UV sensation in adult worms[29]. Generally, *r-opsin*– and *c-opsin*-type *opsins* are thought to be expressed in distinct cell types, with the eyes of onychophorans, basally branching panarthropods, as possible exception[31,32].

Whereas the aforementioned molecular cell type comparisons have largely centred on *Platynereis* larval stages, our interest in brain plasticity let us focus on adult stages and their possible stem cell

systems. The adult brain of *Platynereis* and related nereidid worms has long served as a general annelid reference for classical neurobiology[33]. Adult *Platynereis* heads (schematised in Fig. 1a) harbour two pairs of large camera-type adult eyes that grow over time and have been well investigated on the ultrastructural level[4,34]. Furthermore, adult nereidid brains were among the first invertebrate brains in which neurosecretory cells were characterised[35]. Distinct neurohormonal cues orchestrate a major transition between regenerative, non-reproductive life stages (immature and premature stage) and a non-regenerative, reproductive stage (mature female and male worms)[36–39]. This switch is accompanied by significant changes in the overall head transcriptome[40], suggesting it as a suitable developmental phase to study brain plasticity. Whereas stem cell and regenerative potential in mammals decreases with reproductive maturity and age[41], commonly studied invertebrate model systems like planarians or cnidarians do not exhibit a similar decline of their stem cell capacities[42,43].

Here, we use a combination of single-cell RNA sequencing (scRNA-seq), molecular and proliferation markers, and genetics to characterise the adult *Platynereis* head and investigate its plasticity. After validating several molecularly distinct cell populations, we pinpoint changes in the adult *Platynereis* head and camera-type eye that accompany the transition between the animal's major adult life stages. We identify a bona fide stem cell system in the adult eyes. Molecular markers and the position of these cells at the level of the eye opening and thus at the rim of the cup-shaped retina draw a parallel to the ciliary marginal zone of vertebrate camera-type eyes. Light-deprived animals exhibit reduced proliferation in the eye. Unexpectedly, we find the ciliary opsin gene *c-opsin1* to be co-expressed in the early, proliferative stage of the rhabdomeric photoreceptor lineage. in contrast to wild-type specimens, animals mutated in this light-sensitive opsin don't display light-dependent adult eye growth, and exhibit altered photoreceptor differentiation. This suggests a link between ambient light conditions and the orchestration of adult camera-type eye growth. Finally, the adult *Platynereis* eye and brain harbours different classes of cells with neurogenic potential, which exhibit marked similarities to active and quiescent mammalian aNSCs.

## Results

### Adult brain neurogenesis is regulated during sexual maturation

In order to gain insight into molecularly defined cell populations of the adult *Platynereis* brain, as well as their potential plasticity, we performed scRNA-seq of worm heads (Fig. 1a) of four distinct stages (immature, premature, mature female, mature male). After filtering and matching the sizes of the four libraries, our set comprised 19908 cells and 31811 reference gene sequences, with an average of 671 reference sequences expressed per cell.

To capture changes within cell populations, we combined the individual libraries employing a merging approach[44,45], and represented the data in a uniform manifold approximation and projection (UMAP)-reduced space (Fig. 1b). In the framework of time-resolved scRNA-seq series[44,45], such merging retains biologically relevant differences between datasets, while joining biological replicates for the same condition, exhibiting a sensitivity that we reasoned to be useful for our interest in brain plasticity. Indeed, this approach yielded an overall clustering into major cell populations that was comparable to data integration using the Seurat pipeline[46] (Supplementary Fig. 1–d, f, g), but tended to place cell populations arising from immature and premature libraries adjacent to cell populations arising from mature female and male libraries (Fig. 1b and Supplementary Fig. 1c, d, f). This juxtaposition reflects validated molecular changes over the non-reproductive/reproductive switch (Fig. 1c), such as the *fatty acid binding protein/fabp* and *qpeptin* genes (s.b.) which are downregulated during transition from immature/premature to mature states[40]. Judged by a molecular signature containing the neuronal markers *elav1*, various groups of *synaptotagmins*, *syntaxin*, *snap-25* and *synaptobrevin*

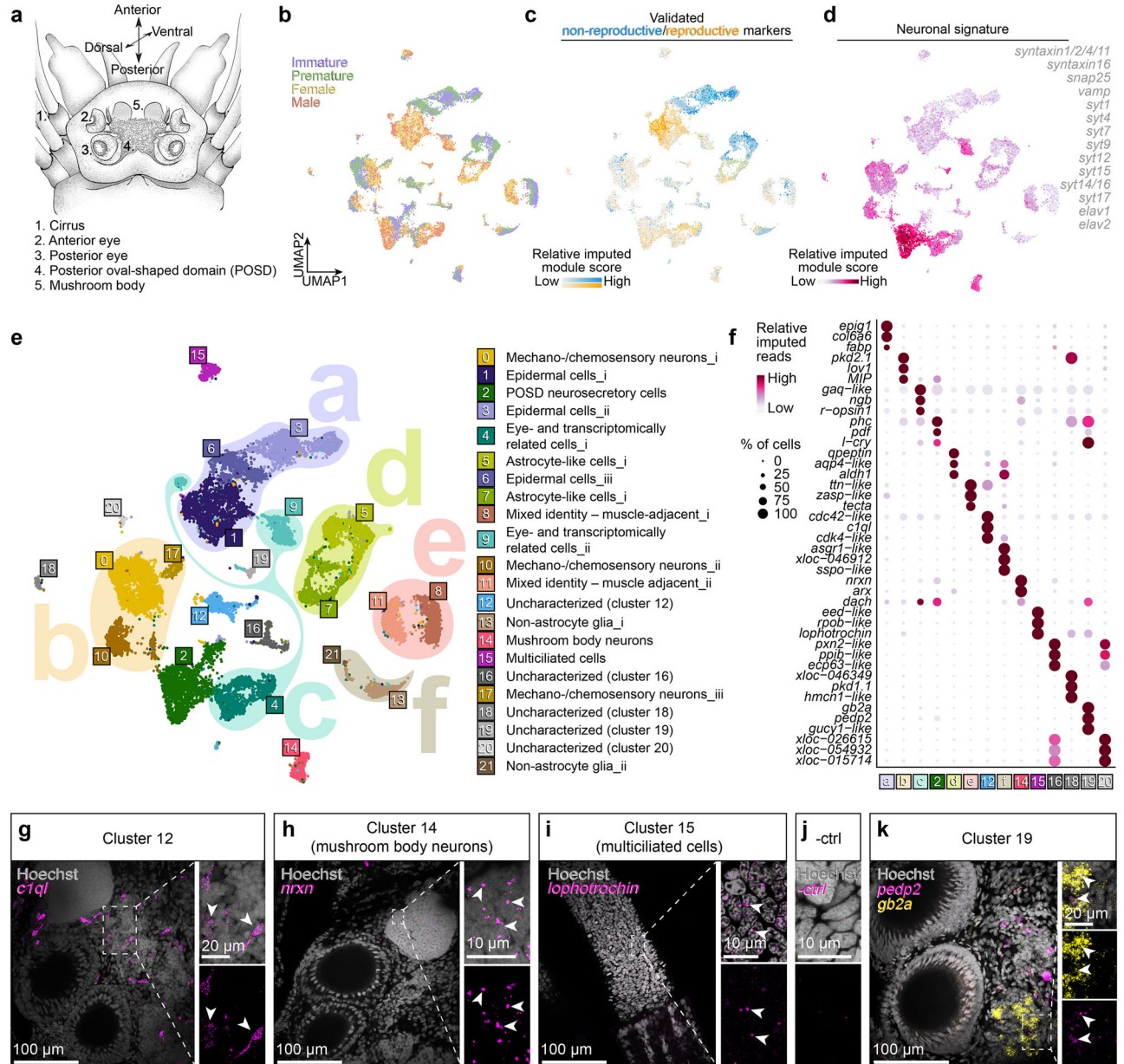

**Fig. 1 | A comprehensive atlas of cell populations of the adult *Platynereis* head, merging single-cell transcriptome data from immature, premature and mature stages. a** Scheme of the anatomy of the adult head. **b–f** Single-cell transcriptomic characterisation of major cell populations of the adult head. **b** UMAP representation of merge-integrated immature, premature, female and male scRNA-seq libraries. **c, d** Visualisation of non-reproductive vs. reproductive cell signatures (**c**) and neuronal cell signatures (**d**) using validated markers (cf. ref. 40). **e, f** Tentative annotation of major cell populations (clusters 0–21) and super-clusters (a-f) of the dataset, along with representative marker genes. See Supplementary Fig1 and Supplementary Data 1 for further details. **g–k** Validation of the indicated markers for distinct cell populations present in the dataset by in situ Hybridisation Chain Reaction (HCR). Thickness of specimens: g: 5 μm; h: 3 μm, i: 2 μm; j: 0.7 μm; k: 6 μm. HCR probes: see Supplementary Data 11.

(Fig. 1d), as well as neurotransmitter signatures (s.b.) at least half of the cell populations on this map have a neuronal identity.

When subdividing our cell populations into different subsets, we chose a granularity that showed major subdivisions, but remained quite stable among subsequent granularity levels (Supplementary Fig. 1d, e), and also was not affected by the adjustment of library sizes (Supplementary Fig. 1f, g). This grouping yielded 22 major cell populations (clusters 0-21) (Supplementary Fig. 1d). We grouped some of these into broadly defined super-clusters, to combine clusters of highly similar identity from reproductive and non-reproductive stages (Fig. 1e; super-clusters a,b, d-f), or because of additional molecular information detailed below (Fig. 1e, super-cluster c). For annotating these groupings, we used unique and shared markers of cell clusters

and super-clusters (Fig. 1f, Supplementary Data 1 and s.b.). We also generated modules of genes showing correlation in their expression (Supplementary Fig. 2a–l and Supplementary Data 2). These modules match individual (or multiple) of the defined clusters (Supplementary Fig. 2a, b, h, l), or super-clusters (Supplementary Fig. 2d–g), or span across more distinct populations (Supplementary Fig. 2c, i, j). Collectively, this indicates the relevance of the defined cell populations as biologically relevant groups harbouring distinct cell types.

To assess how the obtained cell populations distribute in the head and may match known structures or cell types, we next investigated the expression of known markers or some transcripts selected from the clustering (Supplementary Data 1, 2, 6; Fig. 1g–k). Markers for the "epidermal" populations (super-cluster a) contain the previously

characterised *erp4.9* and *fabp* genes (cluster 6) that are expressed in a variety of surface areas of the head[40], whereas *qpeptin*, a marker for the immature/premature state of the "astrocyte-like" clusters (cluster 5), is expressed in the posterior brain and around the eyes[40]. Transcripts of Pkd2.1, a validated mechanoreceptor found in the larval brain[47], demarcate clusters 0 and 17 (mechanoreceptors). Cluster 2 contains a number of markers for sensory and neurosecretory cells, including *l-cry* and *pdf*, which share major expression in a posterior oval-shaped domain (POSD) between the posterior eyes[48,49]. *L-cry* is also among the enriched genes for a newly defined population (cluster 19) that – judged by staining for its marker genes *pedal peptide 2/MLD* and *globin IIα*[50] – localises to a distinct set of cells between the posterior eyes (Fig. 1k). Of these, *pedal peptide 2/MLD* is part of a larval neurosecretory centre[51], in line with the overall neurosecretory nature of cells in the POSD and the associated cluster 19. Detection of transcripts encoding a C1q-domain-containing protein (*c1ql*) enriched in cluster 12 highlights distinct cell bodies and neurites lying between both anterior and posterior eyes (Fig. 1g), whereas a *Platynereis neuroexin/nrxn* homologue enriched in cluster 14 is mostly expressed in the mushroom bodies (Fig. 1h). *Lophotrochin*, one of the genes enriched in the "multiciliated" cluster 15, is weakly expressed in the anterior cirri, known sensory structures (Fig. 1i, j).

When probing the atlas for genes typically associated with distinct neurotransmitter types, these also segregate on the map, with varying selectivity. *Vesicular glutamate transporter (vglut)*, a marker for glutamatergic neurons, generally has a broad pattern, including cells in part of super-cluster b, as well as clusters 2, 4, and 18 (Supplementary Fig. 2n). *Glutamate decarboxylase 1 (dce1)*, a marker for GABAergic cells, exhibits a partially complementary pattern, with strongest expression in super-cluster b (Supplementary Fig. 2o). Judging by the expression of *choline O-acetyltransferase (clat)* and *vesicular acetylcholine transporter (vacht)*, cells in cluster 19 likely use acetylcholine as neurotransmitter, as do cells in cluster 14 (mushroom body neurons), and subsets of cells in clusters 10, 17, 2, and 4 (Supplementary Fig. 2p). Similar cells in clusters 10 and 2 also express *synaptic vesicular amine transporter (vmat)*, a marker expected to be found in mono-aminergic (dopaminergic, (nor)adrenergic, serotonergic, histaminergic) cells (Supplementary Fig. 2q). Consistently, *tyrosine 3-monooxygenase (ty3h)*, relevant for the catecholamines dopamine, norepinephrine and epinephrine, has a similar expression pattern (Supplementary Fig. 2r), while *tryptophane 5-hydroxylase (tph)*, a marker for serotonergic neurons, and *histidine decarboxylase (hdc)*, a marker for histaminergic neurons, demarcate distinct subsets of cells in cluster 2 and, respectively, 4 and 7 (Supplementary Fig. 2s, t). These findings extend a cholinergic signature suggested for larval mushroom bodies[52] to the adult, and further suggest that large clusters like cluster 2 (POSD), despite carrying joint molecular signatures, harbour diverse cells of different neurotransmitter types.

Taken together, all these data indicate that the head atlas represents a set of about 50% neuronal cell populations of broad categories and additional ~20–25% glia-like cells. These populations harbour distinct cell types of known and novel identity that we expect to encode the worm's sensory and information processing, its complex time-keeping abilities, and central output control.

In order to better understand the possible plasticity of these cell populations across adult stages, we probed for cell proliferation in the adult brain by using 5-ethynyl-2'-deoxyuridine (EdU) incorporation, which demarcates S-phase nuclei (Fig. 2a–e and Supplementary Fig. 3a–h). Heads of non-reproductive adult animals exhibited high levels of proliferation (Fig. 2a, b, e, Supplementary Fig. 3a, b, e, f and Supplementary Data 3). By contrast, heads of reproductive worms showed nearly no EdU-positive (EdU⁺) cells (Fig. 2c–e, Supplementary Fig. 3c, d, g, h and Supplementary Data 3).

To assess whether the EdU⁺ cells belonged to neurogenic lineages, we introduced an EdU labelling step into the scRNA-seq protocol and used fluorescence-activated cell sorting (FACS) to isolate and sequence EdU⁺ cells from premature heads. The obtained scRNA-seq transcriptomes of EdU⁺ cells clustered with immature and premature cells in the UMAP representation of the library projected over the UMAP-reduced space of the head/brain atlas (cf. Methods: Predictive dimensional reduction and annotation of new datasets; henceforth called "predicted UMAP projection"), matching their origin (Fig. 2f and Supplementary Fig. 3i). Compared to unlabelled cells, EdU⁺ cells not only showed significantly enriched expression of proliferative markers and cell cycle genes such as *proliferative cell nuclear antigen/pcna*, *cyclin-dependent kinase 1-like/cdk1-like* (Fig. 2g, h and Supplementary Data 4, 5), but also exhibited an enrichment for an established marker of *Platynereis* neurogenesis, *soxB2*[45,53] (originally referred to as *soxB*), as well as a previously uncharacterised ortholog of mammalian *ki-67* broadly used as marker for proliferative neurons in mammalian neuroscience (Fig. 2g, h, Supplementary Fig. 4a, b and Supplementary Data 4, 5). On the sequence level, phylogenetically validated Ki-67 orthologs have, to our knowledge, not yet been reported in other invertebrates. Our annotation is supported by molecular phylogeny and protein domain analyses (Supplementary Fig. 4a, b). The latter revealed both the N-terminal Forkhead associated domain (FHA) and the protein phosphatase 1 (PP1) binding domain diagnostic of vertebrate Ki-67 members (Supplementary Fig. 4b). PP1 is known to be relevant for the function of Ki-67 in mitotic progression[54]. Moreover, the predicted bristleworm Ki-67 protein has a content of serine/threonine residues of 20%, akin to mouse Ki-67, where these residues are targets of cell-cycle-dependent phosphorylation[54]. EdU + cells also express an ortholog to invertebrate *neurofilament/nf* genes. (Supplementary Fig. 4c and Supplementary Data 4, 5). As supported by our phylogeny, these are remote, but joint orthologs of diverse neurofilament genes in mammals, including *gfap* known to mark undifferentiated cells with neurogenic potential in mammals[55].

Taken together, scRNA-seq of the EdU⁺ cells identified several neurogenic markers, pointing at molecular similarities between the tentative aNSCs present in the bristleworm and in the mammalian brain.

## A ciliary marginal zone-like stem cell system in the adult eye generates eye photoreceptors and support cells

After identifying aNSC-like cells in the adult *Platynereis* brain, we next investigated if there were specific neurogenic domains contributing to adult eye and brain growth. Specifically, we investigated the adult eyes of the animals that exhibit an everted, camera-type setup[4]. The eyes show continuous growth, ending with a brief period of rapid augmentation shortly prior to the animals reaching sexual maturity (Fig. 3a–d), when the eyes reach their final morphology (Fig. 3e).

It has been hypothesised that the eyes generally grow by continuous apposition of cells to the rim of the retina, while the final eye enlargement was thought to be primarily caused by expansion of the glass body/lens volume, without further cell number increase[4]. The enlargement of the glass body volume was shown to correlate with a marked elongation of the outer segments of the eye photoreceptors[4]. While we confirmed a marked increase in glass body/lens diameter in reproductive worms (Supplementary Fig. 5a–e), our EdU⁺ single-cell analyses were consistent with the idea that neurogenic cell populations were also present in the eye (Fig. 2f, g).

We therefore focused on super-cluster c of our transcriptomic map (Figs. 1e, f, 3f; eye- and transcriptomically related cells) that is characterised by a cumulative module built from genes previously identified as expressed in *r-opsin1*-positive photoreceptor cells[56], as well as on an overlapping, independently generated gene correlation module (Supplementary Fig. 5f–h; Supplementary Fig. 2g and Supplementary Data 2 (Module 7), Supplementary Data 6). In contrast to the notion that the eye enlargement during maturation was solely caused by the enlargement of existing cells[4], the contribution of cell

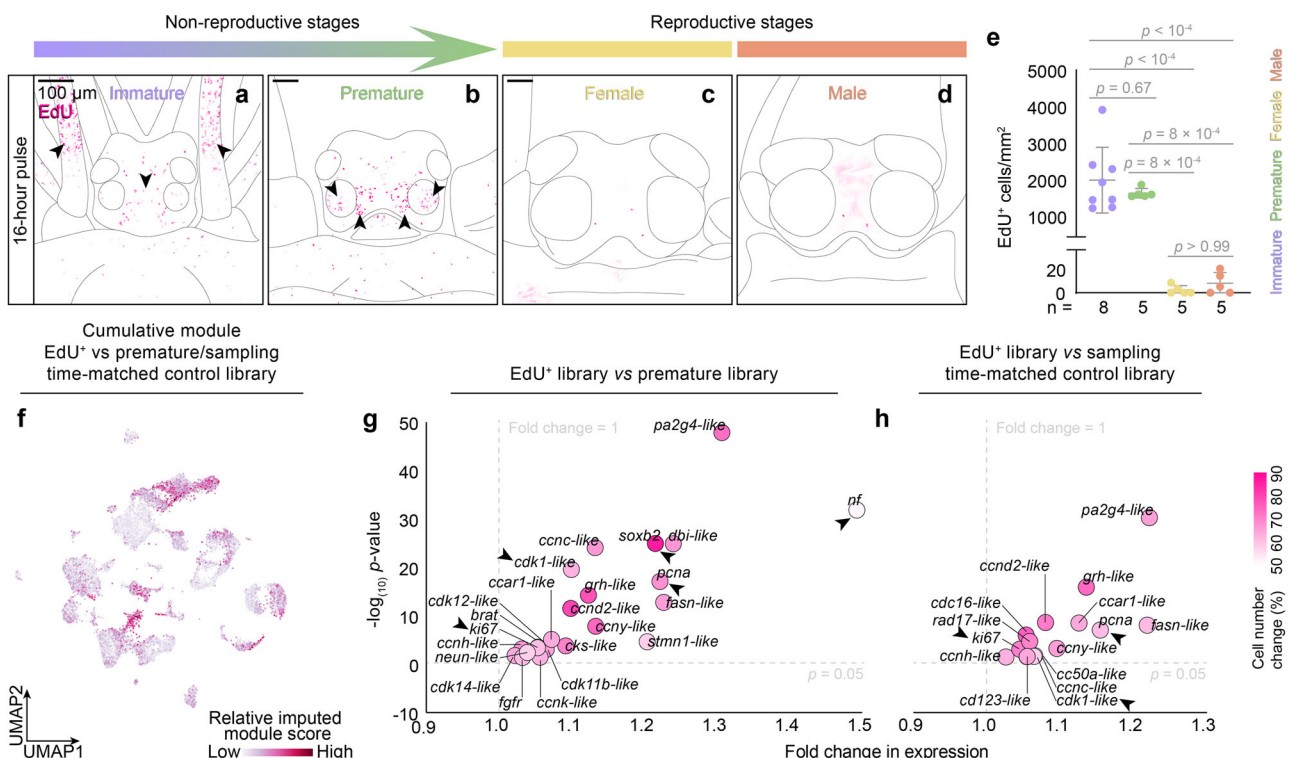

**Fig. 2 | Proliferation and neurogenesis across different cell populations are sharply downregulated in the transition of animals to sexual maturity. a–e** Regulation of proliferation across adult development. Proliferative cells detected in (**a**) immature, (**b**) premature, (**c**) female and (**d**) male worms following a 16 h EdU pulse. Arrowheads indicate major mitotic foci; specimens are 72, 76, 36 and 36 μm thick, respectively. **e** Systematic comparison, normalising the numbers of EdU-positive cells (following a 16 h pulse) to the head area of immature (*n* = 8), premature, female and male worms (*n* = 5, respectively). Mean values ± SD. One-way ANOVA with Tukey's multiple comparisons test. See Supplementary Fig. 3a–h for non-inverted images and controls, and Supplementary Data 3 for source of plotted data. **f** EdU-labelled cells populate diverse cell populations of the head. Visualisation of a cumulative module of genes differentially expressed between sorted EdU-

positive cells and a lunar time-matched control library (see h), showing contributions to diverse cell populations. **g, h** Bona fide neurogenesis- and proliferation-associated genes differentially expressed between whole EdU-labelled cells unlabelled premature libraries; "time-matched" refers to lunar timing. Only genes surpassing the indicated fold change and *p*-value thresholds are displayed. Arrowheads indicate genes mentioned in the main text; statistics by two-sided Wilcoxon Rank Sum test with Bonferroni correction. Candidates were selected on the criterion of playing a role in cell cycle regulation in humans and mice, as well as neurogenesis, neural commitment and proliferation in *Platynereis dumerilii* and *Drosophila melanogaster*. Further details in Supplementary Fig 3, Supplementary Data 4 and methods.

libraries derived from reproductive animals was approximately four-fold higher than those derived from non-reproductive stages (super-cluster c in Supplementary Fig. 5i), with males exhibiting a higher cell count than females, consistent with their final eye-to-head size (Fig. 3c, d, Supplementary Fig. 5j and Supplementary Data 3).

To obtain a better understanding of the possible neurogenic processes contributing to eye growth, we subclustered the cells contributing to super-cluster c into a dataset of 2324 cells (immature: 238, premature: 275, female: 672, male: 1139). When we plotted the expression of the main photoreceptor molecule of the adult eyes, *r-opsin1*[27,56] onto the subclustered dataset (Supplementary Fig. 6a), the cells split into three main subclusters: an *r-opsin1*-negative partition (pA), a strongly *r-opsin1*-positive partition (pB), and a smaller partition positioned between these with weaker and patchier *r-opsin1* expression (pC) (Fig. 3g, h and Supplementary Fig. 6a). Aiding this annotation, we also identified distinct genes enriched in each of the populations and plotted them in both the eye and the brain dataset (Fig. 3f, h and Supplementary Data 7).

Several genes associated with photoreception and -transduction demarcate the *r-opsin1*-positive partition pB, confirming them to be eye photoreceptors (EPs) (Fig. 3f, h; and s.b.). Based on past ultra-structural work, a major cell type of the *Platynereis* eye, besides the EPs are support cells, with roles in formation of the glass body/lens and pigmentation[4,34]. A subset of partition pA showed particular enrichment for *fut10/pofut3* (Supplementary Fig. 6b), encoding the

*Platynereis* ortholog of α−1,3-Fucosyltransferase 10 [57] (Supplementary Fig. 6c). Recent work implicates Fut10 in protein O-fucosylation of elastin microfibril interface (EMI) domains, playing a role in secreted protein quality control, and suggests it to be renamed into POFUT3[58]. In situ HCR analysis showed the strongest expression of *fut10/pofut3* enriched in cells that match eye support cells by frequency and position (Fig. 3i–m and Supplementary Fig. 6d–f). Whereas the substrate of *Platynereis* Fut10/Pofut3 remains unknown, fucosyltransferase activity has also reported in the bovine retinal pigment epithelium[59], in line with a possible role in pigment/support cell function. Outside the eye, *fut10/pofut3* also shows some expression in the posterior oval-shaped domains of the brain (Supplementary Fig. 6d–f). Based on our stainings, we tentatively designate a subset of partition pA as eye support cells (pA1 in Fig. 3g, h), while the remaining section of partition pA (here denominated as "uncharacterised – diverse") likely contributes to photoreceptors and pigment cells known to be present outside the eye, in the worms' brain[26,60,61]. Whereas these are not the only brain photoreceptors (Supplementary Fig. 7a–o), they are molecularly closely related to the eye photoreceptors.

Partition pC is a group of cells interconnecting support cells (partition pA1) and photoreceptor cells (partition pB) (Fig. 3g). Cells in this partition expressed markers associated with neurogenesis like *soxB1, soxB2, prox* and *notch* (Fig. 3g, h, Supplementary Fig. 8 and Supplementary Data 5). Bioinformatic pseudotime analysis by Monocle3[62] is consistent with the notion that these cells are the origin

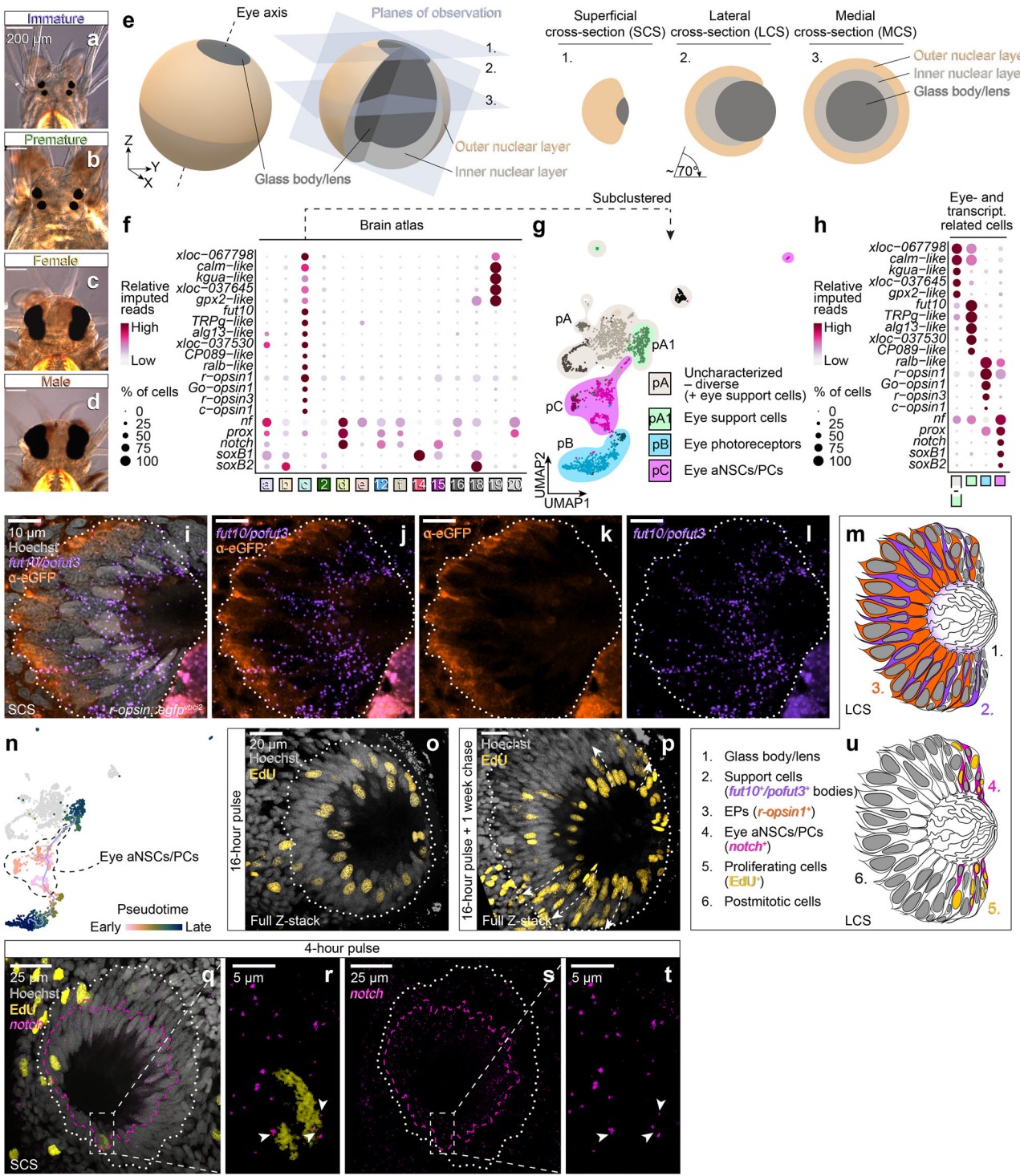

of a differentiation trajectory leading to *r-opsin1*⁺ eye photoreceptors (partition pB) and *fut10/pofut3*⁺ eye support cells (partition pA1) (Fig. 3n and Supplementary Fig. 8g). Furthermore, both the cumulative module computed from transcripts enriched in the EdU⁺ cells in the eye- and transcriptomically related cell dataset, as well as a predicted UMAP projection of EdU⁺ cell transcriptomes in the dataset populated the trajectories of this partition towards partitions pB and pA1 (Supplementary Fig. 8h, i). This is consistent with the four day-long EdU pulse that the animals were exposed to prior to sequencing, hence the majority of the EdU⁺ cells likely belong to a population of actively dividing cells on their path towards differentiation. We thus reasoned that partition pC contained stem cells supplying the eye

photoreceptor and eye support cell populations, and tentatively refer to them as bioinformatically defined eye adult neural stem/progenitor cells (eye aNSCs/PCs) (Fig. 3g, n).

We next wondered whether the tentative aNSCs/PCs reside in a confined area of the eye or are spread throughout the eye cup. For this, we performed EdU uptake assays with and without a chase period. In the pulse-only experiment, the majority of EdU⁺ cells localised to a central ring of cells surrounding the glass body/lens at the level of the eye opening, with other cells located more peripherally (Fig. 3o, see Supplementary Movies 1–3 for 3D renderings of confocal stacks that demonstrate the localisation of EdU⁺ cells at the edge of the cup-shaped retina). Following an additional one-week chase period, the

**Fig. 3 | The adult eye grows via a ciliary marginal zone (CMZ)-like region, giving rise to support- and photoreceptor cell lineages. a-d** Eye size comparison in differential interference contrast micrographs of (**a**) immature, (**b**) premature, (**c**) female and (**d**) male worm of 15, 15, 49 and 63 μm thickness, respectively. **e** Schemes of the adult eye and orientation of the different sections used for microscopic analyses and schemes (**f**) Markers of eye- and transcriptomically related cells in the head/brain atlas. **g** Subclustered eye- and transcriptomically related cell dataset, grouped into three major partitions (bubbles) and standard clusters (cells coloured in different shades within a bubble); eye support cells cluster together with the rest of the population in partition pA. **h** Dot plot for genes shown in (**f**) for the partitions shown in (**g**). **i-l** HCR staining of *fut10/pofut3* and immunofluorescent labelling of eGFP in an *r-opsin1::egfp^{vbci2}* worm. Thickness: 1.15 μm; signal in lower right corner below eye is autofluorescence. **m** Proposed scheme for the arrangement of support cells and eye photoreceptors as a lateral cross-section along the eye axis. **n** Putative differentiation trajectory of cells corresponding to the eye lineages ordered in pseudotime according to inferred differentiation states; cells in grey are suspected brain cells with a transcriptomically similar signature. **o, p** EdU detection following an (**o**) 16 h pulse and (**P**) additional 1-week chase period, respectively. Arrow indicates suggested direction of lineage progression; thickness: 29 and 13 μm, respectively. **q-t** HCR visualisation of *notch* (magenta) along with EdU (4 h pulse, yellow); thickness: 2 μm. **r, t** are enlarged views of the area indicated in (**q**) and (**s**), respectively; arrowheads indicate expression regions of *notch* in EdU-positive cells. Dashed magenta lines demarcate rims of the most prominent HCR staining. **u** Proposed scheme for the localisation of aNSCs in a concentric growth zone at the external surface, abutting the glass body close to the eye opening; orientation as in (**m**). Further details on the denominations of cell clusters to eye, and details on eye growth are provided in Supplementary Figs. 5, 6, and 8.

number of EdU⁺ cells in the innermost ring increased, and we observed trails of EdU⁺ cells arching outwards (arrow in Fig. 3p), reminiscent of the spreading of fish retinal clones along arched continuous stripes[63,64]. Finally, we employed a combined approach of in situ HCR and EdU labelling to probe for a link between the proliferative zone in the eye and the in silico-characterised eye aNSCs/PCs. *Notch*, a marker co-expressed with the neural stem cell marker *soxB1* in partition pC of the eye cluster (Fig. 3h and Supplementary Fig. 8c, d), was expressed in cells of the central region that also exhibited EdU staining (Fig. 3q-t). In summary, these data support the presence of a stem cell zone at the edge of the cup-shaped retina abutting the glass body/lens, giving rise to new eye photoreceptors and eye support cells (Fig. 3u).

## Both light and the ciliary opsin *c-opsin1* are required for proper development of *r-opsin1*-positive photoreceptors

The transcriptomic characterisation of eye photoreceptor cells not only confirmed the presence of the r-opsin genes *r-opsin1* and *r-opsin3* as well as the G₀-opsin gene *G₀-opsin1* in the adult eyes, but also indicated expression of the c-opsin gene *c-opsin1* (Fig. 3f, h). Whereas *r-opsin1*, *r-opsin3* and *G₀-opsin1* had been previously described in the eyes[56,60,65-67], *c-opsin1* was so far only known for its expression in the larval brain, where it exclusively demarcates extraocular non-visual photoreceptor cells[26]. Visualisation of both *r-opsin1* and *c-opsin1* in the subclustered eye-and transcriptomically related cell transcriptome map indicated that *c-opsin1* was co-expressed with *r-opsin1* in a distinct subpopulation of cells that reside on the transition between the tentative aNSCs/PCs and the start of the photoreceptor differentiation trajectory (Fig. 4a-d).

To probe for the validity of these data, we performed in situ HCR analyses on the heads and eyes of non-reproductive worms. Detection of *c-opsin1* (Fig. 4e-g) revealed several distinct cells between the posterior eyes (hatched circles in Fig. 4f), consistent with the aforementioned larval expression data[26]. However, the most prominent *c-opsin1* expression was indeed detected in cells within both pairs of adult eyes (arrowheads Fig. 4e). Furthermore, co-detection of *c-opsin1* and *r-opsin1* mRNA experimentally confirmed that cells co-expressing both of these distinct *opsin* genes exist (Fig. 4h-k). The position of such cells adjacent to the glass body/lens at the level of the eye opening, where the aforementioned results revealed a tentative CMZ-like stem cell region, further supports that they could be part of the predicted aNSCs/PCs on their differentiation trajectory towards eye photoreceptors. If true, at least some of the *c-opsin1*-positive cells should also be proliferative. Indeed, detection of *c-opsin1* RNA in individuals labelled by a short EdU pulse revealed that a subset of the EdU⁺ cells close to the opening of the eye expressed *c-opsin1* (Fig. 4l-s), consistent with the idea that *c-opsin1*⁺ cells were not terminally differentiated, but demarcate a transitory state that gives rise to the *r-opsin1*⁺ photoreceptor population of the adult eye.

Taken together, these data further supported the notion of a CMZ-like stem cell region in the animals' camera-type eye (Fig. 4s), but

also pointed towards the unexpected presence of a ciliary opsin gene in an early step of the differentiation trajectory. This prompted us to investigate the requirement of *c-opsin1* and light for the growth of the eye.

In a first experiment (Fig. 5a, b), we transferred premature individuals of both wild-type (WT) and *c-ops1^{Δ8/Δ8}* genotype to two chambers of distinct controlled light regimes: (i) a 16 h:8 h light-dark cycle ("LD regime") in which naturalistic light was provided that also covered the UV-A spectrum for which c-Opsin1 has been shown to be particularly sensitive[28,29]; and (ii) complete darkness ("DD regime"). After 14 days, an EdU pulse was provided for 24 h (a time span chosen to be independent of any changes in circadian phase), and animals were sacrificed thereafter to investigate the density of EdU-labelled nuclei in the eyes (Fig. 5a). In these experiments, WT specimens under the LD regime exhibited a marked increase of EdU-positive cells per eye area when compared to specimens kept in the DD regime, whereas in *c-ops1^{Δ8/Δ8}* individuals, levels were indistinguishable between LD and DD regimes (Fig. 5b). This indicates that light impacts on proper progenitor proliferation in the eye, and that – indirectly or directly – *c-opsin1* is relevant for this effect to occur.

In order to assess possible changes in the cell populations of the eyes that we had defined, we next performed an scRNA-seq experiment on premature *c-ops1^{Δ8/Δ8}* individuals and their WT counterparts. To facilitate comparisons, we predicted UMAP projections of the resulting *c-ops1^{Δ8/Δ8}* and WT transcriptomes in the UMAP-reduced space of the head/brain atlas and the eye- and transcriptomically related cell subset (Supplementary Fig. 8j, k). Contribution of cells from libraries generated from *c-ops1^{Δ8/Δ8}* specimens *versus* those generated from WT controls to the different clusters differed significantly in five cell populations (Fig. 5c). Among those is super-cluster c (eye- and transcriptomically related cells), which exhibited a notable reduction in the number of cells in partition pB (eye photoreceptor/EP population) (Fig. 5c, d).

When assessing the respective UMAP cell coordinates of transcriptomes from both genotypes, eye photoreceptor cells from the *c-ops1^{Δ8/Δ8}* individuals lacked a group of cells at the end of the cluster in both the standard and predicted UMAP-reduced spaces (Fig.5e-i and Supplementary Fig.9). Taken together, these data are consistent with a function of *c-opsin1* in modulating the production and/or differentiation of *r-opsin1*⁺ eye photoreceptors.

## Reproductive maturation imposes a special cell state combining neurogenic signatures with markers of mammalian aNSC quiescence

After characterising this spatially defined region of neurogenesis, we finally turned our attention back to the question how proliferation was controlled on a temporal scale. Given that we found EdU incorporation to be dramatically reduced in heads of mature animals, as compared to immature and premature specimens (Fig. 2a-e), we aimed to determine molecular changes occurring during this transition.

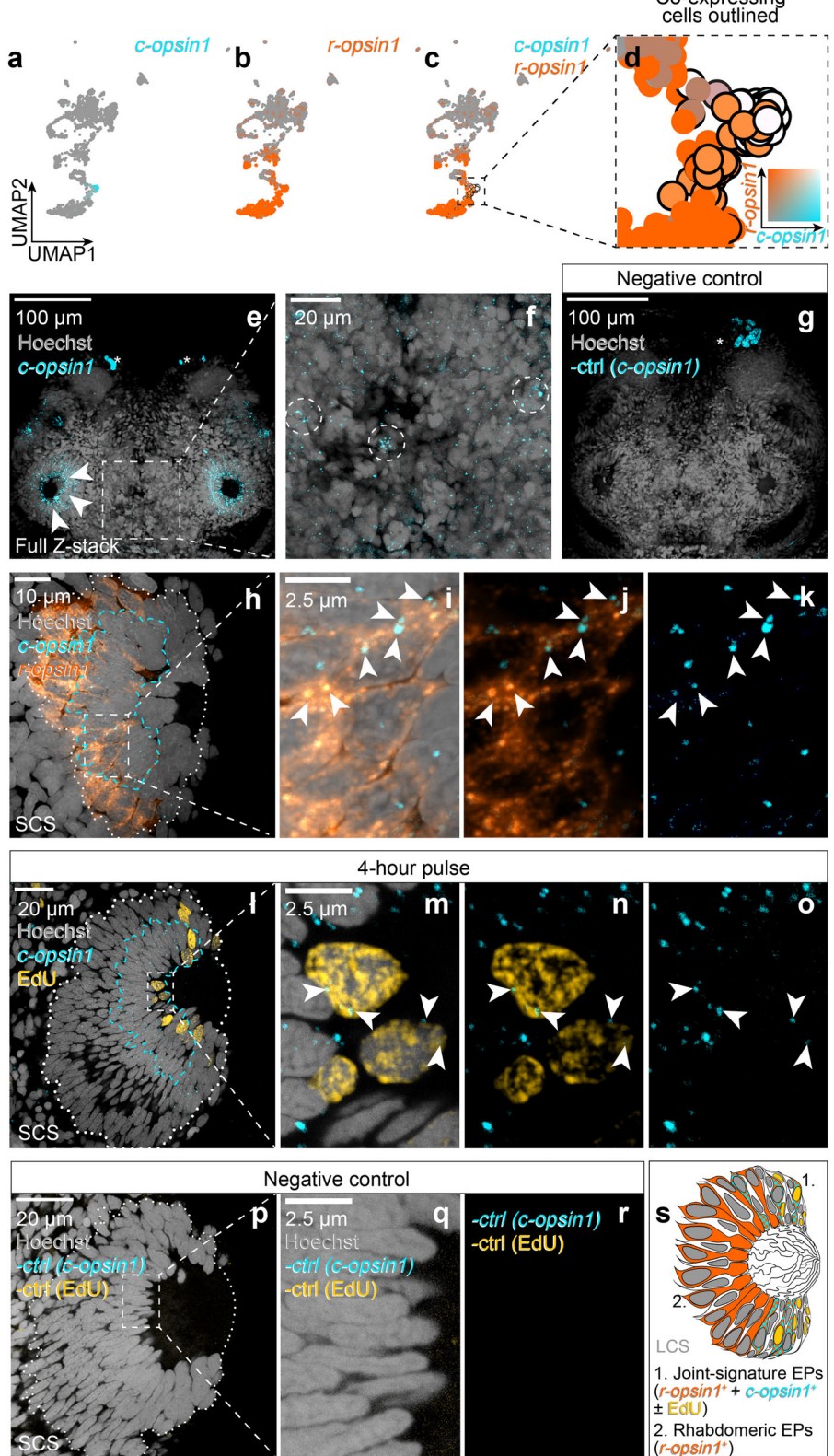

**Fig. 4 | *C-opsin1* is expressed in dividing cells in the CMZ-like region of the adult eye. a–d** Co-detection of *c-opsin1* and *r-opsin1* transcripts in a UMAP projection of presumptive eye cells. Inset in panel (**d**) outlines the colour code used. **e–g** HCR detection of *c-opsin1* in eyes (**e**, arrowheads) and medial posterior brain (**f**, circles) as compared to a negative control (**g**); Dorsal head views, anterior to the top; image thickness: 57, 57 and 48 µm, respectively. Asterisks: autofluorescent glands. **h–k** HCR co-detection (arrowheads) of *c-opsin1* and *r-opsin1* in cells of the eye; thickness: 1.2 µm. **l–r** HCR labelling of *c-opsin1* and EdU incorporation in the eye following a 4 h pulse (**l–o**) and the corresponding negative control (**p–r**). Thickness: 1.2 and 2.6 µm, respectively; arrowheads: double-positive cells. Dashed cyan lines demarcate the region with the most prominent HCR signal. **s** Schematic representation of the observed cells in a lateral cross-section along the eye axis.

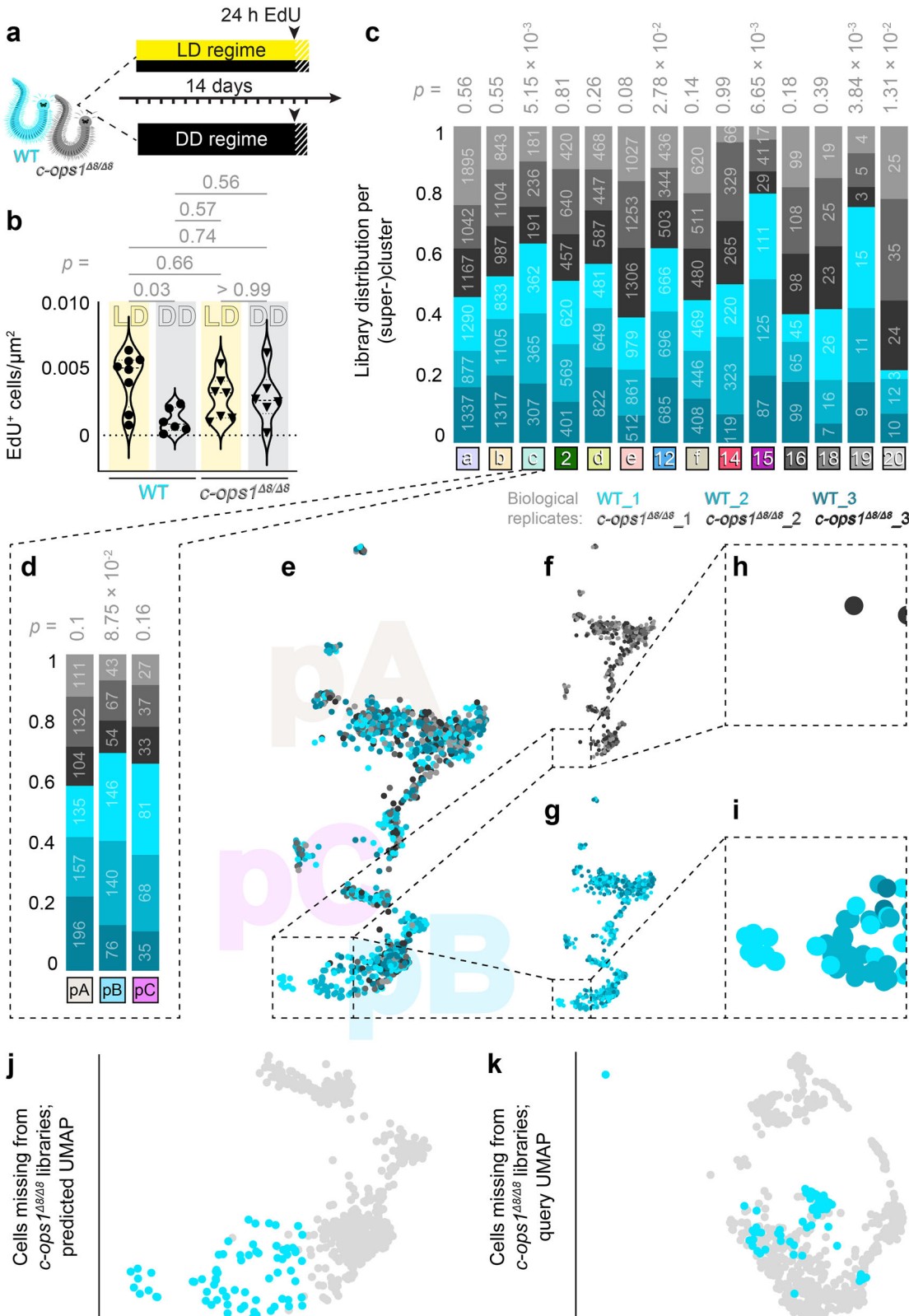

**Fig. 5 | *C-opsin1* mediates an impact of light on eye proliferation and rhabdomeric photoreceptor differentiation. a, b** Experimental setup (**a**) and results (**b**) of an EdU labelling experiment on wild-type animals (WT) and *c-ops1* mutant animals (*c-ops1^{Δ8/Δ8}*) raised in a light:dark (LD) and darkness (DD) regime, respectively. Respectively, differences in (**b**) evaluated by one-Way ANOVA followed by Tukey's multiple comparison test. **c, d** Representation of WT and *c-ops1^{Δ8/Δ8}* scRNA-seq libraries (*n* = 3 for each genotype) in cell populations of (**c**) the head atlas and (**d**) the eye- and transcriptomically related cell dataset; statistics: unpaired two-sided *t*

test; numbers inside the bars represent individual cell numbers. **e–i** Predicted UMAP-reduction of WT and *c-ops1^{Δ8/Δ8}* libraries, generated by projecting WT and *c-ops1^{Δ8/Δ8}* libraries (query) over the eye- and transcriptomically related cell dataset (reference). A subset of the *c-ops1^{Δ8/Δ8}* and WT dataset corresponding to EPs and matching eye aNSCs/PCs visualised as the (**j**) predicted and (**k**) standard UMAP of the query dataset. Highlighted cells are absent from *c-ops1^{Δ8/Δ8}* libraries. Further details in Supplementary Figs. 8 and 9.

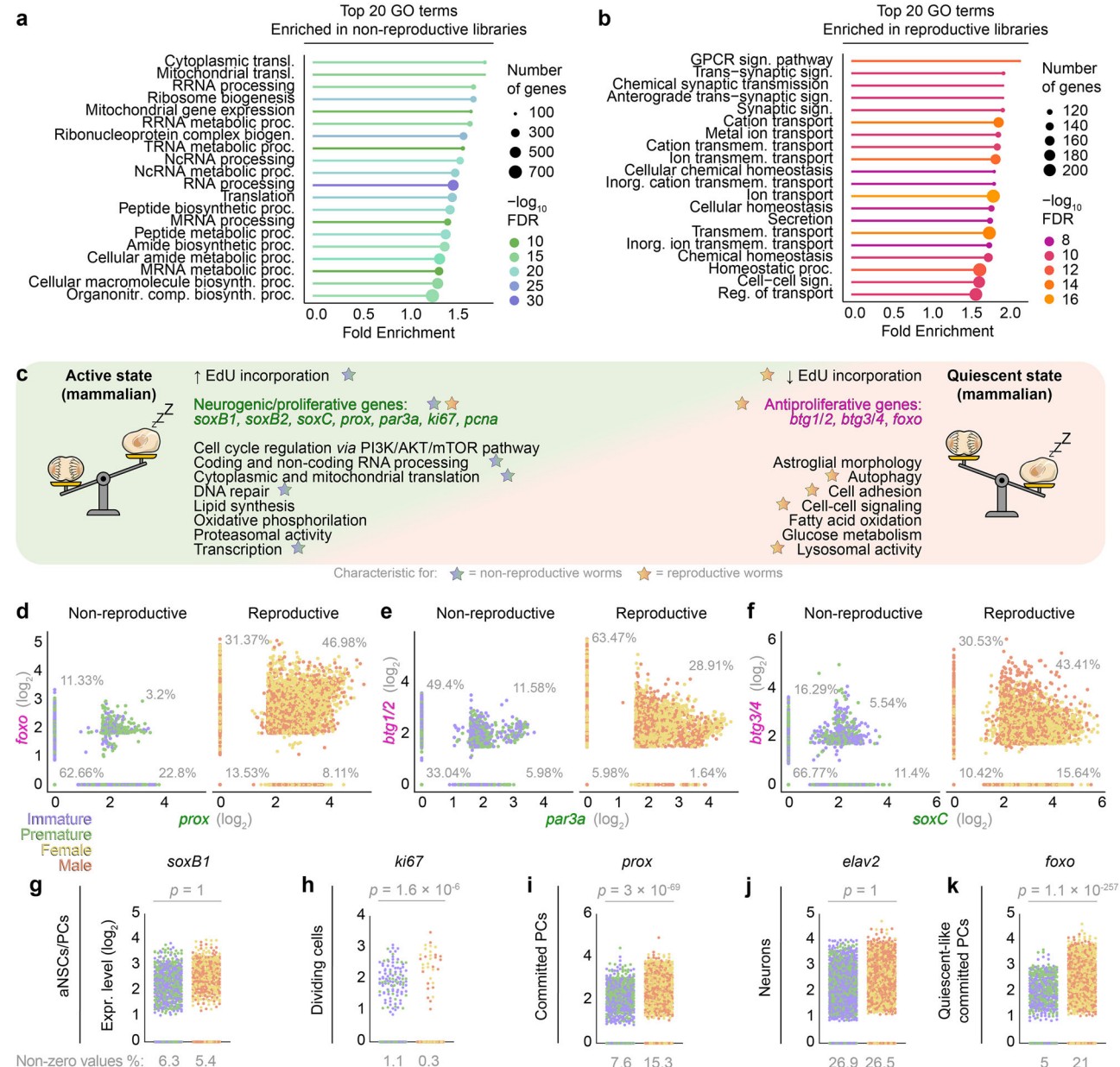

**Fig. 6 | Vertebrate-type neurogenic and quiescent-like signatures are co-regulated in reproductive brains. a**, **b** Top 20 GO terms associated with genes enriched in (**a**) non-reproductive and (**b**) reproductive libraries; FDR = false discovery rate. **c** Hallmarks of active and quiescent aNSCs in mammalian systems; stars indicate physiological and transcriptomic hallmarks detected in non-reproductive (blue stars) and reproductive libraries (orange stars) in the GO term analysis of the genes differentially expressed between the libraries. **d**–**f** Relative imputed values of co-expression of neurogenic (green) and quiescence-associated genes (magenta) in non-reproductive and reproductive libraries; percentages indicate fraction of cells expressing one, both or none of the markers. **g**–**k** Scatterplots showing non-imputed expression of representative genes characteristic for distinct stages of the neurogenic cascade in non-reproductive and reproductive libraries. Statistical significance calculated using a two-sided Wilcoxon Rank Sum test. Further details in Supplementary Figs. 10 and 11.

In order to test for molecular signatures distinguishing cells before and after maturation, we first performed a pseudo-bulk differential expression analysis between all cells from non-reproductive and reproductive stages (Supplementary Data 8). We then conducted gene ontology (GO) analyses on the sets of differentially regulated genes to determine the most distinct biological processes for each set. In non-reproductive, proliferative worms, the top enriched GO terms were related to translation, ribosomal processes, as well as coding and non-coding RNA processing (Fig. 6a and Supplementary Data 9). In reproductive, non-proliferative worms, top-enriched terms included intracellular signalling and synaptic and vesicular processes (Fig. 6b and Supplementary Data 10). These, as well as other enriched terms from non-reproductive (blue stars in Fig. 6c) and reproductive (orange stars in Fig. 6c) gene sets, resembled signatures associated with active and quiescent mammalian aNSCs, respectively[16,68,69] (Fig. 6c). In contrast, no GO terms associated with senescence or cell death were enriched in the cells from reproductive (or non-reproductive) animal heads, indicating that the non-proliferative state in the reproductive worms was likely not the result of an irreversible cell cycle arrest, apoptosis or necrosis (Supplementary Data 9,10; also cf. ref. 70). While the association of mammalian quiescence hallmarks with genes enriched in cells from reproductive animals was generally valid, we noted a few exceptions for terms related to proliferation and neurogenesis, which were significantly overrepresented in supposedly non-proliferative

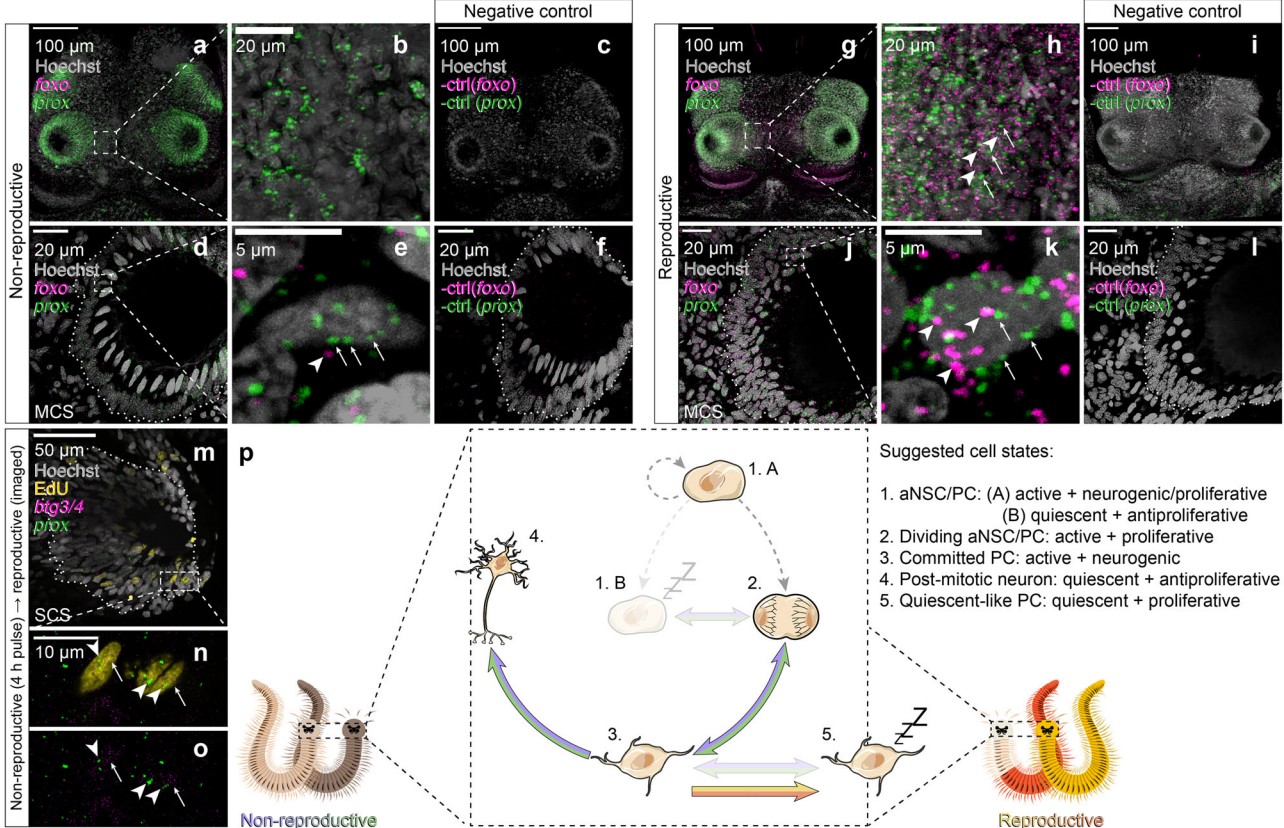

**Fig. 7 | Cellular co-expression of neurogenic and quiescent signatures upon maturation. a–l** HCR co-labelling of *foxo* and *prox* and corresponding negative controls in (**a–c**) a premature head (55, 55and 66 μm thickness, respectively), (**d–f**) a premature eye (each 2 μm thickness), (**g–i**) a mature male head (95 μm thickness in all images) and (**j–l**) a mature male eye (2 μm thickness in all images). **m–o** Co-

labelling of *prox* and *btg3/4* in eye cells of mature animals labelled prior to maturity by a 16 h EdU pulse-chase experiment (1 μm thickness) (**p**) Schematic representation of the proposed progression of the neurogenic cascade in non-reproductive and reproductive animals. Further details in Supplementary Figs. 10 and 11.

reproductive animals (Supplementary Data 10). Specifically, we observed that transcripts contributing to these terms were genes associated with neuronal proliferation in progenitor cells, like *prospero/prox*, *soxC* (ortholog of vertebrate *sox4*, *sox11* and *sox12*) and *par3a* (ortholog of vertebrate *pard3*)[53,71]. While other genes associated with active aNSCs/PCs remained expressed in the reproductive stage, *prox*, *soxC* and *par3a* were even upregulated in reproductive animals (Fig. 6d–i, Supplementary Fig. 10a–f and Supplementary Data 5).

However, consistent with the absence of cell divisions, we also found enrichment of genes associated with the quiescent stem cell state in the same gene set, such as orthologs of the mammalian genes *B cell translocation gene 1/btg1* and *btg3*, as well as *foxo3* and *foxo6*, which we identified as *Platynereis btg1/2*, *btg3/4* and *foxo*, respectively (Fig. 6d–f and Supplementary Fig. 11). We also detected unchanged expression levels of post-mitotic markers *synaptotagmin/syt* and *embryonic lethal/abnormal vision* genes *elav1* and *elav2*, indicating that the switch in neurogenic capacity does not influence the ratio of resulting neurons (Fig. 6j, k and Supplementary Fig. 10g–j). FoxO proteins are relevant for regulating stem cell quiescence in diverse stem cells[72,73]. Btg1 and 2 have been shown to stabilise mammalian stem cell quiescence by reducing polyadenylation of mRNAs[74], while Elav and Syt are involved in neuronal differentiation[75] and neuronal function[76].

The simultaneous co-enrichment of neurogenic and antiproliferative genes in cell signatures of reproductive animals suggests that the cessation of cell divisions is linked to the induction of a quiescence-like molecular programme in the formerly proliferative

cells of the brain and eye of the reproductively maturing animals. Bioinformatic approaches using cumulative gene modules computed for bona fide markers of distinct phases of neurogenesis as well as quiescence-associated genes confirmed a high overlap in cells expressing neurogenic and quiescence-associated markers in reproductive animals (Supplementary Fig. 10b, d, f, h, j).

To experimentally validate this overlap, we performed HCR stainings for representative markers on non-reproductive and reproductive animals[77]. Animals of both stages displayed expression of *prox*, concentrating in the eyes and posterior oval-shaped domains of the brain (Fig. 7a, f). By contrast, and in agreement with the in silico data, the expression of *foxo* was markedly enriched in reproductive worms, coinciding with *prox* in the same cells (Fig. 7g–l). Likewise, reproductive worms also show co-expression of *prox* with *btg3/4* (Fig. 7m–o). This signature also comprises cells previously labelled with EdU in a pulse-chase experiment in which animals were exposed to a 16 h EdU pulse just prior to maturation and then left to mature (Fig. 7m–o). As maturation stages do not incorporate EdU anymore (Fig. 2c, d), labelled cells in this experiment are supposedly two weeks old. Co-detection of *prox* with *btg3/4* in these cells is consistent with the notion that the quiescence-like signature in reproductive animals is also imposed on cells that have only recently differentiated or are still in a molecular state of differentiation.

Taken together, our data suggest that reproductive maturation imposes a quiescence-type signature on formerly proliferative cells, coinciding with a sharp downregulation of proliferation at this stage, despite the simultaneous presence of a molecular signature associated with neurogenesis (Figs. 6c, 7p).

## Discussion

Comparative model systems are valuable sources for neurobiological discovery and mechanistic principles[78]. *Platynereis dumerilii* has served as a reference model for comparing both early nervous system patterning and cell types across bilaterians[26,79–82]. So far, most emphasis has been on larval development, where the trochophore larva facilitates comparisons across different phyla[83], with a focus on functional differentiated cell types as units of comparisons[6,80]. By investigating adult stages, our work allows us to explore signatures of adult neural stem cells and their regulation in the head – processes that are neither accessible in larvae, nor well understood in their evolutionary history. As outlined, our work reveals modulation of neurogenesis on two levels: first, the growth of the adult camera-type eye via a circular stem cell zone, reminiscent of the vertebrate ciliary marginal zone. Proper proliferation and/or development of cells in this zone depends on light as well as the integrity of the ciliary opsin gene *c-opsin1*. Secondly, we find evidence for a global change in molecular signatures correlating with the entrance of animals into the reproductive stage towards the end of their adult life that is linked to a pronounced stop of proliferation across the entire head. Whereas we cannot independently validate the functional role of the involved genes, gene ontology classification suggests similarities to the signature of mammalian quiescent aNSCs.

The tentative bristleworm aNSC system exhibits both similarities and differences to mammalian aNSCs. Molecularly, the *ki-67* ortholog we find in the worm, and its association with tentative aNSCs, supports the notion that there is an ancient molecular signature of proliferative brain cells that is shared between bristleworms and vertebrates[53]. In contrast to mammalian aNSCs, however, where quiescence is best studied as a mechanism regulating the activation/inactivation of an early neuronal stem cell pool, we find the quiescence-like signature in bristleworms to be mostly associated with progenitor cells that stop proliferating during reproductive maturation. We suppose that this association relates to a more general, anti-proliferative function of the signature in worms. While at least some vertebrate orthologs like Btg2 also have an anti-proliferative role in progenitor cells undergoing terminal neuronal differentiation[84], it remains to be tested if the observed similarities relate to ancient functions or are evolutionarily divergent.

Our study has not attempted to address if or not the quiescence-like state of tentative worm aNSCs/ PCs, including its proliferative stop, is reversible. Like other nereidid annelids, *Platynereis* exhibits semelparity, with mature animals dying after reproduction. While this extreme mode of reproduction may have facilitated the discovery of corresponding molecular changes in our study, it also entails that post-reproductive individuals cannot be tested for reactivation of neurogenesis. Semelparity likely represents a secondary evolutionary modification, as related worms have been demonstrated to possess a latent capacity for iteroparity, *i.e.* multiple reproductive cycles within lifetime[85]. Moreover, representatives of other annelid groups, such as the Palolo worm *Eunice viridis* only transform posterior segments during reproduction, and thus exhibit a natural cycle of reproductive and regenerative states[86]. It will be interesting to test if in this setting, neurogenesis and gametogenesis exhibit mutually excluding phases of quiescence, consistent with the idea that in organisms, quiescence is a mechanism to control energy consumption by growth[87].

Cephalopod camera-type eyes have served as textbook examples of convergent morphological evolution. Like cephalopods, *Platynereis* worms exhibit everted camera-type eyes. Our molecular characterisation of cell populations in the growing bristleworm eye exhibits some parallels to gene expression data recently described for the developing squid retina[22]. For instance, our molecular data imply that *soxB1* demarcates early progenitor cells in both systems, as does *notch*, which has been shown to be functionally required for cephalopod retina development[22]. Also, the *soxB2* gene is expressed in the progenitor zone of both systems, even though our data suggest this to be more related to the support cell lineage than the eye photoreceptor lineage. Whereas a spatially defined growth zone has not been reported for cephalopod eyes[22,25], the spatial gene expression data we report here suggest the existence of a dedicated stem cell zone in the bristleworm eye. By the position of these cells in a ring around the eye opening close to the glass body/lens, we suggest this zone to be a functional correlate of the vertebrate ciliary marginal zone, extending a functional building principle in convergently evolved camera-type eyes to the stem cell zone. Our work complements prior studies that have attributed the massive enlargement of eyes in maturing worms exclusively to the significant elongation of the outer segments of photoreceptors[4]. During earlier stages, the addition of new photoreceptors occurs at very slow rates[34]. Whereas it is reasonable to assume that the growth patterns we observe also hold for these earlier stages, this idea currently remains untested.

Our observation that the ciliary opsin *c-opsin1* is expressed at an early stage of *r-opsin1*-positive photoreceptor differentiation contributes in an unexpected way to our view of photoreceptor evolution. Whereas the evolutionary divergence between distinct *r-opsin* and *c-opsin* genes dates back to pre-bilaterian times[26], these receptors are closest neighbours in deep phylogenetic analyses[88], suggesting that they once duplicated from a joint precursor. This scenario implies that, before the emergence of distinct photoreceptor subtypes expressing any single of these *opsin* genes, they would both have been co-expressed in the same cell type, allowing for functional diversification. The situation we observe in today's bristleworm eyes might thus either indicate a more recent co-option of *c-opsin1* into the developmental trajectory of the *r-opsin*⁺ eye photoreceptor lineage, or possibly reflect an ancestral constellation.

Our findings that light impacts on adult eye growth, that *c-opsin1* is required for this effect, and that *c-opsin1* mutants exhibit altered photoreceptor signatures, are all consistent with a role of *c-opsin1* in modulating *r-opsin*⁺ eye photoreceptor proliferation and/or differentiation. At present, we favour a model in which there is a progenitor cell population that transiently expresses *c-opsin1* and – in a light-dependent manner – asymmetrically divides to give rise to a new progenitor cell and a cell that will become a differentiated rhabdomeric eye photoreceptor cell (partition pB).

These results tie in with a diverse range of roles that photoreceptors play in developmental and adult stages of *Platynereis*[27,48,56,60,89–91]. As to light-modulated growth, recent results also imply the light-receptive cryptochrome gene *l-cry* in regulating overall growth and life span of *Platynereis* in a light-dependent manner[92]. As *l-cry* is also present in eye cells and cells of the posterior oval-shaped domain, likely partly overlapping with *c-opsin1* (ref. 48, Supplementary Fig. 7g,l), future work will need to address how these two non-visual light-receptors (and possibly others) tune and time the worms' coordinated growth, physiology and behaviour.

Whereas the co-expression of *c-opsin1* in the *r-opsin1* photoreceptor lineage is compatible with cell-autonomous modulation, the current unavailability of tissue-specific gene inactivation or rescue techniques precludes analyses that could directly test this notion. Moreover, we note the effects of *c-opsin1* ablation on cell populations (clusters 12, 15, 19) outside the eye. As cells in clusters 12 and 19 are localised in the vicinity of the medial brain-residing *c-opsin1*-positive cells, this could either reflect local functions of these cells or indirect effects by either of *c-opsin1*-positive populations, possibly via broader physiological changes[29]. In either scenario, the regulation might help to optimise the worms' sensory system to the different light conditions they encounter. Light in the marine environment provides organisms with a wealth of information[90]. Adjusting growth processes of sensory systems might hence provide organisms with a level of plasticity that is a selective advantage in constantly changing surroundings. Future work will need to determine if such plasticity occurs in other animal

groups, and how anthropogenic change, such as artificial light, might impact on it.

## Methods

### Worm culture and light conditions

*Platynereis dumerilii* worm culture was maintained at the Marine Facility of the University of Vienna, Austria, under a 16:8 light:dark (LD) regime with a cyclical nocturnal illumination simulating a full moon for 8 nights per one lunar cycle[48]. For experiments involving *c-ops1*[Δ8/Δ8] animals and corresponding wild-type (WT) individuals, the animals were kept under full-spectrum sunlight conditions, as *c-opsin1* encodes a UV-sensitive photoreceptor[29,93]; otherwise, the animals were reared in artificial sunlight conditions. All worms received artificial moonlight. Artificial sunlight and moonlight spectra:[48]. Animals were raised in rectangular plastic containers filled with 1-1.5 L of a 1:1 mix of sterile-filtered, salinity-adjusted natural and artificial sea water (Tropic Marin Classic). Batches were fed once a week with minced organic spinach leaves and once a week with a blend of spirulina powder and ground Tetramin fish food.

### Worm staging and sampling

Experiments on animals were conducted following the applicable Austrian and European guidelines for animal research. Wild-type animals were sampled from the PIN strain that has been derived in Vienna by systematic inbreeding efforts and was used as a genetic background for inducing and maintaining *c-ops1*[Δ8/Δ8] mutants[29] and *r-opsin::egfp*[vbci2] transgenic[66] worms. Animals were sampled between Zeitgeber Time (ZT) 2 and 3, a week following the full moon stimulus, unless otherwise indicated. Immature worms were chosen based on age (under 3 months) and size (30–40 segments). Premature worms were distinguished from immature worms by size (over 60 segments) and the presence of developing gametes, microscopically detectable upon clipping the worm's body between two posterior parapodia. Premature animals with an evident onset of metamorphosis (eye enlargement, emptied gut, colour change from brown/grey to light orange, morphological changes in the first parapodia) were omitted from the study. Mature animals were selected for enlarged eyes, sexually dimorphic coloration (females: bright yellow; males: whitish anterior and bright red posterior) and crawling behaviour. Fully mature animals exhibiting swimming behaviour were omitted from the study due to the risk of spontaneous reproduction during handling. For scRNA-seq experiments involving EdU-treated worms, prior to sampling, premature animals were exposed to a 96 h pulse of EdU, as described in "EdU labelling"; for these experiments, the treated and the control group ("sampling time-matched premature library") were sampled at the time of new moon, i.e. two weeks following the full moon stimulus. All worm heads were sampled by anesthetising the animals in 7.5% (w/v) MgCl$_2$ diluted 1:1 in worm culture water, followed by a surgical amputation of the head. For scRNA-seq experiments, 20 immature, 10 premature, 8 female and 8 male worm heads (transcriptomic atlas), 120 EdU-treated premature heads (EdU experiment) and 10 premature WT and *c-ops1*[Δ8/Δ8] worms were collected per replicate (*c-opsin1* experiment) or sample (all other experiments). The heads were amputated anteriorly to the first segment and posteriorly to the nuchal organ commissure. For imaging experiments, heads were amputated posterior to the jaws. Unless otherwise indicated, all imaging experiments were conducted on premature worms as representative of the non-reproductive stage, and mature male worms as representative of the reproductive stage.

### Generation of single-cell suspensions

Following the amputation of the worm heads, the tissue was simultaneously dissociated, permeabilised and fixed following the acetic acid-methanol (ACME) protocol[94] with the following modifications. Heads were incubated for 1 h in the ACME solution prepared according to the

protocol, and tissue dissociation was mechanically aided by resuspending the solution for 2 min every 15 min. The suspension was filtered through a 40 μm cell strainer (Flowmi™ Cell Strainers; Scienceware) and centrifuged at 2000 × g, and the supernatant was discarded. Where applicable, the EdU Click-iT™ reaction was performed on EdU-treated samples for 20 min, according to the manufacturer's protocol for the Click-iT™ EdU Cell Proliferation Kit for Imaging (Thermo Fisher Scientific); the sample was then centrifuged and the supernatant discarded. The pellets of all samples were suspended in the freezing solution prepared according to the protocol, using 0.1% instead of 1% BSA, and stored at −20 °C. On the day of library preparation, the samples were thawed, stained for 7 min with 1 μg/mL nuclear dye (Hoechst 33342, trihydrochloride, trihydrate; Thermo Fisher Scientific) and centrifuged, the supernatant discarded, the pellet suspended in resuspension solution and transferred into tubes for fluorescence-activated cell sorting (FACS). To prevent RNA degradation, all solutions and reaction buffers except for ACME contained 1000 (μ)/mL Recombinant RNase Inhibitor (Takara); all equipment coming in direct contact with the cell suspension was coated with the resuspension solution containing 40 (μ)/mL of the inhibitor. For the same reason, all steps except for the initial dissociation and the EdU Click-iT™ reaction were performed on ice. Centrifugation was performed at 2000 × g and 4 °C (Centrifuge 5910 R; Eppendorf).

### Fluorescence-activated cell sorting (FACS)

EdU-treated cells and the corresponding control were sorted using BD FACSMelody™ Cell Sorter and FACSChorus v. 3.0 software (BD Biosciences). All other samples were processed using BD FACSAria™ III Cell Sorter and FACSDiva software v. 9.0.1 (BD Biosciences). In order to prevent RNase contamination, the devices were cleaned prior to cell sorting and maintained at 4 °C. Sorting was performed in single-cell mode, using a 70 μm nozzle. Gates were set to eliminate debris and multiplets, and sort for cells in G1 and G2 phase, as well as EdU⁺ cells (as shown under FACS Plots 1-4 available at the Zenodo archive). 15000 cells from the EdU⁺ G1 and G2 gate (for the EdU-treated sample) and G1 and G2 gate (for the transcriptomic atlas) and 20000 cells from the P1 gate per sample (for the WT and *c-ops1*[Δ8/Δ8] dataset) were sorted into wells of a 96-well plate (MicroAmp Fast Optical 96-Well Reaction Plate with Barcode (0.1 mL); Life Technologies), containing reagents of the 10x Chromium single cell 3′ reagents kit v. 3.1 (10x Genomics): 18.8 μL RT Reagent B, 2.4 μL Template Switch Oligo and 2 μL Reducing Agent B; following the sorting, enough RT Enzyme C was added to each well to result in 52 μL of total volume.

### Library preparation and sequencing

Libraries and sequences were generated at the Vienna BioCenter Next Generation Sequencing Facility. Suspensions were loaded onto a Chromium Controller chip (10x Genomics) and 3′ GEM libraries were constructed according to the manufacturer's protocol. Library quality control was performed, and the samples were sequenced and demultiplexed; sequencing was conducted on the NovaSeq 6000 sequencing system (Illumina), using the NovaSeq 6000 S4 Reagent Kit v. 1.5 (300 cycles) and following the XP workflow according to the manufacturer's instructions.

### General scRNA-seq analysis pipeline

Raw sequencing data (available at the Zenodo archive) were processed and mapped against version 1.0 of the *Platynereis* genome[95], following the default pipeline of CellRanger v. 7.0.1 or 8.0.0 (for the WT and *c-ops1*[Δ8/Δ8] dataset), forcing the programme to recover 10000 cells from each sample. Mapping results and statistics are listed in Genome Mapping 1–4 available at the Zenodo archive. Unless otherwise stated, all further analyses were conducted using the Seurat package v. 4.4.0[46], run on R software v. 4.4.1 through RStudio interface v. 2023.09. Comprehensive code with step-by-step headings can be found in the

Zenodo repository. In brief, barcodes, features and count matrices were converted into Seurat objects and quality control against multiplets and empty cells was performed for each library individually, removing cells with outlying read and feature numbers. Libraries were randomly downsampled to match the size of the library with the lowest number of cells, in order to ensure a faithful representation of cell type ratios across the maturation stages in the general and the subsequently subclustered datasets. As a control, the libraries in their original size were processed in the same manner as the downsampled ones (Supplementary Fig. 1f, g). Downsampled and control datasets were generated by merging the respective libraries. Merged datasets were normalised, the 2000 most variably expressed genes were identified for each library individually, the lists were collated and used to scale and centre the data. For comparison, a downsampled and control dataset were generated following the standard Seurat integration pipeline: individual libraries were normalised, the 2000 most variably expressed genes were selected for each library, 2000 representative integration genes were selected for all libraries together and integration anchor cells were identified based on the gene list. Libraries were integrated and the resulting dataset was scaled. Principal component analysis (PCA) and UMAP reduction [96] were conducted on all datasets, discarding all but the first 40 or 35 principal components for merged and integrated datasets, respectively. Dataets were clustered and a clustering tree was generated, in order to select a resolution producing a stable number of clusters, using the Clustree package v. 0.5.0 (Supplementary Fig. 1d, e)[97]. For subcluster analyses, cell populations of interest were selected in each library separately and Seurat objects were generated from the subsets. The new libraries were then merged and the resulting dataset processed, annotated, analysed and visualised in the same manner as the original dataset, with the exception that the list of 2000 most variable genes used for scaling was imported from the full subset, as opposed to from each individual library.

### Predictive dimensional reduction and annotation of new datasets

To project the EdU library, as well as the WT and *c-ops1*[Δ8/Δ8] dataset and its corresponding eye- and transcriptomically related cell subset (queries) onto the UMAP-reduced space of the existing brain atlas and eye- and transcriptomically related cell subset (references), and to predict the annotation of the cell populations in the new datasets, a Seurat object was generated out of the new libraries as described above. Following this, a protocol based on the Seurat label transfer pipeline was followed: the UMAP model of the reference library was returned, and the transfer anchors generated setting the reduction model to UMAP. For joint visualisation of the reference and query, reference UMAP embeddings and predicted UMAP embeddings were extracted, collated, imported into a new dimensionally reduced object and visualised. Transfer anchors were then used for predicting the cell labels in the query (set to clusters and super-clusters), and the predicted annotations imported into the object's metadata. As a control, the *c-ops1*[Δ8/Δ8] and WT dataset and its eye- and transcriptomically related cell subset were dimensionally reduced, and the projected cell labels were transferred following the standard pipeline described above.

### Annotation of the adult *Platynereis* head/brain atlas

For annotation, we made use of a systematic comparison of genome-predicted transcripts against available sequence resources[45]. Where available, entries were named based on BLASTN hits to published *Platynereis* sequences for which phylogenetic analyses had been performed; alternatively, best BLASTX hits in RefSeq proteins of selected landmark organisms was used, adding the suffix -like to the name if no phylogenetic analysis was performed. Entries with no BLAST hits were called by their locus identifier (XLOC number) in the annotated genome draft kindly provided by D. Arendt and K. Mutemi[95]. For dataset

annotations, marker genes of each cluster were computed using the nonparametric Wilcoxon rank sum test (default setting) and gene co-expression analysis was run using the fcoex package v. 1.10.0[98] (Supplementary Fig. 2a–m and Supplementary Data 1, 2). An fcoex object was created from the normalised count matrix and Seurat generated clusters; gene co-expression modules and correlation networks were generated, and the dataset reclustered in order to visualise the module-based populations. The results of cluster marker identification and gene co-expression analysis were cross-compared with literature on *Platynereis* and vertebrate cell types and known gene expression domains; this was used to assign putative identities to clusters or group clusters into a super-cluster. *Platynereis* genes mentioned in the text are tabulated in Supplementary Data 5 along with their database identifiers. For quantifying neuronal signatures (Fig. 1d), a module containing *Platynereis* genes matching *syntaxin-1/2/4/11, syntaxin-16, snap-25, vamp, syt-1, syt-4, syt-7, syt-9, syt-12, syt-15, syt-14/16, syt-17, elav1, and elav2* was visualised (for gene identifiers, see Supplementary Data 5).

### Differential expression analysis

In order to identify the genes most differentially expressed between EdU-treated and untreated worms (Supplementary Data 4), as well as between non-reproductive and reproductive worms (Supplementary Data 8), differential expression was performed between the corresponding libraries. Merged datasets were generated containing the libraries of interest, as described in "General scRNA-seq analysis pipeline". Cells of respective libraries were combined into two lists, which were used to generate new identities in the integrated datasets. The identities were then treated as two clusters, and the genes most differentially expressed between them were computed using the default parameters. To determine the genes differentially expressed between EPs and all brain cells, as well as support cells and all brain cells in Supplementary Fig 5g, h and Supplementary Data 6, the procedure was followed as described, with the exception that the negative binomial distribution statistical test was used, in line with methods used in ref. 56.

### Pseudotime analysis

To determine and visualise lineage trajectories, selected cell populations were analysed in pseudotime. The Seurat object was manually converted into a Monocle3 object, and the differentiation trajectory was predicted following the standard vignette of the package Monocle3 v. 1.2.9[62]; cells were then ordered from least to most differentiated along the trajectory, selecting the Monocle3-suggested root nodes with the expression of expected marker genes as the starting points of each individual leaf of the trajectory. Marker genes were selected either for their known involvement in neurogenesis and proliferation in *Platynereis* or through differential expression analysis between EdU and premature or sampling time-matched control library in the eye- and transcriptomically related cell subset (Supplementary Data 4). Expression of differentially expressed genes was individually screened regardless of putative gene function, and very broadly and very sparsely expressed genes were excluded; the resulting genes were plotted as a cumulative module to indicate populations derived from proliferating cells.

### Transcriptomic data visualisation

For visualisation of gene expression, non-biological zero values were imputed using the ALRA package v. 1.0.0[99], unless otherwise stated. Colour palettes used in plotting were selected from the package Paletteer v. 1.5.0. All plots were exported in the PDF format. Gene expression was plotted either individually or as an average expression level of several genes generated using the standard function for calculating cumulative module scores in Seurat. Dotplots were generated with the help of scCustomize package v. 1.1.1; otherwise, the standard

plotting functions of respective packages were used with the following exceptions: in Fig. 1c, clusters and super-clusters transferred from the integrated dataset onto the merged dataset were plotted and exported individually, then overlaid in Adobe Photoshop 2024 as separate layers, using the layer blending option Linear Dodge (Add). Bubble plots in Fig. 2g, h were generated using GraphPad Prism software v. 10.1.0, from differential expression analysis results exported as comma-separated files. Stacked bar plots were generated using Microsoft Excel (Microsoft Office Professional Plus 2019). In Fig. 3n and Supplementary Fig. 8g, the differentiation trajectory and pseudotime-ordered cells were plotted and exported separately, then overlaid in Adobe Illustrator 2023. In Fig. 4c, d, cells coloured in UMAP-reduced space were additionally outlined in Adobe Illustrator 2023. Scatterplots in Fig. 6g–k were generated from unimputed gene expression values that were exported as comma-separated files for each library individually and plotted in GraphPad Prism software v. 10.1.0 as superimposed scatter plots with manually added significance values calculated as described in "Differential expression analysis". The order of cells in all scatterplots in Fig. 6 was randomly shuffled in Adobe Illustrator 2023 to ensure faithful representation of all datasets in a plot. Where applicable, genome identifier numbers were substituted with gene names in Adobe Illustrator 2023

## Worm eye size scoring and statistical analysis

To estimate the eye size in reproductive animals, female and male worms were anesthetised as described above and their heads imaged with the SMZ18 stereomicroscope, DS-Ri2 camera and NIS Elements imaging software (Nikon) under the bright field setting. For each individual, head area and the left and right eye areas were calculated individually in a Z-projected image using Fiji v. 2.9.0[100] by manually selecting the region of interest. Eye areas were then summed up. The head area was determined following the head border, excluding the palpae, cirri and the area posterior to the nuchal organ commissure. Total eye area was normalised to the head area for each worm. Data were plotted and statistically analysed in GraphPad Prism software v. 9.5.1 with an unpaired $t$ test. To compare the eye size between WTs and $c$-$ops1^{\Delta 8/\Delta 8}$ animals, the worms were processed as described in "EdU labelling" and "EdU detection, HCR and immunohistochemistry (IHC)"; the nuclear stain channel was imaged as described in "Microscopy and image processing" and the eye areas calculated in a Z-projected image in Fiji v. 2.9.0 following the borders of both posterior eyes. Data points were plotted as replicates (two posterior eyes per worm) and analysed with a nested $t$ test in GraphPad Prism software v. 10.1.0; all measurements available in Supplementary Data 3.

## EdU labelling

Proliferating cells were labelled using an EdU incorporation assay as previously described[77]. Briefly, selected worms were transferred into small glass beakers and incubated with 10 μM EdU dissolved in worm culture water for 4, 16 or 96 hours; in the case of a 16 h pulse, worms were sampled at ZT 10 and incubated overnight, otherwise as stated in "Worm staging and sampling". For experiments involving a chase period, EdU solution was discarded following the completion of a pulse, the animals and the beaker were rinsed several times with worm culture water, and 100 μM thymidine dissolved in worm culture water was added. The animals were left to incubate for 168 hours (1 week). Control worms were incubated in pure worm culture water during the pulse and thymidine solution during the chase.

## EdU detection, HCR and immunohistochemistry (IHC)

Depending on the experiment, EdU detection was either conducted on single-cell suspensions for scRNA-seq experiments or on whole worm heads for imaging experiments; for the latter, EdU was either detected on its own following the manufacturer's protocol, or in conjunction with the protocol for simultaneous HCR, IHC and EdU[77]. In short,

following the pulse and the optional chase, head tissue was sampled, fixed and dehydrated. The following day, samples were left to incubate overnight in a solution containing HCR probes against transcripts of interest; next, HCR amplifiers were added, and optionally, EdU Click-iT™ reaction was performed according to the manufacturer's protocol. In case that IHC was being conducted in parallel, the primary antibody against the protein of interest was added to the reaction mix and left to incubate overnight. The following day, HCR amplification was terminated, and optionally, the secondary antibody was added; after an overnight incubation, the samples were washed, mounted and imaged. Control samples received HCR probes against unrelated genes designed for non-annelid species and no primary antibody.

## EdU signal quantification

To quantify proliferating cell numbers between distinct worm stages, wild-type worms were labelled with a 16-hour EdU pulse and imaged with the Axio Imager Z2 upright microscope (Zeiss), using the Plan-Neofluar 10x/0.3 (dry) lens, AxioCam MRc5 camera and ZEN Blue software v. 3.3.

For assessing the impact of light and $c$-$opsin1$ on proliferation, wild-type (Vienna PINK strain), homozygous $c$-$ops1^{-/-}$ and $c$-$ops1^{+/+}$ worms – synchronised to the same developmental stage and lunar phase – were maintained for 14 days in constant darkness ("DD regime") or under 16 h:8 h light:dark conditions ("LD regime"). All handling of "DD-regime" animals (feeding, transfer, maintenance) was performed under red light. For 24 h additional hours, each group received a 24 h pulse of 10 μM EdU in filtered seawater.

Immediately after the pulse, whole worms were fixed for one hour in 4% paraformaldehyde (PFA), then dehydrated and stored in 100% methanol at −20 °C. The following morning, samples were rehydrated through a graded methanol/PBST series (25%, 50%, 75%, 100%), decapitated to isolate heads and processed following the SHInE protocol[77] to detect the EdU staining.

For quantifying EdU + cells between distinct worm stages, regions of interest were selected in Z-projected images as described in "Worm eye size scoring and statistical analysis". EdU-labelled nuclei within the region of interest were manually counted, and normalised to the head area (Fig. 2a–e) or eye area (Fig. 5b). Data points were plotted individually or as replicates (two posterior eyes per worm) and statistical significance was calculated in GraphPad Prism software v. 10.1.0 with a one-way ANOVA test and Tukey's multiple comparisons correction.

For the comparison of EdU+ cells in $c$-$opsin1$ mutant and WT eyes under different light regimes, eyes were imaged using an Olympus IX3-series spinning disk confocal microscope equipped with a 100× objective. Z-stacks were acquired at 0.5 μm intervals. Maximum intensity Z-projections were generated, and eye outlines were manually traced to define regions of interest (ROIs) for EdU quantification and eye area measurement (μm²). EdU-positive (EdU+) cells were segmented using the Cellpose-SAM segmentation algorithm[101]. Ambiguous ROIs arising from overlapping cells were manually inspected using the raw Z-stack images. The total number of EdU+ cells was normalised to eye area and reported as cells per μm². Dot plots represent the average of left and right eye values. Statistical analysis was performed using one-way ANOVA followed by Tukey's multiple comparison test; all statistical analyses were performed in GraphPad Prism (v. 10.1.0); all measurements are available in Supplementary Data 3.

## HCR probes and antibodies

All HCR probes against genes of interest were designed using the HCR Probe Maker software (insitu_probe_generator v. 0.3.2), installed with Python v. 3.10.2 and JupyterLab v. 3.4.0[102], with the exception of $Pla$-$du\_prox$, which was designed and manufactured by Molecular Instruments. Using CLC Main Workbench v. 22, probe pairs were mapped against the sequences of interest, and those binding to highly

polymorphic regions were discarded. B1 amplifier coupled to Alexa Fluor™ 546 fluorophore and B2 amplifier coupled to Alexa Fluor™ 647 fluorophore were used for imaging. HCR probe sequences are tabulated in Supplementary Data 11. Antibodies used: Anti-Green Fluorescent Protein Antibody, polyclonal, Aves Labs (GFP-1020); dilution: 1:200; goat anti-Chicken IgY (H + L) Secondary Antibody, Alexa Fluor 488 coupled, Thermofisher (A-11039); dilution: 1:500.

## Microscopy and image processing

Fluorescently labelled worm heads were imaged with a LSM 700 inverted confocal microscope (Zeiss), using EC Plan-Neofluar 10x/0.3 (dry), Plan-Apochromat 20x/0.8 (dry), LD LCI Plan-Apochromat 25x/0.8 (oil immersion), Plan-Apochromat 40x/1.3 (oil immersion) and Plan-Apochromat 63x/1.4 (oil immersion) lenses and the ZEN (black edition) desk software v. 2.3. In Fig. 3a–d, the imaging of unprocessed worm heads for size comparison was achieved with the Axio Imager Z2 upright microscope (Zeiss), using the Plan-Neofluar 10x/0.3 (dry) lens, AxioCam MRc5 camera and ZEN Blue software v. 3.3, set in brightfield mode. Parameters for acquisition were set according to a representative treated sample and applied to other samples belonging to the same experimental batch. Images were processed with Fiji v. 2.9.0[100], optimising brightness and contrast for each channel separately, generating a Z-projection out of selected slices in an image stack using the Maximal Intensity option and optionally enhancing the sharpness of the resulting image. For some image stacks consisting of a larger number of slices, the Stack Contrast Adjustment plugin was used prior to Z-projecting the image, in order to compensate for the loss of fluorescence intensity in deeper tissue layers due to photon scattering[103]. Settings were adjusted to best fit a representative treated sample and applied to the control sample of the same batch; for some of the samples not used for quantification, the settings of the nuclear stain channel were adjusted individually due to the differences in tissue thickness and photon scattering at short wavelengths. Outlines were added, and panels were arranged in Adobe Illustrator 2023. To generate 3D animations (Supplementary Movies 1–3), images of the posterior eyes of EdU-treated animals (16 h pulse) were taken with a 100x magnification objective with 0,2 μm Z-step. Images were preprocessed with rolling ball background subtraction with a radius of 50 pixels, and 3D volumes were visualised using the 3D Viewer plugin from Fiji.

## Statistics and reproducibility, sample sizes

Statistical tests used for data analyses are indicated in legends and methods, where applicable. For the presented stainings (EdU incorporation, HCR-mediated visualisation of gene expression), documented specimens generally represent groups of ≥ 5 specimens that were visually inspected, as an empirical value for reliable patterns. For scRNA-seq analysis of wild-type and c-opsin1-mutant specimens, three independent biological replicates were analysed.

Numbers of worms for scRNA-seq experiments (see "Worm staging and sampling") were chosen in accordance with the different sizes of heads of the respective stages, aiming at similar amounts of cells to be sorted from each sample. Similarly, the number of animals sacrificed for EdU-positive cell sorting was planned based on pilot experiments estimating the number of EdU-positive cells obtainable per head. For experiments involving the quantification of eye size in female and male worms, 20 male and 23 female worms were used. For the quantification of EdU-labelled cells, in worm heads and eyes, 5–8 worms were used per sample to maintain the feasibility of manual cell counting and adhere to standard sample sizes in the field.

## Gene ontology term enrichment analysis

GO terms linked to genes differentially expressed between non-reproductive and reproductive worms were obtained using the web server ShinyGO v. 080[104]. Platynereis genes were matched against the human proteome, and where applicable, Homo sapiens UniProt IDs were used for comparison, as described[45]. Hits of all genes were set as the background, and the hits of differentially expressed genes as the query; the search was filtered for biological processes, and the analysis performed using the default settings of the web server. Highest ranking terms were visualised with the built-in chart function. Conversely, to probe for enrichment of terms related to molecular mechanisms and cellular processes of interest, a reverse search was performed in the AmiGO 2 Ontology database[105] using default filters, and resulting GO terms affiliated with or containing the keywords were looked up in the results of the enrichment analysis (Supplementary Data 8, 9).

## Phylogenetic tree construction

To determine potential orthologs of vertebrate and Drosophila protein-coding sequences in the worm, a cDNA database with genome-predicted transcripts was used. For the reported phylogenies, reference sequences were re-assembled from available cDNA resources of the investigated laboratory strains[40,49,56], also correcting cases of missing or mis-predicted genes (tob and lamin). Best reciprocal matches from Mus musculus and Drosophila melanogaster were retrieved from UniProt, using blastx/tblastn queries in CLC Main Workbench v. 22; transcript variants with the longest reading frame were translated to protein, and additional protein hits from UniProtKB/SwissProt and RefSeq were obtained to identify possible matches in additional reference species (Homo sapiens, Mus musculus, Gallus gallus, Xenopus laevis, Xenopus tropicalis, Strongylocentrotus purpuratus, Saccoglossus kowalevskii, Capitella teleta, Crassostrea gigas, Mizuhopecten yessoensis, Pomacea canaliculata, Biomphalaria glabrata, Drosophila melanogaster, Tribolium castaneum, Apis mellifera, Daphnia pulex, Priapulus caudatus, Lottia gigantea and Helobdella robusta) Hit sequences were aligned in CLC Main Workbench v. 22 and exported in FASTA format for phylogenetic tree construction. Maximum likelihood analysis was run on the sequences using the IQ Tree web server, and the best substitution model was selected by the web application W-IQ-TREE, using the ModelFinder algorithm operating in jModelTest mode[106,107]. A stochastic algorithm was implemented to achieve a high likelihood ratio in order to locate local optima in the tree space, and a bootstrap approximation was performed with UFBoot2[108], using the default settings and performing 1000 iterations. Trees were visualised and arranged using the Interactive Tree of Life tool[109]; trees were manually rooted using a protein outgroup obtained using the second-best hit in the mouse, Drosophila or Platynereis proteome as a starting point. Tree construction was reiterated several times in order to eliminate duplicates and outliers, resulting in a stable tree containing sequences of all relevant reference groups. A comprehensive list of proteins used in the final phylogenetic tree assembly and their respective accession numbers in different reference organisms can be found in Supplementary Data 5.

## Protein domain analysis

Phylogenetic trees were supported with comparative analyses of functional domains in orthologous proteins from selected organisms. Simple Modular Architecture Research Tool[110] was used to identify and visualise domains in the sequences of interest, which were subsequently annotated by retrieving family and domain information for each individual protein from the UniProt database of the respective organism.

## Reporting summary

Further information on research design is available in the Nature Portfolio Reporting Summary linked to this article.

# Data availability

All primary data generated for this manuscript are available online, as part of the Supplementary Information file and in electronic archives.

The single-cell RNA sequencing data generated in this study have been deposited in the European Nucleotide Archive under accession number PRJEB98620. The newly described *Platynereis* genes characterised in our study have been deposited in the GenBank database (identifiers PV864739-PV864760), and are tabulated in Supplementary Data 5 along with other *Platynereis* genes referred to. The Zenodo repository https://doi.org/10.5281/zenodo.17349847 contains additional FACS plots and genome mapping information.

## Code availability

R scripts used for data processing and resulting R objects (RDATA files) are included in the Zenodo repository https://doi.org/10.5281/zenodo.17349847.

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

## Acknowledgements

We are grateful to Andrij Belokurov, Margaryta Borysova and Netsanet Getachew for routine worm cultures and genotyping support, Lena Stumbauer for practical help, as well as all members of the Tessmar-Raible and Raible labs for constructive discussions. This work was supported by, Helmholtz Society, distinguished professorship by the Alfred Wegener Institute Helmholtz Centre for Polar and Marine Research (K.T.-R.), H2020 European Research Council, ERC Grant Agreement #819952 (K.T.-R.), Austrian Science Funds (FWF), SFB F78 (F.R., K.T-R; https://doi.org/10.55776/F78), the Human Frontier Science Program (HFSP), #RGP021/2024, https://doi.org/10.52044/HFSP.RGP0212024.pc.gr.194174 (KT-R), University of Vienna Research Platform SinCeReSt (F.R.), For open access purposes, K.T.-R. has applied a CC BY public copyright license to any author accepted manuscript version arising from this submission. We acknowledge support by the Open Access publication fund of Alfred-Wegener-Institut Helmholtz-Zentrum für Polar- und Meeresforschung. None of the funding bodies was involved in the design of the study, the collection, analysis, and interpretation of data or in writing the manuscript.

## Author contributions

Conceptualisation: N.M., F.R., and K.T.R., Methodology: N.M., F.S., P.O.B., C.L.V.G., G.A., F.R., and K.T.R., Investigation: N.M., F.S., P.O.B., C.L.V.G., G.A., F.R., and K.T.R.,.Visualisation: N.M., F.S., P.O.B., C.L.V.G., and G.A., Funding acquisition: F.R. and K.T.R., Project administration: F.R. and K.T.R., Supervision: F.R. and K.T.R.,. Writing – original draft: N.M., F.R., and K.T.R., Writing – review & editing: N.M., F.S., P.O.B., G.A., F.R., and K.T.R.

## Funding

## Competing interests

The authors declare that they have no competing interests.
