## [Peer Review file · Nature Communications]

Light-modulated stem cells in the camera-type eye of an annelid model for adult brain plasticity

Corresponding Author: Professor Kristin Tessmar-Raible

Version 0:

Reviewer comments:

Reviewer #1

(Remarks to the Author)

Platynereis is a powerful alternate system to study vision and clock (among other things) in an invertebrate other than *Drosophila* or *C. elegans* model systems.

In this paper, the authors performed single-cell mRNA sequencing of the brain and visual system of the worm and address more specifically eye development. This is of particular interest since *Platynereis* has a camera-type eye similar to the cephalopods' eye but that evolved independently. Furthermore, these animals exhibit continuous growth until they reach the reproductive state when they get ready to die to liberate their gametes.

There are two important conclusions from this work, both of significance. However if one is well supported by the data, the second one is not terribly convincing and might not be correct

- The scRNAseq could have used many more cells. The results exhibit a highly surprising enrichment of sensory mechanosensory cells, as well as eye cells and neurosecretory cells. Mushroom body cells are present in low abundance and ependymocyte cells are abundant. Even of these are all neurons, their under-representation is not expected: who process all this sensory information?

- The strong conclusion is that the eye likely grows from a 'ciliary marginal zone-like'. The authors present a series of strong arguments for progenitor cells at the periphery of the eye and the in situ/stainings show clearly a set of cells that divide until the reproductive state and are really excellent candidates for being marginal cell-like. It seems that this was not the prevailing model for eye growth but these data are convincing

- The second argument is that a ciliary opsin, normally found in the brain is also expressed in a very small number of cells that are EdU positive after a short pulse, and are thus likely the progenitor cells described above. The images are difficult to read as expression is very sparse and at low levels with little dots on the picture that are hardly visible. Since only a small subset of the dividing cells express it, their significance is difficult to assess, especially since the ciliary opsin gene is not expressed in the mature eye.

But the major issue is the claim that a mutant in *c-opsin1* leads to eye reduction and to the loss of these cells. The authors present a good quantification of these effects and it might be indeed true that the eye is smaller and that eye cells are lost. However, and if this reviewer reads the figures and legends well (a very difficult task as the legends very cursorily describe the panels), it appears that many other cells types that do not express *c-opsin* are also decreased very significantly, even more so than eye cells expressing the *c-opsin*: "Contribution of cells from libraries generated from *c-ops1Δ8/Δ8* specimens versus those generated from WT controls to the different clusters differed significantly in five cases, in which four showed a significant underrepresentation". It is very difficult to explain this loss and the easiest interpretation is that the *c-opsin* mutant chromosome has off-target mutation(s) that affect to a minor extent the fitness of the animal, leading to small eyes and fewer cells. Is the animal smaller in general? i.e. the size of the eye should be presented as a ratio to the total size of the animal. It would not be reasonable to request a rescue experiment in the mutant animal, but the interpretation is likely not correct. The authors should test whether raising *Platynereis* in complete darkness leads to the same results, which would then support the argument that the function of *c-opsin* has a developmental role

The conclusion from the authors in fact suggests that this is the case: "While such an analysis represents the impact of the mutation over the entire life of the animal and hence may show effects that are also outside of the *c-opsin1* expressing cells

themselves, this finding is in line with the observed reduction of eye size in mutants and the resulting hypothesis that c-opsin1 influences the division rate and/or differentiation of r-opsin1+ eye cells."

Other points:

- Fig.3 A-D: the co-expression of r- and c-opsins is difficult to see (D: which cells co-express?)

It is surprising that both opsins are expressed in dividing cells. In all other systems, opsins are post-mitotic. If, as argued in the discussion, c-opsins are remains of evolution, this would be possible, but expression of r-opsins that will induce phototransduction is more surprising.

In conclusion, extensive data and some nice demonstration of eye growth but the title of the paper that c-opsin functions in a ciliary marginal zone-like might not be justified

Reviewer #2

(Remarks to the Author)

The paper entitled "A c-opsin functions in a ciliary marginal zone-like stem cell region of an invertebrate camera-type eye" compiles several single cell RNA-Seq datasets performed on the head of the annelid *Platynereis dumerilii* representing a fundamental resource for the *Platynereis* community in the future. The main focus is on the neurogenesis process in brain and eye and the changes in the neuronal stem cell population at different phases of the life cycle. The dataset analysis is very thorough, while the HCR results and imaging processing/quantification has to be significantly improved to support the Authors conclusions.

Major comments:

1. While the title and the introduction are mainly focused on the eye and the CMZ, a good part of the results and the discussion is more focused on the neurogenesis and the eye seems an area to look at as well as the brain. In the manuscript it is difficult to grasp what was the main question the Authors were trying to answer. I would recommend making the question clearer and why this experimental approach is the best way to answer it. I would also encourage the Authors to help the reader understanding the jump from the c-opsin mutant to the last part of the manuscript. Currently, these 2 sections seem disjunct from each other. Finally, I am wondering if a title about neurogenesis would be more appropriate and would attract a broader readership interested in neuron formation instead of focusing on eyes and CMZ.
2. SoxB1 and c-opsin HCRs are extremely weak, mainly considering the high number of probe pairs used, and currently it is difficult to determine if the cells are really positive. To support their conclusions, the Authors have to show stronger evidence or ordering a new set of probes or searching for markers co-expressed in the same cells. The images provided are not supporting that the Edu positive cells are expressing c-opsin (Figure 3L-M).
3. If the Authors can answer the previous point, they will need also to update the models in Figure 2T and 3K since the positivity seems to be localized all around the lens and not only in the most superficial area like it is currently presented in the cartoons. These models could be misleading that the positivity is a circle closed to the "cornea", while based on the images showed so far, these cells are localized as a cup or semi-sphere all around the lens. A 3D reconstruction through Imaris or a video of full z-stacks can help elucidating this point. Ideally a quantification of the position of positive cells would be ideal.
4. In Figure 4, the Authors are focusing on the decrease of cells in mega-cluster 3. Please, discuss also the decrease in 10, c19 and the increase in c20. This seems a very interesting piece of data that is not followed up or mentioned for speculations. A marker could be selected for performing HCR to see where these cells are express. Where else is c-opsin expressed in the head?
5. The last section of the manuscript is less focused on the eyes and is supported mainly by in silico analysis. The lack of functional data about foxo and prox in *Platynereis* adult worms together with the HCR on only 2 marker for neurogenesis and only one marker for quiescence does not allow for strong conclusions about the state of the cells. Please, tone down the conclusions or add additional validating data to support the model in 5X. In Figure 5, at which part of the head and eye are we looking at? It seems in the adults all the cells have both foxo and prox genes expressed, is that true or is it only a subpopulation of cells?

Minor comments:

- The abstract mention "modulation of the environment" but the Authors never tested that. Please, replace it with "throughout life cycle".
- I tried to look for the reference 4 that is used several times to talk about *Platynereis dumerilii* eye anatomy, but I was not able to find it. I am wondering if there are more recent references that can be used to support that the *Platynereis dumerilii* has a camera-type eyes. Reference 25 calls the *Platynereis dumerilii* eyes as pigment-cup eyes and I could not find many other information about their anatomy.
- Many time the Authors are using "brain" dataset instead of "head" dataset (such as in the title of Table S1 and S2; line 41 page 5; panel 2E; line 38 page 7; line 1 page 8; line 1 page 10; etc)
- The Authors should implement the explanation about how they got the modules and how this helped them manually design the mega-clusters since in the majority of the cases they do not correspond to each other (modules 3, 9, 10, 11, 12 and 13 include many more clusters than the mega-clusters).
- The scRNA-Seq has been run on the entire head but it looks like there are no epithelial cells, gland cells, fibroblasts, etc. Which other tissues other than brain and eyes are present in the head of *Platynereis* and why they are not represented in the dataset?
- Table has more clusters with rhodopsins and opsins than the one included in the mega-cluster 3. Why are those clusters not considered part of the eyes? Did the Authors run HCR on these genes to verify the location of these cells?

- Please, add more info about fut10 and why it was selected as marker.
- In the eye subset there are cells expressing pcna but those were not analyzed or discussed in the generation of the model. It could be that pcna positive cells are those giving rise to the support cells and c-opsin positive cells those giving rise to the PE.
- In the longer Edu pulse with chase experiments, together with looking at the potential migration of the cells, a co-labeling with r-opsin or fut10 could support the claims about the presence of a population of stem cells that divide and then migrate to differentiate in the retinal cell types.
- It would be helpful to highlight in the discussion more evolutionary aspects of this study and comparison with what is known in other animals other than mammals and vertebrates.
- Please, always add "in silico" in front of aNSC since this is how these cells are defined, and no functional validation has been provided.

Figure 1:

- It would be helpful to bring in the main Figure the Suppl panel S9D. The use of the bubbles could be confusing since in each clusters we can find a few cells whose identity has been assigned to a different cluster, but included in the "wrong" bubble. I would recommend to remove the bubbles and use color code or numbering.
- Panel 1L has to be replaced with the S2C since that is the specific Edu control that was run side by side and it would provide the proper comparison. Also, are the genes showed in 1L and S2C manually selected or are they all the differentially expressed genes? Please, clarify this point in the legend and provide a full list of the differentially expressed genes.

Figure 2:

- In panel 2O seems that the migration is happening only on one side of the eye. Is this true? Please, provide more representative images and Z-stack and provide the specific orientation of the eye.
- The nuclei of 2H are very different in terms of number and localization than 2P as well as between 3H and 3L. Having more uniform images would help the reader.

Figure 3:

- Why the outlined dots in 3D have different colors?
- Could the Authors explain the difference in intensity between 3E and 3H? Although one is a max projection and the other is a cross section, the signal is so bright in 3E that it seems that there might be cross-sections with more intense signal.

Figure 5:

- Is a list in 5c generated by the Authors or was it previously published? Please, add more information.

Mat and Methods:

- Line 26 page 14, there is a broken sentence
- Line 9 page 17, there is a "(reference)"
- List of used Ab is missing
- Table overview stops at Table S8, please add S9-11

Reviewer #3

(Remarks to the Author)

In this manuscript, Nadja Milivojev et al. investigate proliferation and growth in the retina of the bristleworm, *Platynereis dummerilii*. They identify cells in the cell cycle using Edu and investigate retinal cell function through single cell sequence analysis in order to identify a retinal stem cell population. In addition, they use a c-opsin mutant to investigate the role of c-opsin in the growth of the adult eye.

I think that understanding cell cycle control, the process of neurogenesis and regulation of nervous system growth across species is an exciting and impactful question, and not well understood. I believe that the field of evo-devo has much to be gained from leveraging transgenic and mutant lines in non-models to better understand fundamental questions in nervous system evolution. I believe the adult *Platynereis* visual system is an excellent context to understand visual system growth and the impact of life history and sexual maturity on growth and quiescence. I also find the proposed conclusions of the paper to be plausible. It is likely to me that there is a proliferative zone, potentially at the periphery of the retina, that contributes to growth prior to sexual maturity.

However, there are significant flaws in the manuscript as it exists that make it difficult to interpret the authors findings and therefore does not do the ambitious and exciting intellectual goals of the manuscript justice. I therefore cannot support the claims made in the paper as presented.

Summary of major issues:

There is a need for a basic description and understanding of growth and proliferation in the *Platynereis* retina to make the conclusions the authors are making here. It is impossible to define a *Platynereis* CMZ-like zone without doing a first principles description of growth and changes in the bristleworm eye at each of the stages of interest. It is unclear if the larger eyes actually have more cells or just bigger cells, as previously described. Is there cell death? Where are the phosphohistone H3 or PCNA positive cells? EdU positive cells found in the retina do not necessarily originate in the retina. The manuscript has no standard way of collecting the data across time points or across conditions to be able to make comparisons. Are images always going to be taken from central sections? While some images appear central, many of the images do not. I think the authors are taking superficial or optical sections of their samples. Unfortunately, this makes the data difficult if not impossible to interpret as presented. What are the axes of this organ and the axes of the included images? A more in depth survey of growth using EdU or phosphohistone H3 at different time points is necessary. In addition, many of the HCR stains of soxB1, c-opsin, are not convincing as shown.

Specific Comments:

Line 47: "In the eyes of vertebrates, growth is enabled by a dedicated stem cell area, termed the ciliary marginal zone (CMZ), from which new neurons and pigment cells differentiate"

Please be mindful of your summary about the CMZ. Framing the CMZ as a universal stem cell population required for eye growth in vertebrates is not accurate. CMZ associated growth is primary to fish and amphibians. In mammals, almost all growth is a result of embryonic neural progenitor proliferation and then tissue expansion. The amniote CMZ has limited contributions to growth after juvenile stages.

Line 109:

Please do not use the word bona fide unless you define the contextual meaning. It is not clear what a bona fide stem cell system or a bona fide glial cell (line 161) is. Glial cells in particular are not well defined by markers and there is not functional observations included here.

Line 153: The authors don't make a great case that the variation they observed in the merged dataset isn't batch effects versus standard integration pipelines that show a more homogenous integration of the datasets. They mention that the differences reflect differences in a bulk transcriptome. It would be helpful to include examples of these comparisons in a more obvious fashion.

Line 185: "Ki-67 orthologs have not yet been reported in other invertebrates"

Ki-67 has been previously identified in other invertebrates

- Lee, Chan-Jun, Hae-Youn Lee, Yun-Sang Yu, Kyoung-Bin Ryu, Hyerim Lee, Kyunghwan Kim, Song Yub Shin, Young-Chun Gil, and Sung-Jin Cho. "Brain compartmentalization based on transcriptome analyses and its gene expression in Octopus minor." *Brain Structure and Function* 228, no. 5 (2023): 1283-1294.
- Chuang, Po-Shun, Kota Ishikawa, and Satoshi Mitarai. "Morphological and genetic recovery of coral polyps after bail-out." *Frontiers in Marine Science* 8 (2021): 609287.

Line 182: I would hesitate referencing GFAP over any other vertebrate protein in the tree to understand function here. There is literature on invertebrate and lophotrochozoan intermediate filaments that would be a better reference.

Line 218: In addition to identifying potentially proliferative cells in the retina, Fischer and Bokelmann also hypothesized the growth of the eye prior during sexual maturity was not a result of "significant proliferation" because the cells become enlarged. However, as the authors note, there are clearly EdU positive cells in the retina after a 16 hour pulse. Is there cell death? The authors need to provide a description of eye enlargement during these adult stages to better understand proliferation versus cell enlargement. The cross sections in S4A-E are helpful but the manuscript needs more complete observations of the process of growth/enlargement to make accurate hypotheses.

In addition, I would soften the language that there is "continuous presence of neurogenic cell populations in this organ." EdU positive cells after a 16 hour pulse does not confirm that cells are derived within the organ.

Line 256: In the process of Monocle3 pseudotime analysis, the user chooses the root of the pseudotime analysis. The authors say in the methods that they chose their root based off expression of genes known to be involved in neurogenesis and proliferation. This is reasonable but I would be careful about the wording of this sentence. The authors chose the origin, Monocle3 only showed the trajectory. Otherwise, the use of Monocle3 doesn't tell you anything new.

Line 270-276: It would be helpful to know the stage or age of the animals in these experiments. They look younger than the eyes shown in the fut10 HCR data.

Line 273-275: I'm not quite sure what the authors mean here.

- "Following a long chase period, the number of EdU+ cells in the innermost ring multiplied, and we observed trails of EdU+ cells arching outwards (arrow in Figure 2O), reminiscent of the spreading of fish retinal clones along arched continuous stripes"

Are these images central sections of the worm retina? If so, then there are EdU positive cells everywhere, not just the edges. If this is a superficial image or an angled section of the retina (not a central cross section), including a larger portion of the peripheral retina and anterior of the eye, then maybe the distribution of cells around the lens makes more sense. However, if the authors need to provide better orientation if this is the case. In addition, a superficial survey would not account for proliferation in the central retina if it exists. It is possible that there are clones emerging from the periphery of the Platynereis retina, similar to what is observed in the CMZ, however this data does not show this or refute this as presented. If you are not using central sections, please include a diagram of where the section is in the eye.

Line 280: "In summary, these data support the presence of a stem cell zone at the edge of the retina abutting the glass body/lens, giving rise to new eye photoreceptors and eye support cells."

- I don't think the data support this as presented.
- I would recommend changing some of the timing of these Edu experiments and doing a time course experiment. Immunohistochemistry for phosphohistone H3 or HCR for PCNA would help.

Line 300: "The position of such cells adjacent to the glass body/lens, where the aforementioned results revealed the CMZ-like stem cell zone." All cells are adjacent to the lens in the *Platynereis* retina. I assume the authors mean, the peripheral retina. I would encourage the authors to define the axis of the camera type eye and then have an introduction to those axes early in the paper.

Line 315: Please include images of these mutant eyes and their wild type counterparts in Figure 4. The authors aren't counting cells, they are only measuring area. There are multiple ways that the eye could be smaller, and this question is central to the author's hypothesis. Is there cell death? How old are these animals?

Line 404: "coupled to the unexpected expression of the ciliary opsin gene *c-opsin1*, modulating rhabdomeric photoreceptor cell differentiation."

Isn't the main phenotype in the *c-opsin* mutants identified in the single cell data that there are less eye cells all together, not that differentiation is impaired in the *C-opsin* expressing cells becoming photoreceptors. There aren't more of either the support cells or the progenitor cell in the retina that would suggest that cells are stalled in a progenitor state. I would encourage the authors to be more specific about their hypothesis.

Line 408-411:

"Molecularly, the *gfap* and *ki67* orthologs we find in the worm, and their association with bona fide aNSCs, support the notion that there is an ancient molecular signature of neurogenic/ proliferative brain cells that is shared between bristleworms and vertebrates"

Please soften this language. Again, I would encourage the authors to define what they mean by an adult neural stem cell population and what it means to be bona fide or not. If there is nervous system growth, does that not qualify cells to be neural stem cells? Is there an alternative the authors are considering? *Ki67* is a common proliferation marker from neural progenitors to cancer. It is an interesting finding that *Ki67* may have similar function here, especially in light of the fact that cephalopods also express *Ki67* in their proliferative nervous system. However, a proliferation marker would be expected. The authors did not find a specific *gfap* sequence. They found a homolog of many vertebrate intermediate filaments. I find this study interesting, but I think the statement is not an accurate framing of the conclusions.

Line 437-439: Relevant citation.

• Koenig, Kristen M., Peter Sun, Eli Meyer, and Jeffrey M. Gross. "Eye development and photoreceptor differentiation in the cephalopod *Doryteuthis pealeii*." *Development* 143, no. 17 (2016): 3168-3181.

Line 439-443 This seems like a reasonable hypothesis, and was proposed by Fischer & Brökelmann, however, the presentation of the data is difficult to interpret.

It would be interesting for the authors to discuss the previously published and known regulatory activity of opsins in the context of their finding.

It would be interesting to talk more specifically about growth and quiescence in the context of sexual maturity and/or metamorphosis.

Figure Problems

Figure 1 G, H, I, J

It is difficult to distinguish the EdU labeling from background. It would help if there was a higher resolution, longer pulse or sectioned images. The data as presented is not convincing. In addition, to compare different z-stacks across datasets to show that one group has EdU incorporation, and the other group does not, the Z-stacks need to be of similar size or there needs to be statistical measurements. "72, 76, 36 and 36 μm thick, respectively"

Figure 2 L and T

These summaries are not well supported by the data. I think L is a reasonable guess, but there isn't great evidence for it, I would add qualifying language in the figure legend. Figure 2T is hard to understand relative to the data shown. SoxB1 in situ is not convincing but if SoxB1 were expressed where the arrowheads are pointing, it would not be in accordance with the schematic. Maybe this is a superficial image?

Figure 2 N and O

Again, to make data comparable as presented, the authors need to choose comparable images. These z stacks need to be the same thickness.

Figure 2P and Q

The SoxBI HCR is not convincing, unfortunately. Although according to the single cell data, that may not be totally unexpected. The EdU is convincing in one cell, but it would be helpful to get more time resolution on this experiment. Is there some significance for 4 hours? Or maybe some statistics on the replicates for these data would help?

Figure 3H-J

The whole mount images of the c-opsin HCR are definitely convincing, but these retinal sections are not.

Figure 3L and M

The EdU integration is in more nuclei than the 4 hour pulse in figure 2. Why? Are these representative? Are all these images center sections of the eye? Are these the same age animals?

Figure 4 C and D

Although I appreciate that it is helpful to see distribution of wt and mutant this way, it is not a great way for the reader to understand the statistical significance of your data. If one of your samples is responsible for the majority of difference it is not representative but still appears significantly different. I would suggest the bar graph be made into control and mutant box plots and statistical significance marked or bar graphs with individual lines, not a percent total box.

Version 1:

Reviewer comments:

Reviewer #1

(Remarks to the Author)

The review of the initial paper was quite positive although this reviewer had issues with the developmental phenotype of the c-opsin1 mutants in the ciliary zone.

The authors have addresses many of the problems raised and have explained why several of the pictures were poorly convincing, apparently due to a PDF conversion.

The 36 page (!) response to reviewers attempts to explain and to solve most of the issues. An important additional experiment that appears to confirm a potential role for c-opsin1 is the phenotype observed when the animals develop in the dark. The potential non-autonomy is however not resolved

At this point, and considering the strength of the first part of the manuscript and the fact that the authors have toned down their statements about the light dependence, this reviewer believes that the paper should be published as a significant contribution to understanding how stem cells support the development of a type of eye that grows continuously through life... and might require light!

Reviewer #2

(Remarks to the Author)

The Authors answered the majority of the comments and the manuscript significantly improved. I am excited to see this paper published.

There are a few minor comments that I am going to list below for the Authors to consider:

- The title is better reflecting the story, but it is pretty long, difficult to read and a little bit vague.
- I would encourage to try to connect the first and the second part of the abstract. There is a big jump between the camera-type eyes and the neuroplasticity. I am wondering if moving the concept of the neuroplasticity at the end of the abstract showing how this study is very relevant also to understand neuroplasticity could help the flow.
- The Supplementary figures are not referenced in order in the text.
- It would be very informative, in Fig 4B, to run statistical analysis on LD WT vs LD mutant and DD WT vs DD mutants. I would have expected LD mutant and DD mutant to be more similar to DD WT.
- It would be helpful to me to understand how *Platynereis dumerilii* is defined long-life growth since the Authors are showing a severe decrease in EdU positive cells and in the last Fig an overall decrease in cell proliferation shown through the GO term.

Reviewer #3

(Remarks to the Author)

I still think this paper is important and interesting with high impact.

My primary concerns from the original draft were:

- (1) A lack of convincing in situ
- (2) Difficulty in interpreting the data as presented
- (3) A need for a first principles approach to the question of growth during development
- (4) A number of textual changes concerning interpretation and clarity

The authors have addressed some of my concerns but not all of them. I have included my assessment of these for clarity below.

- (1) I agree with the authors that the jpg/pdf conversion did make a difference in the previous draft. I find the opsin in situ

convincing. Unfortunately, I still find the SoxB1 in situ difficult to interpret and not convincing. I suggest that the authors remove it and any argument based on that expression pattern, other than bioinformatic analysis, from the paper. The new notch in situ is more convincing but it is also not robust. See (2) for additional concerns about in situs and their interpretation.

(2) The authors have now labeled their images in orientation which is a significant help to the reader. I maintain that saying NSC cells are adjacent to the lens is confusing, when all retinal cells are adjacent to the lens in this visual organ, because there is no vitreous.

In addition, and more importantly, now that the authors have made the data easier to interpret by the reader, it has become clear that the foundation of the hypothesis that there is a CMZ-like region on the periphery of the retina is not adequately shown. The authors never show that there are no Edu positive cells in the central retina. Figure 2, O and P show a whole z of the eye, but they show it in max projection. These labeled cells could be anywhere in the eye. The remaining images are superficial cross sections. At the very least, I encourage the authors to include a 3D rendering of the whole z-stack shown in 2O and 2P. The presence of the proliferative markers at the periphery of the retina is as important for their hypothesis as is the absence in the central retina. This includes the in situs and the Edu analysis. Without 3D renderings or central imaging the hypothesis doesn't hold. I understand that the PCNA has been technically difficult but I don't understand the pH3 argument. PH3 shows where cells are dividing, which is one of the fundamental arguments in the paper. An alternate option would be to remove the direct comparison to the CMZ and any conclusive reference to a restricted location of stem cells in the retina.

(3) I still feel strongly the paper would benefit from a first principles approach. With all due respect to the papers that the authors referenced in the rebuttal, current tools have revolutionized the simple experiments that can assess growth and proliferation in the development of this specific visual organ. As this paper itself shows, there are plenty of unknowns to address. Very simply, if the authors had included a time course assessment of growth of the eye, and anti-ph3 in a manner that could be assessed both peripherally and centrally in the retina, the paper would be much stronger. This would also allow the authors to standardize their imaging of the central retina versus the peripheral retinal.

(4) The authors made a number of moderating textual changes that I think are generally fine. I still do not prefer the use of bone fide but will leave its usage up to the editor. I am also happy with the box plots in supplement.

RESPONSE FOR THE REVISED MS

We wish to thank all reviewers for the constructive feedback, which we aimed to address by additional experiments and extensive rephrasing of the text. This has certainly helped us to significantly improve the manuscript.

Below we provide a point-to-point response to each reviewer. To avoid tracing problems- any reference to text or figures is to the manuscript in which the changes are already accepted. (We also provide a version in the track change mode for better comparability to the previous version, but have noticed that depending on the individual Word application settings, can lead to difficulties to find the sentences exactly referred to.)

REVIEWER COMMENTS

Reviewer #1 (Remarks to the Author):

Platynereis is a powerful alternate system to study vision and clock (among other things) in an invertebrate other than Drosophila or C elegans model systems.

In this paper, the authors performed single-cell mRNA sequencing of the brain and visual system of the worm and address more specifically eye development. This is of particular interest since Platynereis has a camera-type eye similar to the cephalopods' eye but that evolved independently. Furthermore, these animals exhibit continuous growth until they reach the reproductive state when they get ready to die to liberate their gametes.

There are two important conclusions from this work, both of significance. However if one is well supported by the data, the second one is not terribly convincing and might not be correct

We thank the reviewer for his/her general remarks on the significance of our work, and specifying where he/she sees need for more analyses and clarification.

- The scRNAseq could have used many more cells. The results exhibit a highly surprising enrichment of sensory mechanosensory cells, as well as eye cells and neurosecretory cells. Mushroom body cells are present in low abundance and ependymocyte cells are abundant. Even if these are all neurons, their under-representation is not expected: who process all this sensory information?

The two main concerns expressed by this comment are

- (i) a possible under-representation of (processing) neurons in the data and**
- (ii) a concern that not enough cells were sampled.**

a) In order to assess if or not we have underestimated the neuronal cell identities, we first extended our analysis on neuronal markers (compared to only three, previously shown in Fig.S8H). This new analysis integrates the expression of 14 broad neuronal marker genes, including various markers for neurotransmitter identities (Fig.S1R-X) and of the synaptic release machinery (module in Fig.1D, but not individually shown to avoid overfilling the supplemental material). Our data show that about half of the cell populations has a neuronal signature.

b) We next wanted to test if this subdivision of about 50% neurons and 50% others (discussion of their identity below) could be caused by undersampling. We merged all libraries generated in the course of the entire study. This merged set of ~ 70k cells represents more than three times of the original set. Importantly, it yields highly similar results as merging of the initial four libraries and no previously unidentified set of cells emerged and the identity of the large clusters was re-identifiable (compare Fig.1D versus Fig. S6N).

This lets us conclude that our data are consistent with the notion that we can identify major cell type clusters of the brain. Small ones like cluster 19 show in the order of 100 cells per brain (cf. Fig. 1K).

Does the distribution between neuronal and other cell types make sense? We would like to point out that we have sampled and analyzed whole heads. The non-neuronal cell clusters include clusters denominated as glial cells, muscles and epidermal cells. All of these are cell types expected to significantly contribute to the head. In analogy- if one was analyzing the cell types of the entire human head, a large proportion would be non-neuronal cells as well.

c) In order to better validate the annotation of our major clusters, we performed additional experiments on the expression of genes that our atlas predicts to mark certain major cell type clusters (new Fig.1G-K), and also systematically tested if marker genes that were previously analyzed for their localization make sense in the context of their annotation. For all cases tested, the denomination based on the enriched transcripts was consistent with the observed patterns. For details, please see text: “To assess how the obtained cell populations distribute..... known sensory structures (Figure 1I, J).”

d) In direct response to the reviewer’s concern that processing neurons are not well covered, we also analysed the expression of the neurexin / nrxn gene, a signature gene of cluster 14 (see Fig. 1F) that our analysis predicted to demarcate mushroom body-like cells. Detection of the nrxn RNA clearly labels the mushroom bodies of the adult brain (Fig. 1H) that comparative neuroanatomy and molecular fingerprinting suggest to be conserved processing centers present in diverse invertebrate brains (cf. Heuer et al. (2010). *Front. Zool.* 7, 13. <https://doi.org/10.1186/1742-9994-7-13>). Taken together with established markers like arx (Fig. 1F; Tomer et al. (2010). *Cell* 142, 800–809. <https://doi.org/10.1016/j.cell.2010.07.043>), these data support the idea that processing neurons are clearly represented in the map.

e) Further in response to the question “where all this information is processed”, we would also like to point out that information processing in animals does not necessarily require dedicated “processing neurons”, but that processing is rather a feature of neuronal networks. The posterior-oval shaped domain (POSD, cluster 2) is a large neuronal center, whose neurons are also neurosecretory, but this does not exclude them from information processing (as is the case in other model systems, such as *C.elegans*).

f) This interpretation of a complex neuronal networks that are well capable of information processing is also supported by our detailed analyses of neurotransmitter marker localization. This revealed complex patterns across the neuronal subpopulation. We reason that these transmitter patterns belong to different neuronal circuits that process different external and internal information. For details, please see the newly introduced text: “When probing the atlas for genes typically associated with..... These populations harbor distinct cell types of known and novel identity that we expect to encode the worm’s sensory and information processing, its complex time-keeping abilities, and central output control.”

- The strong conclusion is that the eye likely grows from a 'ciliary marginal zone-like'. The authors present a series of strong arguments for progenitor cells at the periphery of the eye and the in situ/stainings show clearly a set of cells that divide until the reproductive state and are really excellent candidates for being marginal cell-like. It seems that this was not the prevailing model for eye growth but these data are convincing

We thank the reviewer for their supportive comment. We think that the additional EdU stainings and stainings for markers of cell proliferation performed in the course of the revision have further solidified this conclusion. For details, please see the revised Figures 2 and text in the section: “A ciliary marginal zone-like stem cell system in the adult eye generates eye photoreceptors and support cells”

- The second argument is that a ciliary opsin, normally found in the brain is also expressed in a very small number of cells that are EdU positive after a short pulse, and are thus likely the progenitor cells described above. The images are difficult to read as expression is very sparse and at low levels with little dots on the picture that are hardly visible. Since only a small subset of the dividing cells express it, their significance is difficult to assess, especially since the ciliary opsin gene is not expressed in the mature eye.

The comments of this and the other reviewers led us discover that we overlooked that the pdf (or jpeg) conversion of images during the upload had an unexpected and unfortunate impact on the representation of in situ HCR results:

***In situ* HCR is a single-molecule detection technique (cf. Choi et al. (2018). Development 145, dev165753. <https://doi.org/10.1242/dev.165753>) that by its nature yields small fluorescent pixels rather than large areas of color staining (as we are used to for**

conventional *in situ* hybridisation). For *c-opsin1*, some of our images had shown individual confocal slices (to allow for the best detection of cellular co-localisation), meaning that the signal is necessarily rather sparse. In turn, pdf (or jpeg, and other graphic compaction modes) apparently systematically reduces such small pixels to reduce the supposedly “noisy” data content of the image.

In order to make our images more robust to data loss during image format conversion, we now included zoom-ins for various fluorescent images in the revised figures (see in the context of *c-opsin1*: Fig. 3H-R, but also more generally see Fig. 1G-K, F.2Q-T). These now show the relevant HCR stainings at strongly magnified scales. We also included dashed lines to indicate the edge of expression domains in Fig. 3H and 3L, as well as arrowheads. We hope that these measures now make the expression clearer to spot. Please also note that the negative controls (run in parallel and are representative for our results) show no sign of such staining.

We respectfully disagree with the comment that only a small subset of cells express *c-opsin1* in the adult eye. Actually, we find it to be a rather clear domain in the adult eye (as e.g. compared to its brain expression): In Fig. 3E (which represents a Z-projected confocal stack), the region highlighted by arrowheads is a clear domain surrounding the eye opening.

As to its significance: Based on our scRNA seq trajectory and EdU analyses we interpret this expression as likely to be transient for a given cell during its differentiation, so it naturally will not be as broadly expressed as the *r-opsin1* photoreceptor molecule.

But the major issue is the claim that a mutant in *c-opsin1* leads to eye reduction and to the loss of these cells. The authors present a good quantification of these effects and it might be indeed true that the eye is smaller and that eye cells are lost. However, and if this reviewer reads the figures and legends well (a very difficult task as the legends very cursorily describe the panels), it appears that many other cells types that do not express *c-opsin* are also decreased very significantly, even more so than eye cells expressing the *c-opsin*: "Contribution of cells from libraries generated from *c-ops1Δ8/Δ8* specimens versus those generated from WT controls to the different clusters differed significantly in five cases, in which four showed a significant underrepresentation". It is very difficult to explain this loss and the easiest interpretation is that the *c-opsin* mutant chromosome has off-target mutation(s) that affect to a minor extent the fitness of the animal, leading to small eyes and fewer cells. Is the animal smaller in general? i.e. the size of the eye should be presented as a ratio to the total size of the animal. It would not be reasonable to request a rescue experiment in the mutant animal, but the interpretation is likely not correct.

The authors should test whether raising *Platynereis* in complete darkness leads to the same results, which would then support the argument that the function of *c-opsin* has a developmental role.

The conclusion from the authors in fact suggests that this is the case: "While such an analysis represents the impact of the mutation over the entire life of the animal and hence may show effects that are also outside of the *c-opsin1* expressing cells themselves, this finding is in line with the observed reduction of eye size in mutants and the resulting hypothesis that *c-opsin1* influences the division rate and/or differentiation of *r-opsin1+* eye cells."

We thank the reviewer for outlining their thoughtful points about the *c-opsin1* mutant data, and whether the conclusions should be balanced in a different way. We have spent significant experimental efforts to better work out the underlying biology. After these revisions we believe that we have found an appropriate way forward that emphasizes conceptual insights as well as caveats. For our answer, we separated what we understood as the two main concerns expressed by the reviewer:

- (a) That the *c-opsin1* mutants might carry unrelated off-target mutations on the same chromosome, leading to reduced fitness, entirely unrelated to *c-opsin1* and light
- (b) That the *c-opsin1* mutation is causative, but has indirect (non-cell-autonomous) effects outside the expression domains of *c-opsin1*, itself. This would weaken the relevance of *c-opsin1* being expressed in the *r-opsin1* trajectory; likewise, the unavailability of conditional knock-out technology or adequate rescue tools in our system (of which the reviewer is kindly aware of) means that it is at present impossible to discern cell-autonomous from non-cell autonomous effects.

Concerning (a), we would like to make the following arguments:

(i) The original *c-opsin1* $\Delta 8$ mutation we capitalize on for our study (reported in Veedin Rajan et al. (2021). Nat Ecol Evol 5, 204–218. <https://doi.org/10.1038/s41559-020-01356-1>) was still generated by TALEN technology, which has a higher site specificity than the conventional gRNA of the Cas9 system. Specifically, *c-opsin1* was targeted by co-injection of two engineered TALE nuclease proteins, each of which was directed against 15 nucleotides of target sequence, coupled to an obligate heterodimer of the FokI nuclease. This design therefore provides a total sequence recognition length (30 nucleotides) exceeding the default 20nt targets of a regular Cas9 system and is hence less likely to generate off-target mutations.

(ii) The mutant has been generated more than 7 years ago and has been continuously propagated, including repeated outcrosses against wild-type animals. Thus, the probability of retaining off-site mutations outside of the targeted chromosome appears negligible from a statistical perspective.

(iii) Systematic assessment of off-target sites for TALENs by the Zhang lab (Sanjana et al. (2012). Nat. Protoc. 7, 171–192) reports a spacer tolerance of 14-20 nucleotides, and a reduced binding of TALENs to targets with more than one mutation per half site. Based on these studies, and capitalizing on the first chromosome-scale genome of *Platynereis*

dumerilii that we have recently generated for the incrossed strain in which the mutants were also created, we have generated a script interrogating the *Platynereis* genome for degenerated forms of the TALEN target sites. The script settings tolerate a 14-20 nucleotide spacer and allow for up to 3 mismatches per half-site. This yields a single potential off-target site, but this would be located at a different chromosome (Chromosome 14) than the one (Chromosome 2) where *c-opsin1* is located.

(iv) We thank the reviewer for their specific constructive suggestion that experiments with animals under different light regimes might help to make the case that eye development is indeed susceptible to light (of a quality matching the *c-opsin1* sensitivity), thus supporting our conclusions. Accordingly, we analyzed if shifting wild-type animals during the growth phase of the eye to complete darkness (“DD regime”) causes any differences in the eye to a regular daily Light-Darkness light regime (“LD regime”). As we show in the revised Fig. 4 (Fig. 4A,B), this is indeed the case: the number of EdU-positive cells in the eyes (normalized by eye size, following another suggestion of the reviewer) of DD regime specimens is significantly lower than the number in LD regime specimens. So light exposure (matching the sensitivity spectrum of the *c-Opsin1* protein) impacts acutely on adult eye development.

See also revised text: “In a first experiment (Figure 4A,B), we transferred premature individuals This indicates that light impacts on proper progenitor proliferation in the eye, and that – indirectly or directly – *c-opsin1* is relevant for this effect to occur.”

(v) Importantly, when we performed the experiments with *c-opsin1* mutant specimen, the difference in EdU+ cells which results under LD and DD light regimes did not differ from each other and the data distribution showed what could be considered as an intermediate to wt LD vs. DD (Fig.4B). This is consistent with *c-opsin1* mediating the light-dependent function in the process. In order to clearly avoid an overstatement, we now also specify in the text that this effect might be directly or indirectly mediated by *c-opsin1*. For details see revised Fig.4A,B and text: “We transferred both wild-type (WT) and *c-ops1Δ8/Δ8* individuals.... *c-opsin1* is relevant for this effect to occur.”

Taken together, all these arguments indicate that *c-opsin1* indeed plays a functional role in late adult eye growth and differentiation, and that its light-sensitive function is relevant for this. They do not, however, address the issues subsumed under (b) (cell-autonomy? Secondary effects of early developmental disturbance?).

Concerning (b), we would however ask to consider the following arguments:

(i) The aforementioned light-dark experiment made clear that light is directly relevant during the time window we focus on (eye growth preceding maturation). Thus, the new experiments support the notion that the observations we make at that stage are not just secondary effects of the preceding developmental stages.

(ii) As laid out in our initial manuscript and pointed out by the reviewer, the scRNAseq experiment we presented for *c-opsin1* mutants and controls exhibits significant changes in

four other clusters (12, 15, 19, 20). In the course of the revision, we successfully characterized three of these (clusters 12, 15, 19) using *in situ* HCR stainings – data that we have now also included in the revised Fig. 1 (panels G-K). From these, it is apparent that cells from clusters 12 and 19 localize to the medial brain. We note that this is close to the second site (beyond the eyes) where we have described *c-opsin1*-positive cells to be present (cf. Fig. 3E,F). While correlational, this makes it at least more plausible that there could be distinct, local effects of the medial brain-specific *c-opsin1* expression and the one that we uncover in the eye.

(iii) We currently do not have the genetic tools for *c-opsin1* conditional knock-outs or tissue-specific rescue. While we still consider a cell-autonomous modulation of the *r-opsin1*-positive photoreceptor lineage the most plausible explanation for the eye phenotype, the mentioned technical limitations prevent us to rigorously verify that the impact of light and *c-opsin1* on the eye is not caused in an indirect way, possibly relating to the medial brain cells that are well positioned to impact on overall physiology of the animal.

We have therefore toned down our mechanistic interpretations, portraying both the impact of light and the requirement of *c-opsin1* as observations that might be linked to the curious *c-opsin1* eye expression domain, but cannot be cell-autonomously tied to this expression. To accommodate this, we also modified the title and abstract, and the respective section in the discussion. We hope that this is the most appropriate way how to adequately reflect both data and concepts.

For further details, see new revised heading and text in discussion section: “Our findings that light impacts on adult eye growth, that *c-opsin1*.... In either scenario, the regulation might help to optimize the worms’ sensory system to the different light conditions they encounter.”

Other points:

- Fig.3 A-D: the co-expression of *r-* and *c-*opsins is difficult to see (D: which cells co-express?) It is surprising that both opsins are expressed in dividing cells. In all other systems, opsins are post-mitotic. If, as argued in the discussion, *c-*opsins are remains of evolution, this would be possible, but expression of *r-*opsins that will induce phototransduction is more surprising.

The reviewer’s remark about the difficulty to validate the co-expression is understandable, given the issue outlined above (single-molecule stainings in pdf/ jpeg converted images). This should now be resolved by the magnified views in which HCR puncta should be well visible (see revised Fig.3H-K).

As to the conceptual question how to interpret the expression of opsins in dividing cells: from an earlier study on comparing eye photoreceptors and related non-cephalic cells (Revilla-i-Domingo et al. (2021). *Elife* 10, e66144. <https://doi.org/10.7554/elife.66144>), we know that *r-opsin1* is among the most highly expressed genes in *Platynereis* eyes (reaching levels of actin in fully differentiated eye cells). This is also reflected by our single-cell data, where *r-opsin1* has among the highest expression out of all genes in the entire *Platynereis*

genome. Thus, we think that *r-opsin1* expression is starting to build up already at an early stage of differentiation, but reaches its maximal (and then functionally relevant) levels only in the partition pB that we consider mature eye photoreceptors. As indicated in Figure 3D, at the time point of co-expression of *c-opsin1* and *r-opsin1*, the levels of both genes are similar (hence the white color in the scheme). It is well possible that during the transient phase of *c-opsin1/r-opsin1* co-expression is just starting to be functional. The latter is consistent with an observation of *r-opsin1::eGFP* transgenic worms in which visible GFP is only present with a slight delay relative to transcript levels. Thus, we avoid a statement on the functionality of *r-opsin1* while it is co-expressed with *c-opsin1*.

As to the functionality of *c-opsin1* in possible neural stem cells, we believe that it is highly interesting (and also supporting our findings) that the closest ortholog of c-opsin1 in mice, encephalopsin/opn3, is expressed in adult neural stem cells of the mammalian dentate gyrus (see graph below). We obtained these data from our collaborator N.Urban (IMBA, Austria) on a different project. They are not yet published, but we would like to mention them to this reviewer to allow for better judgement of the potential significance of our work.

Bulk sequencing data of different neural stem cell populations from the murine dentate gyrus and the absence/presence of different opsin transcripts. Deep Q- deep quiescent adult neural stem cells (aNSCs), shallow Q- shallow quiescent aNSCs, active aNSCs. Mammalian *Opn3*, the closest ortholog to *Pdu-c-opsin1*, is specifically expressed in these aNSCs. (Data: collaboration partner N.Urban, IMBA, Austria)

In conclusion, extensive data and some nice demonstration of eye growth but the title of the paper that c-opsin functions in a ciliary marginal zone-like might not be justified

We thank the reviewer again for his/her thoughtful comments. As outlined, we strongly believe that the extensive additional data included in the revised manuscript serve well to support our major claims – including the functional relevance of *c-opsin1*. At the same time, in recognition of this (and other) reviewers' caution, we have balanced both our presentation and discussion of the *c-opsin1* work, and in this context also adjusted the title, as suggested by the reviewer.

Reviewer #2 (Remarks to the Author):

The paper entitled “A c-opsin functions in a ciliary marginal zone-like stem cell region of an invertebrate camera-type eye” compiles several single cell RNA-Seq datasets performed on the head of the annelid *Platynereis dumerilii* representing a fundamental resource for the *Platynereis* community in the future. The main focus is on the neurogenesis process in brain and eye and the changes in the neuronal stem cell population at different phases of the life cycle. The dataset analysis is very thorough, while the HCR results and imaging processing/quantification has to be significantly improved to support the Authors conclusions.

We thank the reviewer for their overall positive judgement. As we detail below, the reviewer comments led us uncover a general issue about the incompatibility of images from HCR stainings and pdf/ jpeg conversions. We apologize for our unawareness that the provided HCR images lacked the necessary detail in the document that was generated during submission. The additional experiments and images, along with more zoom-in images in the revised work should now make it easier to appreciate the actual data.

Major comments:

1. While the title and the introduction are mainly focused on the eye and the CMZ, a good part of the results and the discussion is more focused on the neurogenesis and the eye seems an area to look at as well as the brain. In the manuscript it is difficult to grasp what was the main question the Authors were trying to answer. I would recommend making the question clearer and why this experimental approach is the best way to answer it. I would also encourage the Authors to help the reader understanding the jump from the c-opsin mutant to the last part of the manuscript. Currently, these 2 sections seem disjunct from each other. Finally, I am wondering if a title about neurogenesis would be more appropriate and would attract a broader readership interested in neuron formation instead of focusing on eyes and CMZ.

We thank the reviewer for raising these questions, which ultimately relate to providing a clearer narrative for the paper. In the revised version, we now more clearly spell out our overarching interest in brain plasticity, and then subsume more precisely how our discovery of (i) a spatial stem cell zone in the eye and (ii) temporal regulation of stem cell activity over maturation contribute to this overall goal, also linking the sections more clearly. We have also adjusted the title to accommodate arguments made by both reviewer 1

and reviewer 2, removing the specific mentioning of *c-opsin1* in favor of incorporating the aspect of brain plasticity. We reason that these adjustments should satisfy the suggestions made by both reviewers. Please also see extensive rephrasing in the introduction: “Whereas the aforementioned molecular cell type comparisons have largely.... similar decline of their stem cell capacities”, as well as many changes in results and discussion (not individually listed here, because of number).

2. SoxB1 and *c-opsin* HCRs are extremely weak, mainly considering the high number of probe pairs used, and currently it is difficult to determine if the cells are really positive. To support their conclusions, the Authors have to show stronger evidence or ordering a new set of probes or searching for markers co-expressed in the same cells. The images provided are not supporting that the Edu positive cells are expressing *c-opsin* (Figure 3L-M).

As alluded to in the beginning of this response, the comments of this reviewer and others have led us to discover that the pdf/ jpeg conversions of images during the upload has had an unexpected and unfortunate impact on the representation of in situ HCR results:

In situ HCR is a single-molecule detection technique (cf. Choi et al. (2018). *Development* 145, dev165753. <https://doi.org/10.1242/dev.165753>) that by its nature yields small fluorescent pixels rather than large areas of color staining (as we are used to for conventional in situ hybridisation). For *c-opsin1*, some of our images showed individual confocal slices (to allow for the best detection of cellular co-localisation), meaning that the signal is necessarily rather sparse. In turn, pdf/jpeg conversion apparently systematically reduced such small pixels to reduce the supposedly “noisy” data content of the image.

We repeated several stainings and imaging to improve the staining intensity and representation quality. We also included an additional marker (see below). In addition, we included zoom-ins for various fluorescent images in the revised figures that should now show the relevant HCR stainings at strongly magnified scales. We also included dashed lines to indicate the edge of expression domains in Fig. 3H and 3L. As to the specific request of this reviewer,

- Zoom-ins were added to the *c-opsin1* stainings in Fig. 3 (panels I,J,K and M,N,O)
- we repeated *soxB1* stainings with a new probe set and now present a more robust staining, again shown with zoom-ins (Fig. 2Q-S)
- Following the advice of the reviewer, we also selected another marker (*notch*) co-expressed with *soxB1* in the eye, and indeed obtain similar stainings (Fig. 2U-W)

We believe that these changes should clarify the raised points.

3. If the Authors can answer the previous point, they will need also to update the models in Figure 2T and 3K since the positivity seems to be localized all around the lens and not only in the

most superficial area like it is currently presented in the cartoons. These models could be misleading that the positivity is a circle closed to the “cornea”, while based on the images showed so far, these cells are localized as a cup or semi-sphere all around the lens. A 3D reconstruction through Imaris or a video of full z-stacks can help elucidating this point. Ideally a quantification of the position of positive cells would be ideal.

We apologize if the representation of the eye was misleading to this reviewer. The orientation of our schemes was a lateral medial section through the eye, aligned with the classical scheme provided by Fischer and Brökelmann (1966), *Zeitschrift Für Zellforschung Und Mikroskopische Anatomie* 1948 71, 217–244. Their Fig. 22 had been adapted and reproduced various times throughout comparative literature of animal eyes (see e.g. Arendt, D., and Wittbrodt, J. (2001). *Philosophical Transactions Royal Soc Lond Ser B Biological Sci* 356, 1545–1563, Arendt et al. (2002). *Development* 129, 1143–1154. Arendt, D. (2003). *Int J Dev Biology* 47, 563–571) and we built on this, but agree that we could have done better to orient the readers.

The misunderstanding probably results from the interpretation that the HCR images in these figures would have been taken as latera cross-sections along the axis of the eye (as the scheme), whereas they are superficial cross sections more towards the surface of the eye, showing part of the circumference there. As the *Platynereis* retina is not a multi-layered structure, but a pseudo-stratified epithelium of support cells and photoreceptors, there is indeed no extra layer close to the lens that could serve as a stem cell layer in the area that the reviewer refers to. But the bona fide stem cells form a ring close to the lens opening.

To help interpreting the way how sections have been taken, and upon suggestion by another reviewer, we now included an overview of the eye as well as the orientation of optical sections in the images, and the plane of the Fischer and Brökelmann scheme used for our own schemes as well (revised Fig. 2E); likewise, the orientation of sections is noted in relevant panels of Figs. 2 and 3. We hope that these measures will help the reader to better appreciate the anatomy of the worm eye and the interpretations drawn from the stainings.

4. In Figure 4, the Authors are focusing on the decrease of cells in mega-cluster 3. Please, discuss also the decrease in 10, c19 and the increase in c20. This seems a very interesting piece of data that is not followed up or mentioned for speculations. A marker could be selected for performing HCR to see where these cells are express. Where else is c-opsin expressed in the head?

We thank the reviewer for highlighting the additional interest that these changes raise. In response to this reviewer’s comments – and a request of reviewer 1 to clarify if or not *c-opsin1* was likely to act directly or indirectly – we have also followed up on additional of the regulated clusters. In the course of the revision, we characterized three of these (clusters 12, 15, 19) using *in situ* HCR stainings. (to clarify, cluster 15 corresponds to the former super-cluster 10 that the reviewer asked for).

The stainings of the respective markers have now been included in the revised Fig. 1 (panels G-K). From these, it is apparent that cells from clusters 12 and 19 localize to the medial brain. We note in the text that this is close to the location of brain-residing *c-opsin1*+ cells, the second site beyond the eye where we describe *c-opsin1*-positive cells to be present (cf. Fig. 3E,F), consistent with previous observations in larvae (ref: Arendt, Tessmar-Raible et al.(2004) doi: 10.1126/science.1099955). While correlational, this makes it at least more plausible that there could be distinct, local effects of the medial brain-specific *c-opsin1* expression and the one that we uncover in the eye. In keeping with the reviewer's request to comment on the other clusters and other sites of *c-opsin1* expression, we have now included these pieces of information in the revised discussion section of the paper. This also helps to balance our statements of how *c-opsin1* is likely to act on these cells. For details see especially text sections:

“To assess how the obtained cell populations distribute in the head and may match known structures or cell types, we next investigated the expression of known markers or some transcripts selected from the clustering cirri, known sensory structures (Figure 1I, J).”

“Our findings that light impacts on adult eye growth, that *c-opsin1* is required..... In either scenario, the regulation might help to optimize the worms' sensory system to the different light conditions they encounter.”

5. The last section of the manuscript is less focused on the eyes and is supported mainly by in silico analysis. The lack of functional data about *foxo* and *prox* in *Platynereis* adult worms together with the HCR on only 2 marker for neurogenesis and only one marker for quiescence does not allow for strong conclusions about the state of the cells. Please, tone down the conclusions or add additional validating data to support the model in 5X.

In Figure 5, at which part of the head and eye are we looking at? It seems in the adults all the cells have both *foxo* and *prox* genes expressed, is that true or is it only a subpopulation of cells?

The reviewer has already in their first comment pointed out that the section on quiescence needs to be better connected to the remaining parts of the manuscript, and we hope that our clarification and the revised text help to point out that the focus of the last section is on the temporal control of signatures, which stands out as another clear result from our data that we felt important to include. Whereas we cannot provide independent functional evidence that genes like *foxo* and *btg3/4* play a role in proliferation control (a task that likely would require conditional gene ablation, which we currently cannot perform in the model system), we have added additional supportive data:

We have validated *btg3/4* as an additional “quiescent” gene expressed in mature animals, also in conjunction with *prox*, that we have also used with *foxo* (panels X,Y,Y' of revised Fig. 5). In these experiments, we also co-detected *btg3/4* and *prox* in cells labeled in a pulse-chase experiment with EdU, in which we pulsed non-reproductive animals with EdU to specifically investigate cells that divided just before the animals enter the reproductive

stage (upon which they won't up-take EdU anymore: see Fig.1). This new dataset also provides direct evidence for our interpretation that the quiescence-like signature affects cells that have recently arisen by cell division.

Taken together with our previous data, we consider this strong evidence that there is a massive change in cellular signatures. As the reviewer supposes, this change is occurring throughout the brain, and as we show in revised Fig. 5X,Y, this includes cells arising from relatively recent cell divisions, including the eyes (also linking back to the eye section of the manuscript).

As to the orientation of images, we have tried to make this clear by providing the overview panels 5L and 5R (dorsal views), and the dashed lines leading to the zoom-in panels. In Fig. 5O and Fig. 5U, we have generated medial cross-sections through the eyes, whereas Fig. 5X,Y are superficial cross-section as defined in the new panel Fig. 2E mentioned above. We hope that this clarifies the orientation and position of images.

Minor comments:

- The abstract mention "modulation of the environment" but the Authors never tested that. Please, replace it with "throughout life cycle".

This section of this abstract has been reworded, and in keeping with this request, the environmental aspect has been removed.

- I tried to look for the reference 4 that is used several times to talk about *Platynereis dumerilii* eye anatomy, but I was not able to find it. I am wondering if there are more recent references that can be used to support that the *Platynereis dumerilii* has a camera-type eyes. Reference 25 calls the *Platynereis dumerilii* eyes as pigment-cup eyes and I could not find many other information about their anatomy.

Reference 4 can be found (German original) here:

<https://link.springer.com/article/10.1007/BF00335748>

We now add this link to the reference list, but are unsure as to whether this will be permitted by the journal.

In any case, and as indicated above, the scheme has been re-used in later reviews like this one: Arendt, D. (2003). Evolution of eyes and photoreceptor cell types. *Int J Dev Biology* 47, 563–571. We have now also included this when introducing the original Fischer & Brökelmann reference, as it may be more widely accessible.

- Many time the Authors are using "brain" dataset instead of "head" dataset (such as in the title of Table S1 and S2; line 41 page 5; panel 2E; line 38 page 7; line 1 page 8; line 1 page 10; etc)

The reviewer is correct, thank you. We have now paid attention to introduce sampling and the cellular landscape as "head" dataset first. After discussing the distribution of the pan-

neuronal signature and neurotransmitter, we then use the term “brain” when more specifically referring to predominantly those neuronal/glia cell type clusters. We consider this an appropriate solution for adequately referring to the non-neuronal cells, but retaining our scientific focus on brain plasticity (also see comment on epidermal cells below).

- The Authors should implement the explanation about how they got the modules and how this helped them manually design the mega-clusters since in the majority of the cases they do not correspond to each other (modules 3, 9, 10, 11, 12 and 13 include many more clusters than the mega-clusters).

Upon the reviewer’s comment, we have extended our description of the clustering.

See text: “To capture changes within cell populations,.... Collectively, this indicates the relevance of the defined cell populations as biologically relevant groups harboring distinct cell types.”

The modules (whose generation is detailed in the Materials and methods) support the clustering, but are not the main source for the super-clusters. Also, to provide more clarity, we have, in the revised version, retained the Arabic numbers exclusively for the actual clusters - which are our central reference - and refer to the super-clusters with small letters. We hope that this makes the distinction between the different organisational units easier. In order to further verify our clustering, we have also added additional investigated the spatial localisation of cluster marker transcripts (by looking how they match already published patterns in adult heads and by additional *in situ* HCRs)- see text: “To assess how the obtained cell populations distribute in the head and may match known structures or cell types,..... These populations harbor distinct cell types of known and novel identity that we expect to encode the worm’s sensory and information processing, its complex time-keeping abilities, and central output control.”

- The scRNA-Seq has been run on the entire head but it looks like there are no epithelial cells, gland cells, fibroblasts, etc. Which other tissues other than brain and eyes are present in the head of *Platynereis* and why they are not represented in the dataset?

We understand that our initial nomenclature was insufficiently clear and thank the reviewer for pointing this out. As indicated above, we have now more explicitly pointed out that the sampling concerned entire heads, not just brains, even though our scientific interest in neuronal and brain plasticity has made us focus primarily on distinct neuronal clusters. In response to the reviewer’s comment, we have also more clearly pointed out the presence of an “epidermal” super-cluster (super-cluster a) and refer to the established genes expressed in this super-cluster that support this assignment.

- Table has more clusters with rhodopsins and opsins than the one included in the mega-cluster 3. Why are those clusters not considered part of the eyes? Did the Authors run HCR on these genes to verify the location of these cells?

As alluded to above, we have chosen to focus on the eye, as we felt that the finding of the growth zone was conceptually highly interesting, and allowed us to uncover an unexpected expression and function of *c-opsin1*.

The presence of photosensitive proteins (opsins and cryptochromes) outside the eyes of *Platynereis* is very well established and documented, including stainings, e.g. doi: 10.1038/s41467-022-32562-z; doi: 10.7554/eLife.66144; doi: 10.1038/s41559-020-01356-1; doi: 10.7554/eLife.36440; doi: 10.1186/s12915-018-0505-8; doi: 10.1016/j.cell.2014.07.042; doi: 10.1016/j.cell.2007.04.041; doi: 10.1126/science.1099955

The expression patterns of these and other genes on the scRNAseq map is included in Supplementary Fig. S7, providing an overview over diverse photoreceptors. The results fit very well with our expectations and previous analyses by us and others in *Platynereis* and the field of non-visual photoreceptor cells.

More generally, *Platynereis* is a highly interesting model system for the broader topic of light biology and biological timing, and there are indeed fascinating aspects about non-ocular light receptors in the system, but this is a topic on its own. We would like to refer the reviewer to further published overview work (cf. Mat et al. (2024). *Physiology* 39, 30–43, Wulf et al. (2025). *Zool. Sci.* 42., and references therein).

- Please, add more info about *fut10* and why it was selected as marker.

Fut10 was one of the distinct markers we identified based on its expression in the scRNAseq data and verified by HCR, without specific reference to its presumptive enzymatic function. We added some additional information, for details see “A subset of partition pA showed particular enrichment for *fut10*... in line with a possible role in pigment/support cell function.”

- In the eye subset there are cells expressing *pcna* but those were not analyzed or discussed in the generation of the model. It could be that *pcna* positive cells are those giving rise to the support cells and *c-opsin* positive cells those giving rise to the PE.

We thank the reviewer for this interesting suggestion. *pcna* has been difficult to visualize by HCR for reasons unclear to us. As furthermore gene-/cell-specific lineage tracing technology is not yet available in our system, we are currently technically limited to follow up on this interesting possibility.

- In the longer Edu pulse with chase experiments, together with looking at the potential migration of the cells, a co-labeling with *r-opsin* or *fut10* could support the claims about the presence of a population of stem cells that divide and then migrate to differentiate in the retinal cell types.

As we now show that (a) EdU+ cells in the eye co-label for *c-opsin1*; (b) that *c-opsin1* and *r-opsin1* are also co-expressed in eye cells; (c) that the number of eye cells is increasing; and (d) as we know from previous work that there is no detectable cell death in the eyes of

untreated wild-type worms (doi: 10.1371/journal.pone.0075811), we believe that this is very strong evidence that the EdU+ eye cells will also differentiate to functional cells in the eye. Given all the other additional experiments we performed in the course of the revisions, we did not prioritize this particular experiment.

- It would be helpful to highlight in the discussion more evolutionary aspects of this study and comparison with what is known in other animals other than mammals and vertebrates.

We thank the reviewer for their interest in evolutionary aspects, which is an area to which we have also contributed in prior work, and to which we also contribute at various sections of our current discussion (also by referencing to cephalopods and onychophorans). Given the restrictions of the article format, however, we are limited in how much we can expand our comparisons to additional models. In addition, we would like to note that for the most meaningful comparisons, we would require data of similar qualities/type also in the compared species. In the mentioned cases (onychophorans and cephalopods), we already faced the problem that we are comparing our single-cell analyses with qPCR data on eye tissue (onychophorans; Eriksson et al. BMC Evol. Biol. 13, 186.) and regular in situ stainings (Napoli et al. Curr. Biol. 32, 5045-5056.e3.). But we agree that the evolutionary aspects are certainly worth further discussing, e.g. in the context of a follow up review article (or similar).

- Please, always add “in silico” in front to aNSC since this is how these cells are defined, and no functional validation has been provided.

We understand that the reviewer would like us to be more explicit about the fact that the neural stem cell state has not been validated by an experiment like clonal lineage tracing experiment. In the revised version, when first introducing the idea, we now clearly refer to the cells as “bioinformatically defined eye adult neural stem/progenitor cells (eye aNSCs/PCs)”. However, we would also like to argue that the follow-up experiments, including the short-term EdU labeling and pulse-chase experiments, as well as the validated expression of the stem cell marker *soxB1* in these cells, goes significantly beyond mere in silico work. We therefore prefer the term “tentative aNSCs” over the label “in silico aNSCs” and have reworded the text accordingly.

Figure 1:

- It would be helpful to bring in the main Figure the Suppl panel S9D. The use of the bubbles could be confusing since in each clusters we can find a few cells whose identity has been assigned to a different cluster, but included in the “wrong” bubble. I would recommend to remove the bubbles and use color code or numbering.

In line with these comments, we have, in the revised version, re-arranged the clustering panels, and placed more emphasis on the cluster identifiers, choosing small letters and a different style to highlight the super-clusters. We believe that this now more adequately describes the different organizational layers of the data.

- Panel 1L has to be replaced with the S2C since that is the specific Edu control that was run side by side and it would provide the proper comparison. Also, are the genes showed in 1L and S2C manually selected or are they all the differentially expressed genes? Please, clarify this point in the legend and provide a full list of the differentially expressed genes.

In keeping with the reviewer's request, we have included both analyses side by side in the revised figures. However, we have placed the entire set to the supplement to make space for the additional validations that are now included in Figure 1.

We had previously already mentioned all details on the selection of the plotted transcripts and differentially expressed genes in the methods section (under "Differential expression analysis"), including the mentioning of Supplementary table 4, which already included a list of all differentially expressed genes. However, we understand that this can easily be overlooked and now also added a reference to this in the caption of the respective panels.

A brief explanation to the two different comparisons: the staging of the two different comparison groups is the same. When we refer to "timing", this means "lunar timing", which refers to another focus of the lab that is not directly related to the current manuscript. In the context of these chronobiological analyses we have noticed that transcripts related to cell cycle can be quite strongly regulated in the head, depending on lunar time. In the context of the comparison to the transcriptome of the sorted EdU+ cells this could mean that specific enrichments are only visible during certain phases of the lunar cycle. We reason that an explicit discussion of this would go far beyond the focus of the current manuscript, but in order to be complete, we included both analyses side by side.

Figure 2:

- In panel 2O seems that the migration is happening only on one side of the eye. Is this true? Please, provide more representative images and Z-stack and provide the specific orientation of the eye.

The proposed arched continuous stripes are visible on the other sides of the eye as well, and are a consistent feature in the performed pulse-chase experiments. The respective panel has been edited to make this point clearer. Moreover, we hope that the clarification of the different orientations in Fig. 2E now helps to make the orientation of the eye clearer in the respective panel.

- The nuclei of 2H are very different in terms of number and localization than 2P as well as between 3H and 3L. Having more uniform images would help the reader.

The magnifications of 2H vs. 2L and 3H vs. 3L in the original manuscript were different, and we would like to refer the reviewer to the respective scale bars that were already visible in the images.

We use different scales, depending on which features are in focus and also to provide different perspectives. We ensured that scale bars are always present.

Figure 3:

- Why the outlined dots in 3D have different colors?

The revised legend indicates now more clearly that the inset in panel D defines the color code of the digital panels.

- Could the Authors explain the difference in intensity between 3E and 3H? Although one is a max projection and the other is a cross section, the signal is so bright in 3E that it seems that there might be cross-sections with more intense signal.

Indeed, the differences are in the thickness of the samples (Z-projected stack vs. single confocal slice). We have aimed at providing thicknesses for all panels provided to make this difference more comprehensive.

Figure 5:

- Is a list in 5c generated by the Authors or was it previously published? Please, add more information.

The lists from mammals is a collection from published literature, that we had referred to in the text (previously references 15,55,56). We now added these references to the caption of Fig.5, too. The details of the Go-term analyses for the worms were already represented in tables S8,S9. We now also added this information to the caption of Fig.5C.

Mat and Methods:

- Line 26 page 14, there is a broken sentence

By the provided position, this appears to refer to the last sentence of the discussion. This sentence has been reworded.

- Line 9 page 17, there is a "(reference)"

In this section, "references" refers to the UMAP set onto which the new data were projected, so "reference" is not for a missing literature reference

- List of used Ab is missing

The antibodies were added to the methods, and are also displayed in the NatComms methodological reporting table.

- Table overview stops at Table S8, please add S9-11

Done

Reviewer #3 (Remarks to the Author):

In this manuscript, Nadja Milivojev et al. investigate proliferation and growth in the retina of the bristleworm, *Platynereis dummerilii*. They identify cells in the cell cycle using Edu and investigate retinal cell function through single cell sequence analysis in order to identify a retinal stem cell population. In addition, they use a c-opsin mutant to investigate the role of c-opsin in the growth of the adult eye.

I think that understanding cell cycle control, the process of neurogenesis and regulation of nervous system growth across species is an exciting and impactful question, and not well understood. I believe that the field of evo-devo has much to be gained from leveraging transgenic and mutant lines in non-models to better understand fundamental questions in nervous system evolution. I believe the adult *Platynereis* visual system is an excellent context to understand visual system growth and the impact of life history and sexual maturity on growth and quiescence. I also find the proposed conclusions of the paper to be plausible. It is likely to me that there is a proliferative zone, potentially at the periphery of the retina, that contributes to growth prior to sexual maturity.

We thank the reviewer for providing these positive comments and also for expressing their appreciation for research in “non-models” and its significance.

However, there are significant flaws in the manuscript as it exists that make it difficult to interpret the authors findings and therefore does not do the ambitious and exciting intellectual goals of the manuscript justice. I therefore cannot support the claims made in the paper as presented.

Summary of major issues:

There is a need for a **basic description and understanding of growth and proliferation** in the *Platynereis* retina to make the conclusions the authors are making here. It is impossible to define a *Platynereis* CMZ-like zone without doing a **first principles description of growth and changes of the bristleworm eye** at each of the stages of interest.

We appreciate the author’s interest in a more comprehensive description of the eye growth and changes. To put our work in context: At present, Fischer and Brökelmann’s detailed analysis in from 1966, as well as a follow-up by Rhode (1992) – both cited in our study – remain authorities on the adult eye structure, but also on the notion that cell divisions are too rare to be systematically assessed.

Fischer & Brökelmann state: “Das Wachstum des juvenilen zum ausgewachsenen Auge stellen wir uns folgendermaßen vor: Neue Augenzellen lagern sich nur am Rande des

Augenbechers an, und die Füllmasse-Ausläufer der neuen Stützzellen verkleben dort mit der schon vorhandenen Füllmasse. Der zwiebelschalenartige Aufbau der Füllmasse gerade am Pupillenrand stützt diese Auffassung. Zellteilungen der Augenzellen haben wir noch nicht beobachtet.”

(“We imagine growth of the juvenile eye into the eye of a mature specimen to occur as follows: new eye cells are added only at the edge of the eye cup. The extensions of the support cells that bear the filling mass aggregate with the existing filling mass. The onion-shaped setup of the filling mass at the very end of the pupil supports this assumption. We have not yet observed any cell division of eye cells.”).

Rhode has performed cell counts across various stages, confirming the increase of cells to 2300 (posterior eye) and 700 (anterior eye) in immature animals, emphasizing that these numbers are equivalent to the ones of Fischer & Brökelmann for mature eyes, in line with the theory that proliferation does not play a role in maturation.

Basic molecular developmental analyses of earlier stages have been addressed by Arendt et al 2002 (Development 129, 1143–1154; <https://doi.org/10.1242/dev.129.5.1143>) and have been put in wider context in annelids by Purschke et al, 2006 (Arthropod Struct Dev 35, 211–230; <https://doi.org/10.1016/j.asd.2006.07.005>). Additional descriptions on initial cell composition and wiring are included in Backfisch et al. (2013). PNAS 110, 193–198; <https://doi.org/10.1073/pnas.1209657109>.

These sources

(i) underline that cell divisions in the immature adults are very rare events; this is consistent with our observation that there are rarely EdU-positive cells in eyes of immature animals labeled for 16h (see Fig. 1L, Supplementary Fig. 2A) and also

(ii) make clear why we were intrigued to uncover a brief period of proliferative boost in the premature animals (see Fig. 1M arrowheads in posterior eyes, Supplementary Fig. 2B), where Fischer and Brökelmann have postulated the increase in eye size to rely merely on size increase of photoreceptors (which is an effect they quantified carefully).

Given this background, we hope this reviewer agrees that it is unlikely we can realistically establish additional “first principles” of growth and change that would go beyond the existing literature – except for the very stage that actually we focus on – unless we had ways to image the same animal at cellular resolution for weeks, which at present is technically not possible.

It is unclear if the **larger eyes actually have more cells or just bigger cells**, as previously described.

As outlined above, Fischer and Brökelmann carefully quantified the enlargement of cells that they reported, and our data were never meant to contradict this claim – in fact, we refer to the enlargement of inner eye mass in Supplementary Fig. 4A-E. However, as also shown in Supplementary Fig. 4I, cell counts in super-cluster c greatly increase from the immature stages to mature stages, much in line with the massive burst in EdU signal we observe. We have reworded a passage in the section describing these data to increase clarity.

Is there cell death?

A prior study establishing conditional cell ablation by the expression of nitroreductase has tested for cell death in *Platynereis* heads and eyes using TUNEL staining and did not find evidence for apoptosis in untreated controls (Veedin-Rajan et al. (2013). PLoS One 8, e75811;Fig. 3).

In addition, and consistent with these previous data, when discussing our bioinformatic analyses on immature/premature and mature stages, we have indicated that neither molecular markers nor GO terms associated with senescence nor those associated with cell death were enriched in the cells from reproductive (or non-reproductive) animal heads, much in contrast to the molecular markers up-regulated in quiescence (Fig.5B,C).

Further consistent with these above notions, the classical studies on the eye cited above do not report cell death.

Where are the phosphohistone H3 or PCNA positive cells? EdU positive cells found in the retina do not necessarily originate in the retina.

Whereas we understand the reviewer's point that nominally, EdU detection could mark cells that are already the offspring of stem cells located elsewhere, we consider an origin of stem cells outside of the retina highly unlikely for a combination of arguments:

- (i) The shortest EdU pulse used in the presented data is 4 hours and clearly labels cells in the eyes. In additional work (not included in the study), we have systematically analyzed the number of EdU positive cells in heads after different pulse periods. These measurements allowed us to estimate the cell cycle length in the brain to be longer than 12h and equal/shorter than 24 hours, based on the doubling of EdU+ cells at least every ~24hrs (see figure below). Thus, a 4 hour pulse is highly unlikely to stain just the offspring of stem cells, but should include the stem cells in the eye themselves.

Labeling of EdU+ cells after different pulse periods with/without chase periods indicates that a doubling of cells requires more than 12hrs and equal/less than 24hrs, indicative of cell cycle length.

- (ii) Consistent with this, panels 3L-N (4h pulse) show mostly single cells, clearly positioned within the eye, and not outside the eye.
- (iii) When using a longer chase (e.g. Fig. 2P), labeled cells are often found in radial lines, fully consistent with additional cell divisions occurring from the initially pulse-labeled cells. Most plausibly, this indicates that the cells in the eye observed upon pulse labeling continue to divide, again arguing that EdU-positive cells in the eye are locally produced, and not postmitotic cells that migrate into the organ.
- (iv) While Fischer and Brökelmann generally did not observe dividing cells, they also did not report on any evidence for migration of cells from outside the eye in to the eye, as would need to be postulated if the source was indeed somewhere else.
- (v) *soxB1* is broadly regarded a very early marker of neuronal lineages, and we can show it to be expressed within the eye, consistent with a local stem cell population. Also, the newly obtained stainings for *notch* (another marker for early stem cells) – co-expressed with *soxB1* in our bona fide stem cell population – stains in the same region (cf. Fig. 2Q-T).

As to the specific request to perform HCR analyses for *pcna* and anti-pH3: Unfortunately, *pcna* stainings have so far not yielded consistent results in our hands, as also mentioned to reviewer #2. This is likely due to technical reasons that are, however, unclear to us at present. We assume it could be due to interindividual sequence polymorphisms.

Whereas we have performed anti-pH3 stainings in the context of our work on regeneration, it is specific to the mitotic phase of the cell cycle and thus only stains a subset of cells positive for EdU. As we already find EdU cells in the eye, and know that they are locally dividing, we do not see how this could provide a stronger argument in favor of the specific source of these cells.

The manuscript has **no standard way of collecting the data across time points** or across conditions to be able to make comparisons.

We are rather uncertain what to take from this general statement. Obviously, the way how to compare data across time points will differ for the type of data acquired, but throughout our study, we have made all reasonable efforts to make data comparable, and this is spelled out in the materials and methods. Examples include:

– scRNAseq data: As described, datasets were brought to the same number of cells by downsampling to make the datasets directly comparable across time points.

– **eye size comparisons:** As described, eye sizes were normalized to the head area of the respective specimens; again, this served to make the data more directly comparable across time points

– **comparison of proliferative cells in the eyes between wild-type animals and c-opsin1 mutants:** Based on input of one of the reviewers, the comparison of wild-type and mutant specimens now relates the number of cells to the area of each eye, which appears to us the most rigorous way of assessing density of proliferative cells irrespective of any slight variation of eye size.

So, we politely disagree with the generalized statement that we have no standard way of collecting data across time points and conditions. We have addressed any of the specific issues (such as the question about the thickness of confocal sections) below.

Are images always going to be taken from **central sections**? While some images appear central, many of the images do not. I think the authors are **taking superficial or optical sections** of their samples. Unfortunately, this makes the data **difficult if not impossible to interpret as presented**.

What are the **axes of this organ and the axes of the included images**?

We thank the reviewer for being attentive to possible issues arising from misunderstandings on the orientation of optical sections. In response to the reviewer's comment, we have now started our description of the eye-related data with an additional panel (Fig. 2E) that clearly shows both the axes and the orientation of the different optical sections used in the study. Moreover, where needed, the orientation is also indicated in panels and text. Together, we believe that this helps to make the stainings and their interpretation more intelligible to the readers.

A more in depth survey of growth using EdU or phosphohistone H3 at different time points is necessary.

As to phosphohistone H3, please see our reply above. As to the in-depth survey of growth, please see our above at the beginning of the section on the question of “first principles”. As outlined above, we build on extensive, systematic analyses and aim to address specific open questions. We would argue that the EdU stainings provided at the diagnostic stages shown in Fig. 1L-O, surveys key steps in eye growth. The images have been inverted now to enhance clarity, and are accompanied by a statistically solid quantification (Fig. 1P). Un-inverted images and negative controls are provided as reference in Supplementary Figure 2A-H

In addition, many of the HCR stains of soxB1, c-opsin, are not convincing as shown.

We fully agree with the reviewer on this point and apologize. The comments of this reviewer and others have led us to discover that the pdf/jpeg conversion of images during the upload has had an unexpected and unfortunate impact on the representation of in situ HCR results:

In situ HCR is a single-molecule detection technique (cf. Choi et al. (2018). *Development* 145, dev165753) that by its nature yields small fluorescent pixels rather than large areas of color staining (as we are used to for conventional in situ hybridisation). For *c-opsin1*, some of our images (Fig. 3H-O) showed individual confocal slices (to allow for the best detection of cellular co-localisation), meaning that the signal is necessarily rather sparse. In turn, pdf/jpeg apparently systematically reduced such small pixels to reduce the supposedly “noisy” data content of the image.

In order to make our images more robust to data loss during pdf/jpeg conversion, we repeated several HCRs, designed new probe sets (e.g. for *soxB1*), included additional markers (e.g. *notch*, *btg3/4*) and included zoom-ins for various fluorescent images in the revised figures that should now show the relevant HCR stainings at strongly magnified scales. We also included dotted lines to indicate the edge of expression domains. All these measures should address the issue raised by the reviewer on the expression of these genes.

Specific Comments:

Line 47: “In the eyes of vertebrates, growth is enabled by a dedicated stem cell area, termed the ciliary marginal zone (CMZ), from which new neurons and pigment cells differentiate”

Please be mindful of your summary about the CMZ. Framing the CMZ as a universal stem cell population required for eye growth in vertebrates is not accurate. CMZ associated growth is primary to fish and amphibians. In mammals, almost all growth is a result of embryonic neural progenitor proliferation and then tissue expansion. The amniote CMZ has limited contributions to growth after juvenile stages.

We thank the reviewer for emphasizing the need to relate this statement more clearly to amphibian and teleost fishes (which were the groups introduced in the preceding sentence). We have corrected the sentence in question now to more clearly point this fact out. See text: “In the eyes of those vertebrates exhibiting life-long growth, a dedicated stem cell area, termed the ciliary marginal zone (CMZ) is the source of new neurons and pigment cells”

Line 109:

Please do not use the word *bona fide* unless you define the contextual meaning. It is not clear what a *bona fide* stem cell system or a *bona fide* glial cell (line 161) is. Glial cells in particular are not well defined by markers and there is not functional observations included here.

We had opted for using the Latin term *bona fide* (“in good faith”) to acknowledge that we cannot experimentally prove (in the framework of a single research paper) certain conceptual implications that we – nonetheless – believe to be of interest and scholarly relevance; but also to express that we are aware of this limitation. In that sense, we see our use of the term more of an expression of faithfulness to a general scientific principle, where we credit our awareness of limitations.

To exemplify this for the case of the “*bona fide* stem cells” we propose: While we cannot, with the currently available tools, follow individual cells through repeated cycles of cell division to unambiguously show that they ultimately yield differentiated neurons, we present experimental results like the EdU pulse / pulse & chase experiments that are in perfect alignment with the presented concept. If the reviewer preferred the term “tentative”, or “presumptive”, we are open to replace this systematically – and we have already done so in parts of the discussion. Ultimately, the question will be if or not readers will appreciate if we enrich the observations we make with a conceptual framework that is more comprehensive to the reader than the naked data alone.

We would argue that such an approach is useful, and that the use of “*bona fide*” actually also serves as a label for concepts that should still be put to experimental test. In the field of stem cells, there are good examples how this approach has been fruitful to formulate concepts, while making clear where experimental evidence was still required. One classical example is the concept of stem cells for formation of mammalian cartilage and bone. The concept of “mesenchymal stem cells” was popularized by Caplan in 1991 (Caplan, A.I. (1991). *J. Orthop. Res.* 9, 641–650), but due to mismatches between *in vivo* and *in vitro* experiments, it remained unclear if such cells really constituted a uniform stem cell pool, or were rather a heterogeneous pool of progenitors. This controversy was still unresolved more than 25 years later (cf. Sipp et al. (2018). *Nature* 561, 455–457; Caplan, A.I. (2019). *Tissue Eng. Part B: Rev.* 25, 291–293), and only recent single-cell RNAseq analyses have helped to distinguish stem cells from stromal cells in this system (Yan et al. (2025). *Heliyon* 11, e42311). Throughout these decades, the term “*bona fide* stem cells” was used to circumscribe both the concept and its limitations (see e.g. Seruya et al. (2004). *Cell Transplant.* 13, 93–101).

As to glial cells, we do acknowledge the reviewer’s argument that these cells, while being discussed to be likely conserved, are notoriously more difficult to compare between invertebrates and vertebrates on the molecular level. For instance, the reviewer has more specifically commented below on the neurofilament/nf gene that our phylogeny demonstrates to be a one-to-many ortholog not only to gfap, but also other filament genes. By contrast, others have used anti-GFAP immunoreactivity to suggest an equivalence between annelid and vertebrate glial cells (Csoknya et al. (2012). *Acta Biol. Hung.* 63, 114–128), but this does only little to establish shared origin, as cross-reactivity is more often a confounding factor than a hint on homology or even orthology.

In the revised version, we have therefore toned down our comparison of nf and gfap, as well as the claims on glial cells, to match the reviewer’s caution. We have also used the term “*bona fide*” more rarely and contextualized our first reference to the tentative eye stem cells as being based on bioinformatic data.

We would reason that this strikes the right balance between the representation of primary data and their conceptual interpretation.

Line 153: The authors don't make a great case that the variation they observed in the merged dataset isn't batch effects versus standard integration pipelines that show a more homogenous integration of the datasets. They mention that the differences reflect differences in a bulk transcriptome. It would be helpful to include examples of these comparisons in a more obvious fashion.

We thank the reviewer for raising this technical point. This had been more prominent in the first drafts of our study, but then shrunk to a very compact format to give more room to the biological insight. Briefly, after significant benchmarking in our lab and another lab (U. Technau, Uni Vienna, Austria) working on time-resolved developmental transcriptome data, we concluded that the phenomenon of batch effects – and the need to counter it – is far more relevant in settings where datasets are genuinely very similar, but that the algorithms do no good service in biological time series, where they tend to eliminate biological signal. In a recent paper (Stockinger, Adelman et al., Nat. Commun. 15, 9882, referenced in our study), we had included biological samples for the same condition taken almost a year after the initial sampling, and found these biological replicates to align very well with their counterparts in the merge approach, while at the same time reproducing biologically relevant and validated changes compared to the other biological conditions that the Seurat clustering would not reveal. It may well be that our conditions particularly benefit from the ACME approach (García-Castro et al. (2021). Genome Biol. 22, 89. <https://doi.org/10.1186/s13059-021-02302-5>), in which tissues are instantly fixed and dissociated. This may prevent distortions in gene expression that are unavoidable in settings where tissues are enzymatically dissociated and cells are sorted alive, allowing for stress signatures and other variations to occur. As alluded to, we have cut down all of this background in favor of a clear biological narrative, and believe that the technical aspects will be more suitable for a separate, tool-oriented publication.

In reading the reviewer's comment, we do realize, however, that the arguments in favor of the merge approach are now not sufficiently clear, so we have taken the following measures:

- We cite both of the mentioned publications with explicit reference to the arguments of robustness and sensitivity, more clearly motivating why we chose them.
- In line with the reviewer's suggestion, we have compiled a module of genes previously found to be differentially expressed between immature/premature and mature stages (see Schenk, Bannister et al. (2019). Elife 8, e41556) and mapped it on the UMAP representation. This is now included as panel C of the revised Fig. 1, and clearly confirms the inherent "polarization" of some of the superclusters we obtain and that would not be well visible in the Seurat approach.
- In addition to this bioinformatically defined module, we explicitly name and reference two specific genes (*qpeptin*, *fabp*) that we have experimentally validated to be differentially expressed between immature/premature and

mature specimens. These very clearly fit the overall assignment of the module.

Line 185: "Ki-67 orthologs have not yet been reported in other invertebrates"

Ki-67 has been previously identified in other invertebrates

- Lee, Chan-Jun, Hae-Youn Lee, Yun-Sang Yu, Kyoung-Bin Ryu, Hyerim Lee, Kyunghwan Kim, Song Yub Shin, Young-Chun Gil, and Sung-Jin Cho. "Brain compartmentalization based on transcriptome analyses and its gene expression in Octopus minor." *Brain Structure and Function* 228, no. 5 (2023): 1283-1294.

- Chuang, Po-Shun, Kota Ishikawa, and Satoshi Mitarai. "Morphological and genetic recovery of coral polyps after bail-out." *Frontiers in Marine Science* 8 (2021): 609287.

We thank the reviewer for pointing out these studies. As far as we can see, the Lee et al. study uses immunohistochemistry with a commercially available anti-Ki67 antiserum. According to the manufacturer (Abcam, <https://www.abcam.com>), the antiserum used in this study (ab15580) is a polyclonal serum against an undisclosed immunogen (proprietary information) that has only been validated in mouse and human. This case can therefore be compared to the case of anti-GFAP immunoreactivity mentioned above, which – as the reviewer will likely agree – is poor argument for conservation.

The Chuang et al. paper performs qRT-PCR, but does not provide analysis of the respective gene sequence (the provided identifier – 51301_c2_g1_i1 – cannot be correlated with any public sequence).

So, as it stands, none of these studies actually compares to the molecular phylogeny and domain analysis we provide in our study, which is not only considered state-of-the-art, but also a required level of detail, to determine homology relationships.

Line 182: I would hesitate referencing GFAP over any other vertebrate protein in the tree to understand function here. There is literature on invertebrate and lophotrochozoan intermediate filaments that would be a better reference.

As already outlined above in the section discussing glial cells, the reviewer is correct in pointing out that the molecular phylogeny argues for the identified gene to be a one-to-many orthologue with a large group of vertebrate filament proteins. Thus, we have toned down the vertebrate comparison for this case.

Line 218: In addition to identifying potentially proliferative cells in the retina, Fischer and Bokelmann also hypothesized the growth of the eye prior during sexual maturity was not a result of "significant proliferation" because the cells become enlarged. However, as the authors note, there are clearly EdU positive cells in the retina after a 16 hour pulse. Is there cell death? The authors need to provide a description of eye enlargement during these adult stages to better understand

proliferation versus cell enlargement. The cross sections in S4A-E are helpful but the manuscript needs more complete observations of the process of growth/enlargement to make accurate hypotheses.

The question on cell death has been raised above (“Is there cell death?”), and we refer to our answers above on this question. We also would like to reiterate that our work was not meant to question the detailed account on the elongation of photoreceptor outer segments by Fischer and Brökelmann. In the specific context of this question, while Fischer and Brökelmann apparently missed the specific time frame in which EdU+ cells are generated and cell numbers factually increase (see Supplementary Fig. 4I), there is no reason to assume that they would equally have missed dying cells in the retina. Neither our functional annotation of expressed genes nor prior work on apoptosis (Veedin Rajan et al, 2013 doi: 10.1371/journal.pone.0075811) has picked up evidence for cell death in the normal wild-type eye.

As the reviewer’s comment suggests that our original text might have wrongly suggested that we would deny the growth model of Fischer and Brökelmann, we have now added clarifying text both in the results and discussion to mention that model and address this point. Likewise, we explicitly state in the discussion that the CMZ-like zone may also hold for earlier stages, but currently remains untestable there, as cell divisions are very rare, and hence patterns will likely not be as conclusive as at the stage we focus on.

In addition, I would soften the language that there is “continuous presence of neurogenic cell populations in this organ.” EdU positive cells after a 16 hour pulse does not confirm that cells are derived within the organ.

The sentence that the reviewer refers to was a guiding argument that leads over from the gene profiles enriched in EdU-positive cells to the focused experiments on the eye. We acknowledge that at this stage of the manuscript, the argument in favor of neurogenesis in the eye is not yet as strong as later. In keeping with the reviewer’s suggestion, we have toned down the argument.

See amended text: “While we confirmed a marked increase in glass body/lens diameter in reproductive worms (Supplementary Fig. 4A-E), our EdU+ single-cell analyses were consistent with the idea that neurogenic cell populations were also present in the eye (Supplementary Fig. 2 I, J).”

Line 256: In the process of Monocle3 pseudotime analysis, the user chooses the root of the pseudotime analysis. The authors say in the methods that they chose their root based off expression of genes known to be involved in neurogenesis and proliferation. This is reasonable but I would be careful about the wording of this sentence. The authors chose the origin, Monocle3 only showed the trajectory. Otherwise, the use of Monocle3 doesn’t tell you anything new.

The reviewer is right in pointing out that Monocle3 cannot determine the origin of trajectories, and we have corrected the respective formulations. Please see amended text:

”Bioinformatic pseudotime analysis by Monocle3⁶² is consistent with the notion that these cells are the origin of a differentiation trajectory leading to *r-opsin1*⁺ eye photoreceptors (partition pB) and *fut10*⁺ eye support cells (partition pA1) (Figure 2N, Supplementary Fig. 6G).”

We’d like to note that the trajectory is consistent with the marker analysis in the eye and the association of markers in the predicted stem cell population with EdU labeling.

Line 270-276: It would be helpful to know the stage or age of the animals in these experiments. They look younger than the eyes shown in the *fut10* HCR data.

We kindly refer the reviewer to the methods section that specifies that unless otherwise stated, all animals used for staining of non-reproductive states were premature, and also details the segment numbers of these worms. The differences in apparent size that the reviewer refers to likely result from the different scale used in the panels (please see scale bars) and the fact that the *r-opsin1* / *c-opsin1* staining is in superficial cross-sections (individual slice), whereas the EdU incorporation is displayed as a Z projection. With the additional guides we introduced for the orientation of the images (see Fig.2E), this issue is hopefully resolved.

Line 273-275: I’m not quite sure what the authors mean here.

- “Following a long chase period, the number of EdU+ cells in the innermost ring multiplied, and we observed trails of EdU+ cells arching outwards (arrow in Figure 2O), reminiscent of the spreading of fish retinal clones along arched continuous stripes”. Are these images central sections of the worm retina? If so, then there are EdU positive cells everywhere, not just the edges. If this is a superficial image or an angled section of the retina (not a central cross section) , including a larger portion of the peripheral retina and anterior of the eye, then maybe the distribution of cells around the lens makes more sense. However, if the authors need to provide better orientation if this is the case. In addition, a superficial survey would not account for proliferation in the central retina if it exists. It is possible that there are clones emerging from the periphery of the *Platynereis* retina, similar to what is observed in the CMZ, however this data does not show this or refute this as presented. If you are not using central sections, **please include a diagram of where the section is in the eye.**

As outlined above, we have, in response to the feedback of this reviewer, and also the other reviewers, introduced overviews of the orientation of different optical sections (Fig. 2E), and also use more consistent labels and language to refer to these. In the panel in question (now Fig. 2P), we now also highlight additional lines at the other side of the glass body, a change that also accommodates a suggestion by reviewer 2. Comparisons of Fig. 2P with Fig. 2O (pulse label only) clearly suggest further proliferation of cells in the eye, making the origin of cells outside of the eye even less likely.

Line 280: “In summary, these data support the presence of a stem cell

zone at the edge of the retina abutting the glass body/lens, giving rise to new eye photoreceptors and eye support cells.”

- I don't think the data support this as presented.
- I would recommend changing some of the timing of these Edu experiments and doing a time course experiment. Immunohistochemistry for phosphohistone H3 or HCR for PCNA would help.

The reviewer here summarizes the points raised above. By addressing the individual points raised before, we hope we have provided sufficient clarification why the model that we present is plausible and consistent with all available data.

Line 300: “The position of such cells adjacent to the glass body/lens, where the aforementioned results revealed the CMZ-like stem cell zone.” All cells are adjacent to the lens in the Platynereis retina. I assume the authors mean, the peripheral retina. I would encourage the authors to define the axis of the camera type eye and then have an introduction to those axes early in the paper.

Also here, we hope that the clarification of optical sections and orientation of schemes has helped to make the model more comprehensive. This is particularly important as the cited literature indicates that in cephalopod camera-type eyes, neurogenesis in the retina is more disperse, happening all along the extension of the retina.

Line 315: Please include images of these mutant eyes and their wild type counterparts in Figure 4. The authors aren't counting cells, they are only measuring area. There are multiple ways that the eye could be smaller, and **this question is central to the author's hypothesis**. Is there cell death? How old are these animals?

As detailed in the answer to the next question, the focus of the displayed data and their interpretation in Fig. 4 has shifted somewhat in the revised version, incorporating reviewer responses and feedback. In response to reviewer 1, we have normalized EdU-positive cells to the eye area, thereby reducing the impact of eye size variability. Moreover, as reviewer 1 also requested more focused experiments on light requirement, the introductory experiment has been entirely redone, and now compares (normalized) EdU frequencies in the eye across light regimes and *c-opsin1* genotypes.

Line 404: “coupled to the unexpected expression of the ciliary opsin gene *c-opsin1*, modulating rhabdomeric photoreceptor cell differentiation.”

Isn't the main phenotype in the *c-opsin* mutants identified in the single cell data that there are less eye cells all together, not that differentiation is impaired in the *C-opsin* expressing cells becoming photoreceptors. There aren't more of either the support cells or the progenitor cell in the retina that would suggest that cells are stalled in a progenitor state. I would encourage the authors to be more specific about their hypothesis.

We thank the reviewer for asking us to refine our hypothesis about the role of *c-opsin1* for eye growth and differentiation and outline why we think that the eye photoreceptor

population is particularly affected. Our focus on a possible role of *c-opsin1* on the eye photoreceptor lineage not only results from the presence of *c-opsin1* in early stages of the presumptive differentiation trajectory (which suggests the possibility of cell-autonomous functions). But as the quantification in Fig. 4D shows, within the eye super-cluster c, partition pB (containing the eye photoreceptors) is the one partition in which we observe a significant reduction of cells from the mutant head samples when compared to wild-type samples, whereas changes in pA and pC are not significant. Then, Fig. 4F-I make a rather clear point about the absence of a particular “tip” of partition pB that the trajectory analysis suggests to be a differentiated state of the EPs.

As to our hypothesis, at present we favor a model in which there is a progenitor cell population that transiently expresses *c-opsin1* and – in a light- dependent manner- asymmetrically divides to give rise to a new progenitor cell and a cell that will become a differentiated rhabdomeric eye photoreceptor cell (in partition pB). We have added this now to the discussion, please see text:

“Our findings that light impacts on adult eye growth.... that photoreceptors play in developmental and adult stages of *Platynereis*”

However, as reviewer 1 has rightly cautioned that we cannot provide final proof of cell autonomy of *c-opsin1* requirement, we have not gone deeper into discussing this possible cell-autonomous role, but rather mention the possibility that there could also be non-cell autonomous roles. This appears to be the more balanced approach to interpreting the *c-opsin1* mutant phenotype. Please see text: “Whereas the co-expression of *c-opsin1* in the *r-opsin1* possibly via broader physiological changes”

Line 408-411:

“Molecularly, the gfap and ki67 orthologs we find in the worm, and their association with bona fide aNSCs, support the notion that there is an ancient molecular signature of neurogenic/ proliferative brain cells that is shared between bristleworms and vertebrates”. Please soften this language. Again, I would encourage the authors to define what they mean by an adult neural stem cell population and what it means to be bona fide or not. If there is nervous system growth, does that not qualify cells to be neural stem cells? Is there an alternative the authors are considering? Ki67 is a common proliferation marker from neural progenitors to cancer. It is an interesting finding that Ki67 may have similar function here, especially in light of the fact that cephalopods also express Ki67 in their proliferative nervous system. However, a proliferation marker would be expected. The authors did not find a specific gfap sequence. They found a homolog of many vertebrate intermediate filaments. I find this study interesting, but I think the statement is not an accurate framing of the conclusions.

In keeping with the reviewer’s suggestion, and following the arguments we have discussed above on the one-to-many orthology relationship of *Platynereis nf* and vertebrate intermediate filaments, we have removed arguments building on this relationship. For Ki-67, we consider this to be a more important finding due to the missing molecular phylogenies in cephalopods (also detailed above). We acknowledge the reviewer’s point that Ki-67 is not a specific neuronal proliferation marker (even though it is a key factor in the field of aNSC quiescence and activation).

We have reworded the respective texts accordingly. For details, please see text: “The tentative bristleworm aNSC system exhibits both similarities and differences..... or are evolutionarily divergent.”

Line 437-439: Relevant citation.

- Koenig, Kristen M., Peter Sun, Eli Meyer, and Jeffrey M. Gross. "Eye development and photoreceptor differentiation in the cephalopod *Doryteuthis pealeii*." *Development* 143, no. 17 (2016): 3168-3181.

We thank the reviewer for pointing out this interesting reference. We have now included it along with the more recent Napoli et al. study already cited in an earlier part of the text. Both make a strong argument that growth in cephalopod retinae does not appear to derive from a CMZ-like zone, at least at the tested stages.

Line 439-443 This seems like a reasonable hypothesis, and was proposed by Fischer & Brökelmann, however, the presentation of the data is difficult to interpret.

We thank the reviewer for their judgement of this as a reasonable hypothesis and hope that (i) the clearer representation of orientations in panels and schemes; (ii) the provided zoom-ins to better visualize HCR stainings, and (iii) the inclusion of additional data make our case now easier to follow.

It would be interesting for the authors to discuss the previously published and known regulatory activity of opsins in the context of their finding.

***Platynereis* is a particularly well known and established animal model for research into photobiology, as well as the impact of diverse photoreceptors, and we thank the reviewer for encouraging us to discuss this aspect in more depth. Whereas this is an area that indeed is very interesting to discuss, we believe that a too extensive discussion will go beyond the space and focus constrains of the manuscript. Furthermore, while our study provides significant new insights, new open questions arise that we could only discuss in a speculative way (e.g. cell autonomous vs. non-cell autonomous), which may draw criticism in the context of a primary research manuscript.**

We have therefore decided to only add a relatively small section on this topic to the discussion part of the manuscript. However, in this section we provide several references to additional functional studies on non-visual light receptors, and especially also conceptual recent reviews that cover more speculative thoughts.

Please see added text: “These results tie in with a diverse range of roles that photoreceptors play in developmental and adult stages of *Platynereis* plasticity that is a selective advantage in constantly changing surroundings.”

It would be interesting to talk more specifically about growth and quiescence in the context of sexual maturity and/or metamorphosis.

Like in the previous comment, we think this is a fascinating topic that deserves more discussion. We have therefore made some edits to the discussion to explain better that and why we think that the quiescent brain signature may reflect that energy in maturing animals is channeled towards growth of reproductive tissues, and to formulate testable hypotheses resulting from this suggestion. However, due to the restrictions on article length, the current study may not be the best place to expand this discussion much beyond this.

Figure Problems

Figure 1 G,H, I, J

It is difficult to distinguish the EdU labeling from background. It would help if there was a higher resolution, longer pulse or sectioned images. The data as presented is not convincing. In addition, to compare different z-stacks across datasets to show that one group has EdU incorporation, and the other group does not, the Z-stacks need to be of similar size or there needs to be statistical measurements. "72, 76, 36 and 36 μm thick, respectively"

We thank the reviewer for pointing out the issue with labeling visibility; while it appears that the issue arose due to the compromised resolution of the figures, the images the reviewer referred to have now been inverted as to better showcase the EdU signal and a corresponding supplement (Supplementary Figure 2A-H) has been introduced containing EdU labeling, nuclear stain, head contour outlines and corresponding representative negative controls for more clarity. The varying Z-stack dimensions have also been corrected in microscopy images throughout the manuscript so that all groups within one experiment are of comparable thickness. (Note that in Fig. 1L-O these represent entire head scans, but the heads are of different thickness.)

Figure 2 L and T

These summaries are not well supported by the data. I think L is a reasonable guess, but there isn't great evidence for it, I would add qualifying language in the figure legend. Figure 2T is hard to understand relative to the data shown. SoxB1 in situ is not convincing but if SoxB1 were expressed where the arrowheads are pointing, it would not be in accordance with the schematic. Maybe this is a superficial image?

This comment relates to a point already mentioned in the "Major issues" section, and we kindly refer to the general answer there. In particular, we think that the newly introduced panel E should significantly help in the orientation of the reader, and also helps to standardize language referring to the different orientation of views. The overview panels (now panels M and W) orient themselves on the classical, EM-based drawings of Fischer and Brökelmann, and we have more clearly pointed out their orientation as lateral cross sections along the eye axis (as defined in panel E), so indeed their orientation differs from the orientations of the panels, and the reviewer's comment thus has helped to make this more intelligible. Consistent with the reviewer's request for qualifying language, we also more clearly refer to the schemes as "proposed models", rather than "representations" of the shown panels, which should further help to interpret them correctly.

soxB1 stainings have been entirely replaced with acquisitions obtained from a fresh and stronger staining, adding zoom-ins to overcome the issues with pdf/jpeg conversion. Likewise, we have also added the *notch* HCR staining to support the argument with a second marker predicted by our analyses to be present in the stem cell zone. The results of these stainings are fully consistent with each of the markers being part of the stem cell population.

Figure 2 N and O

Again, to make data comparable as presented, the authors need to choose comparable images. These z stacks need to be the same thickness.

We thank the reviewer for raising this point. As indicated in the response to the comment before the previous one, we have made sure to compare images of comparable thickness throughout the manuscript.

Figure 2P and Q

The SoxBI HCR is not convincing, unfortunately. Although according to the single cell data, that may not be totally unexpected. The EdU is convincing in one cell, but it would be helpful to get more time resolution on this experiment. Is there some significance for 4 hours? Or maybe some statistics on the replicates for these data would help?

As indicated above, *soxB1* stainings have been replaced with acquisitions obtained from a fresh and stronger staining, adding zoom-ins to overcome the technical issue of invisibility of the HCR stains. *Notch* has been added as additional marker.

Figure 3H-J

The whole mount images of the c-opsin HCR are definitely convincing, but these retinal sections are not.

This is another point referring to the detection of HCR stainings. The panels in this figure have been modified accordingly, providing high-magnification views of the areas.

Figure 3L and M

The EdU integration is in more nuclei than the 4 hour pulse in figure 2. Why? Are these representative? Are all these images center sections of the eye? Are these the same age animals?

The mentioned images were taken at different positions of the eye (what is now SCS vs. LCS in Fig.2).

As outlined above, *soxB1* panels have been replaced by repetitions in which EdU detection also has worked well. Imaging was done in a very similar way to the EdU pulse panels in the same figure, and the results appear well comparable. Section planes are indicated

Figure 4 C and D

Although I appreciate that it is helpful to see distribution of wt and mutant this way, it is not a great way for the reader to understand the statistical significance of your data. If one of your samples is responsible for the majority of difference it is not representative but still appears significantly different. I would suggest the bar graph be made into control and mutant box plots and statistical significance marked or bar graphs with individual lines, not a percent total box.

We understand the reviewer’s interest in seeing the data represented in a different format and attach the respective box plots in this point-to-point response (see below: box plot 1- same Y-axis scale for all, box plot 2- Y-axis scale adjusted if interested in the smaller differences). Statistical data are identical with those already reported in the study. For the figure, the bar graphs provide a more compact view that we would prefer over the box plots, as this is also the more common representation in manuscripts containing scRNAseq data. In response to the reviewer’s comment, we have, however, now revised the bar graphs, adding the number of cells in each replicate. We reason that this is a good balance to display the raw data in a more quantitative way, without sacrificing the compactness that the bar graphs provide.

We would like to leave it up to the reviewer and editor to decide if they want us to go for revising the figure and incorporate box plots or provide the box plots as an additional supplemental figure. The reason why we have not done this immediately is that we also want to avoid to increase the amount of supplemental material too much, and thus felt we leave the decision to the discretion of the reviewer/editor.

Box Plot 1. Number of cells from wild-type (blue) versus *c-opsin1*^{-/-} (grey) heads (3 independent biological replicates each), represented as box plots.

Box Plot 2. Number of cells from wild-type (blue) versus *c-opsin1*^{-/-} (grey) heads (3 independent biological replicates each), represented as box plots with individual Y-axis scales for better visibility of smaller clusters.

MILIVOJEV ET AL. FINALIZED VERSION – POINT-TO-POINT RESPONSES TO REVIEWERS' COMMENTS

Reviewer #1 (Remarks to the Author):

The review of the initial paper was quite positive although this reviewer had issues with the developmental phenotype of the c-opsin1 mutants in the ciliary zone. The authors have addresses many of the problems raised and have explained why several of the pictures were poorly convincing, apparently due to a PDF conversion. The 36 page (!!) response to reviewers attempts to explain and to solve most of the issues. An important additional experiment that appears to confirm a potential role for c-opsin1 is the phenotype observed when the animals develop in the dark. The potential non-autonomy is however not resolved

At this point, and considering the strength of the first part of the manuscript and the fact that the authors have toned down their statements about the light dependence, this reviewer believes that the paper should be published as a significant contribution to understanding how stem cells support the development of a type of eye that grows continuously through life... and might require light!

A: We wish to thank the reviewer very much for their response, and for the helpful and constructive feedback during the review process.

Reviewer #2 (Remarks to the Author):

The Authors answered the majority of the comments and the manuscript significantly improved. I am excited to see this paper published.

A: We thank the reviewer very much for this judgement, and for the helpful and constructive feedback during the review process.

There are a few minor comments that I am going to list below for the Authors to consider:

- The title is better reflecting the story, but it is pretty long, difficult to read and a little bit vague.

A: In response to this reviewer's comments and also the formatting guidelines of the journal, we now provide a shorter title for the manuscript:

Light-modulated stem cells in the camera-type eye of an annelid model for adult brain plasticity

- I would encourage to try to connect the first and the second part of the abstract. There is a big jump between the camera-type eyes and the neuroplasticity. I am wondering if moving the concept of the neuroplasticity at the end of the abstract showing how this study is very relevant also to understand neuroplasticity could help the flow.

A: In line with this suggestion, we have changed the flow of the abstract which now mentions brain plasticity only in the end. We agree that this makes the abstract more comprehensive.

- The Supplementary figures are not referenced in order in the text.

A: This has been adjusted. Likewise, Supplementary Tables have been re-ordered to match the order in which they are referred to in the text.

- It would be very informative, in Fig 4B, to run statistical analysis on LD WT vs LD mutant and DD WT vs DD mutants. I would have expected LD mutant and DD mutant to be more similar to DD WT.

A: p values for all comparisons are now included in the panel (now: Fig. 5b).

*- It would be helpful to me to understand how *Platynereis dumerilii* is defined long-life growth since the Authors are showing a severe decrease in EdU positive cells and in the last Fig an overall decrease in cell proliferation shown through the GO term.*

A: Our statement on life-long growth refers to almost the entire lifespan of the animal which takes several months, and during which both the nervous system and body are subject to continuous growth. The transition to sexual maturation that we characterize takes place towards the end of the adult life and so indeed concerns only a minor fraction

of the entire lifetime of the animal. As we outline in the introduction, also teleosts exhibit a decline of retinal stem cells at the end of their lives, and yet are considered examples of “life-long” nervous system growth.

To make this aspect clearer without deviating too much from the main narrative, we now included the statement: “towards the end of their adult life” as an additional specifier for the reproductive transition in the discussion.

Reviewer #3 (Remarks to the Author):

I still think this paper is important and interesting with high impact.

We thank the reviewer very much for their overall assessment.

My primary concerns from the original draft were:

(1) A lack of convincing in situs

(2) Difficulty in interpreting the data as presented

(3) A need for a first principles approach to the question of growth during development

(4) A number of textual changes concerning interpretation and clarity

The authors have addressed some of my concerns but not all of them. I have included my assessment of these for clarity below.

(1) I agree with the authors that the jpg/pdf conversion did make a difference in the previous draft. I find the opsin in situ convincing. Unfortunately, I still find the SoxB1 in situ difficult to interpret and not convincing. I suggest that the authors remove it and any argument based on that expression pattern, other than bioinformatic analysis, from the paper. The new notch in situ is more convincing but it is also not robust. See (2) for additional concerns about in situs and their interpretation.

A: In response to this comment and in line with an editorial suggestion, we now removed the spatial expression data for soxB1 and instead rely on the marker notch as a proxy for the soxB1-positive cell population identified in the transcriptome map (following an earlier suggestion in the review process).

(2) The authors have now labeled their images in orientation which is a significant help to the reader. I maintain that saying NSC cells are adjacent to the lens is confusing, when all retinal cells are adjacent to the lens in this visual organ, because there is no vitreous.

In addition, and more importantly, now that the authors have made the data easier to interpret by the reader, it has become clear that the foundation of the hypothesis that there is a CMZ-like region on the periphery of the retina is not adequately shown. The authors never show that there are no Edu positive cells in the central retina. Figure 2, O and P show a whole z of the eye, but they show it in max projection. These labeled cells could be anywhere in the eye. The remaining images are superficial cross sections. At the very least, I encourage the authors to include a 3D rendering of the whole z-stack shown in 2O and 2P. The presence of the proliferative markers at the periphery of the retina is as important for their hypothesis as is the absence in the central retina. This includes the in situs and the Edu analysis. Without 3D renderings or central imaging the hypothesis doesn't hold. I understand that the PCNA has been technically difficult but I don't understand the pH3 argument. PH3 shows where cells are dividing, which is one of the fundamental arguments in the paper. An alternate option would be to remove the direct comparison to the CMZ and any conclusive reference to a restricted location of stem cells in the retina.

A: We believe that there are probably still misunderstandings concerning the position of the stem cell region we propose with respect to the eye morphology. But the reconstruction of the eye that the reviewer suggested actually helps to make this issue more comprehensible.

In line with the reviewer's request, we now include – as supplementary movies 1, 2, and 3 – animated views of 3D stacks for EdU-labeled eyes. What becomes more intuitively clear with these stacks is that indeed, the 3rd dimension of the eye (even though it is compressed due to mounting on the microscope) clearly distinguishes an area at the edge of the cup-shaped retina, at the level of the eye opening – in which the vast majority of EdU-labeled cells is located – from a part closer to the brain that has few if any labeled cells. All of this is consistent with the notion that the region around the eye opening is the major source of stem cells.

Given this reviewer's argument about anti-phospho-H3 staining as a more precise tool to visualize M phase nuclei, we would like to clarify that our model does not exclude that a cell produced by the original stem cell population would not undergo a second division, when it has been displaced to the more lateral aspects of the eye. In fact, according to relevant literature, this is also the case for retinal progenitors produced in the CMZ of the zebrafish eye, where this has been carefully mapped (see e.g. Wan et al., *Development* 2016; doi:10.1242/dev.133314).

(3) I still feel strongly the paper would benefit from a first principles approach. With all due respect to the papers that the authors referenced in the rebuttal, current tools have revolutionized the simple experiments that can assess growth and proliferation in the development of this specific visual organ. As this paper itself shows, there are plenty of unknowns to address. Very simply, if the authors had included a time course assessment of growth of the eye, and anti-ph3 in a manner that could be assessed both peripherally and centrally in the retina, the paper would be much stronger. This would also allow the authors to standardize their imaging of the central retina versus the peripheral retina.

A: We are aware that our work cuts across different aspects, each of which would justify deeper analyses, and that our experiments and tools would provide interesting starting points for such analyses. Repeating a full developmental time series of eye development would be one of these directions, and we appreciate this reviewer's interest in this topic. Given the restrictions of working time, finances, biological material, and manuscript space, we have not been able to extend our investigations into the stages for which the classical work already provides high-level ultrastructural insight. Rather, we focused on a biological aspect that had obviously been missed by the classical authors, and which at the same time exhibits compelling parallels to the strategies nature uses in teleost eye growth.

(4) The authors made a number of moderating textual changes that I think are generally fine. I still do not prefer the use of bone fide but will leave its usage up to the editor. I am also happy with the box plots in supplement.

The term “bona fide” now appears only two times in the main MS, and a single time in a figure legend of the supplementary. We would leave it there, provided the editor has no objections.